

# Quantifying new water fractions and transit time distributions using ensemble hydrograph separation: theory and benchmark tests

James W. Kirchner[1,2,3]

[1]Dept. of Environmental Systems Science, ETH Zurich, CH-8092 Zürich, Switzerland
[2]Swiss Federal Research Institute WSL, CH-8903 Birmensdorf, Switzerland
[3]Dept. of Earth and Planetary Science, University of California, Berkeley, CA, 94720, U.S.A.

*Correspondence to*: James W. Kirchner (kirchner@ethz.ch)

**Abstract.**   Decades of hydrograph separation studies have estimated the proportions of recent precipitation in streamflow
using end-member mixing of chemical or isotopic tracers.   Here I propose an ensemble approach to hydrograph separation
that uses regressions between tracer fluctuations in precipitation and discharge to estimate the average fraction of new water
(e.g., same-day or same-week precipitation) in streamflow across an ensemble of time steps.   The points comprising this
ensemble can be selected to isolate conditions of particular interest, making it possible to study how the new water fraction
varies as a function of catchment and storm characteristics.   Even when new water fractions are highly variable over time,
one can show mathematically (and confirm with benchmark tests) that ensemble hydrograph separation will accurately
estimate their average.   Because ensemble hydrograph separation is based on correlations between tracer fluctuations rather
than on tracer mass balances, it does not require that the end-member signatures are constant over time, or that all the end-
members are sampled or even known, and it is relatively unaffected by evaporative isotopic fractionation.   Ensemble
hydrograph separation can also be extended to a multiple regression that estimates the average (or "marginal") transit time
distribution directly from observational data.   This approach can estimate both "backward" transit time distributions (the
fraction of streamflow that originated as rainfall at different lag times) and "forward" transit time distributions (the fraction
of rainfall that will become future streamflow at different lag times), with and without volume-weighting.   It makes no
assumption about the shapes of the transit time distributions, nor does it assume that they are time-invariant, and it does not
require continuous time series of tracer measurements.   Benchmark tests with a nonlinear, nonstationary catchment model
confirm that ensemble hydrograph separation reliably quantifies both new water fractions and transit time distributions
across widely varying catchment behaviors, using either daily or weekly tracer concentrations as input.   Numerical
experiments with the benchmark model also illustrate how ensemble hydrograph separation can be used to quantify the
effects of rainfall intensity, flow regime, and antecedent wetness on new water fractions and transit time distributions.



## 1 Introduction

For nearly 50 years, chemical and isotopic tracers have been used to quantify the relative contributions of different water sources to streamflow following precipitation events (Pinder and Jones, 1969; Hubert et al., 1969); see also reviews by Buttle (1994) and Klaus and McDonnell (2013), and references therein. As reviewed by Klaus and McDonnell (2013),

chemical and isotopic hydrograph separation studies have led to many important insights into runoff generation. Foremost among these has been the realization that even at stormflow peaks, stream discharge is often composed primarily of "old" catchment storage rather than "new" recent precipitation (Sklash et al., 1976; Sklash, 1990; Neal and Rosier, 1990; Buttle, 1994). The previous dominant paradigm, based on little more than intuition, had held that because streamflow responds promptly to rainfall, the storm hydrograph must consist primarily of precipitation that reaches the channel quickly. Isotope

hydrograph separations showed that this intuition is often wrong, because the isotopic signatures of stormflow often resemble baseflow or groundwater rather than recent precipitation. These observations have not only overthrown the previous dominant paradigm, but also launched decades of research aimed at unraveling the paradox of how catchments store water for weeks or months, but release it within minutes following the onset of rainfall (Kirchner, 2003).

The foundations of conventional two-component hydrograph separation are straightforward. If one assumes that streamflow is a mixture of two end-members of fixed composition, which I will call for simplicity "new" and "old" water, then at any time $j$ the mass balance for the water itself is

$$Q_j = Q_{\text{new}_j} + Q_{\text{old}_j} \quad , \tag{1}$$

and the mass balance for a conservative tracer is

$$Q_j C_{\text{Q}_j} = Q_{\text{new}_j} C_{\text{new}} + Q_{\text{old}_j} C_{\text{old}} \quad , \tag{2}$$

where $Q$ denotes water flux and $C$ denotes the concentration of a passive chemical tracer or the δ value of $^{18}$O or $^{2}$H. One can straightforwardly solve Eqs. (1) and (2) to express the fraction of new water in streamflow at any time $j$ as:

$$F_{\text{new}_J} = \frac{Q_{\text{new}_j}}{Q_j} = \frac{C_{\text{Q}_j} - C_{\text{old}}}{C_{\text{new}} - C_{\text{old}}} \quad . \tag{3}$$

In typical applications, the "new" water is recent precipitation and the tracer signature of the "old" water is obtained from

pre-event baseflow, which is generally assumed to originate from long-term groundwater storage.

The assumptions underlying conventional hydrograph separation can be summarized as follows:

1. Streamflow is a mixture formed entirely from the sampled end-members; contributions from other possible streamflow sources (such as vadose zone water or surface storage) are negligible.





2. The samples of the end-members are representative (e.g., the sampled precipitation accurately reflects all precipitation, and the sampled baseflow reflects all pre-event water).

3. The tracer signatures of the end-members are constant through time, or their variations can be taken into account.

4. The tracer signatures of the end-members are significantly different from one another.

5 As reviewed by Rodhe (1987), Sklash (1990), Buttle (1994) and Klaus and McDonnell (2013), each of these assumptions can be problematic in practice:

1. Hydrograph separation studies often lead to implausible (including negative) inferred contributions of new water, and such anomalous results are sometimes attributed to contributions from un-sampled end-members (e.g., von Freyberg et al., 2017). In such cases, assumption #1 is clearly not met.

10 2. The isotopic composition of precipitation can vary considerably within an event, both spatially and temporally, even in small catchments (e.g., McDonnell et al., 1990; McGuire et al., 2005; Fischer et al., 2017; von Freyberg et al., 2017). Likewise, the isotopic signature of the baseflow or groundwater end-member has been shown to vary in space and time during snowmelt and rainfall events (e.g., Hooper and Shoemaker, 1986; Rodhe, 1987; Bishop, 1991; McDonnell et al., 1991). In these cases, assumptions #2 and #3 are not met. Various schemes have been 15 proposed to address this spatial and temporal variability by weighting the isotopic compositions of individual samples, but the validity of these schemes typically rests on strong assumptions about the nature of the runoff generation process and the heterogeneity to be averaged over.

3. When the difference between $C_{\text{new}}$ and $C_{\text{old}}$ is not large compared to their uncertainties, Eq. (3) becomes unstable and the resulting hydrograph separations become unreliable. This problem can be detected using Gaussian error 20 propagation (Genereux, 1998), but Bansah and Ali (2017) report that less than 20% of the hydrograph separation studies they reviewed have used it.

One can agree with Buttle (1994) that "despite frequent violations of some of its underlying assumptions, the isotopic hydrograph separation approach has proven to be sufficiently robust to be applied to the study of runoff generation in an 25 increasing number of basins," at least as a characterization of the community's widespread acceptance of the technique. Nonetheless, there is clearly room for new and different ways to quantify stormflow generation. In addition, weekly or even daily isotope measurements are now becoming available for many catchments, sometimes spanning periods of many years, and despite their many uses (particularly for calibrating hydrological models), there is an obvious need for new ways to extract hydrological insights from such time series.

Here I propose a new method for using isotopes and other conservative tracers to quantify the origins of streamflow. This method is based on statistical correlations among tracer fluctuations in streamflow and one or more candidate water sources, rather than mass balances. As such, it exploits the temporal variability in candidate end-members, rather than requiring them to be constant. It also does not require strict mass balance, and thus is relatively insensitive to the presence of unmeasured



end-members. Because this method quantifies the average proportions of source waters in streamflow across an ensemble of events or time steps, it does not answer the same question that traditional hydrograph separation does (namely, how fractions of new and old water change over time during individual storm events). Instead, it can answer new and different questions, such as how the average fractions of new and old water vary with stream discharge or precipitation intensity, antecedent

moisture, etc. The proposed method is designed to provide insights into stormflow generation from regularly sampled time series, even if those time series have gaps and even if they are sampled at frequencies much lower than the storm response timescale of the catchment.

The purpose of this paper is to describe the method, document its mathematical foundations, and test it against a benchmark

model, in which the method's results can be verified by age tracking. Applications to real-world catchments will follow in future papers. Because the proposed method is new and thus must be fully documented, several parts of the presentation (most notably Sects. 4.2-4.4 and Appendix B) necessarily contain strong doses of math. The math can be skipped, or lightly skimmed, by those who only need a general sense of the analysis. A table of symbols is provided at the end of the text.

## 2 Estimating new water fractions by ensemble hydrograph separation

Here I propose a new type of hydrograph separation based on correlations between tracer fluctuations in streamflow and in one or more end-members. This new approach to hydrograph separation does not have the same goal as conventional hydrograph separation. It does not estimate the contributions of end-members to streamflow for each time step (as in Eq. 3). Instead, it estimates the *average* end-member contributions to streamflow over an ensemble of time steps; hence its name, *ensemble* hydrograph separation. The ensemble of time steps may be chosen to reflect different catchment conditions, and

thus used to map out how those catchment conditions influence end-member contributions to streamflow.

### 2.1 Basic equations

I will first illustrate this approach with a simple example of a time-varying mixing model. Let's assume that we have measured tracer concentrations in streamflow, and in at least one contributing end-member, over an ensemble of time intervals $j$. The simplest possible mass balance for the water that makes up streamflow would be

$$Q_j = Q_{\mathrm{new}_j} + Q_{\mathrm{old}_j} \qquad , \tag{4}$$

where $Q_{\mathrm{new}}$ represents the water flux in streamflow $Q$ that originates from recent precipitation (or, potentially, any other end-member in which tracers can be measured) during time interval $j$. All other contributions to streamflow are lumped together as $Q_{\mathrm{old}}$. Conservative mixing implies that

$$Q_j\, C_{\mathrm{Q}_j} = Q_{\mathrm{new}_j}\, C_{\mathrm{new}_j} + Q_{\mathrm{old}_j}\, C_{\mathrm{old}_j} \qquad , \tag{5}$$





where $C_Q$ and $C_{new}$ are the tracer concentrations in the stream and the new water, and $C_{old}$ is the tracer signature of all other sources that contribute to streamflow. Combining (4) and (5), we directly obtain

$$C_{Q_j} = F_{new_j} C_{new_j} + \left(1 - F_{new_j}\right) C_{old_j} \quad , \tag{6}$$

where $F_{new_j} = Q_{new_j}/Q_j$ is the fractional contribution of $Q_{new}$ to streamflow $Q$. Equation (6) can be rewritten as

$$C_{Q_j} - C_{old_j} = F_{new_j} \left(C_{new_j} - C_{old_j}\right) \quad , \tag{7}$$

which in turn could be rearranged as a conventional mixing model (Eq. 3), with the important difference that the new and old water concentrations are time-varying rather than constant. If we represent the "old" water composition using the streamwater concentration during the previous time step, equation (7) becomes

$$C_{Q_j} - C_{Q_{j-1}} = F_{new_j} \left(C_{new_j} - C_{Q_{j-1}}\right) \quad , \tag{8}$$

The lagged concentration $C_{Q_{j-1}}$ serves as a reference level for measuring fluctuations in precipitation and streamflow tracer concentrations and the correlations between them. Thus, it is not necessary that $C_{Q_{j-1}}$ consists entirely of "old water" as defined in conventional hydrograph separations (i.e., groundwater or baseflow water). It is only necessary that $C_{Q_{j-1}}$ contains no "new" water (that is, no precipitation that fell during time step $j$), and this condition is automatically met because $C_{Q_{j-1}}$ is determined during the previous time step.

The ensemble hydrograph separation approach is based on the observation that (8) is similar to the conventional linear regression equation,

$$y_j = \beta\, x_j + \alpha + \varepsilon_j \quad , \quad y_j = C_{Q_j} - C_{Q_{j-1}} \quad , \quad x_j = C_{new_j} - C_{Q_{j-1}} \quad , \tag{9}$$

where the intercept $\alpha$ and the error term $\varepsilon_j$ can be viewed as subsuming any bias or random error introduced by measurement
noise, evapoconcentration effects, and so forth. The analogy between (9) and (8) suggests that it may be possible to estimate the average value of $F_{new_j}$ from the regression slope of a scatterplot of the streamflow concentration $C_{Q_j}$ against the new water concentration $C_{new_j}$, both expressed relative to the lagged streamflow concentration $C_{Q_{j-1}}$.

However, astute readers will notice an important difference between (8) and (9): the regression slope $\beta$ is a constant in (9),
whereas in (8), $F_{new_j}$ varies from one time step to the next. It is not obvious how an estimate of the (constant) slope $\beta$ will be related to the (non-constant) $F_{new_j}$ or whether this relationship could be affected by the other variables in Eq. (8). The answer to this question can be derived analytically and tested using numerical experiments (see Appendix A). As explained





in Appendix A, the regression slope in a scatterplot of $C_{Q_j} - C_{Q_{j-1}}$ versus $C_{new_j} - C_{Q_{j-1}}$ (Fig. A1d) will closely approximate the average value of $F_{new_j}$ (averaged over the ensemble of time steps $j$), under rather general conditions:

1. The slope of the relationship between $F_{new_j}$ and $C_{new_j} - C_{Q_{j-1}}$, times the mean of $C_{new_j} - C_{Q_{j-1}}$, should be small. This will usually be true for conservative tracers, for two reasons. First, because all streamflow is ultimately

derived from new water, mass conservation implies that the mean of $C_{new_j} - C_{Q_{j-1}}$ should be nearly zero. Second, unless there is a correlation between storm size and tracer concentration (not just between storm size and tracer variance), the slope of the relationship between $F_{new_j}$ and $C_{new_j} - C_{Q_{j-1}}$ should also be small. Thus the product of these two small terms should be small.

2. Points with large leverage in the scatterplot (i.e., with $C_{new_j} - C_{Q_{j-1}}$ values far above and below the mean) should

not be systematically associated with either high or low values of $F_{new_j}$. Such a systematic association is unlikely unless large storms (which are likely to generate large new water fractions) are also associated with both very high and very low tracer concentrations.

3. As expected for typical sampling and measurement errors, the error term $\varepsilon_j$ should not be strongly correlated with $C_{new_j} - C_{Q_{j-1}}$.

Thus the analysis in Appendix A shows that an ensemble average estimate of $F_{new}$ should, under typical conditions, be obtainable from the regression slope $\hat{\beta}$ of a plot of $x_j = C_{Q_j} - C_{Q_{j-1}}$ versus $y_j = C_{new_j} - C_{Q_{j-1}}$ (i.e., Eq. 9; Fig. A1d).

The least-squares solution of Eq. (9) can be expressed in several equivalent ways. For consistency with the analysis that will be developed in Sect. 4 below, I will use the following formulation, which is mathematically equivalent to those more

commonly seen:

$$F_{new} = \hat{\beta} = \frac{\text{cov}(y_j, x_j)}{\text{var}(x_j)} \quad , \tag{10}$$

where $\hat{\beta}$ is the least-squares estimate of $\beta$, and $F_{new}$ is the average of the $F_{new_j}$ over the ensemble of points $j$. Values of $y_j$ that lack a corresponding $x_j$, or vice versa (due to sampling gaps, for example, or lack of precipitation), are omitted.

**2.2 Uncertainties**

The uncertainty in $F_{new}$, expressed as a standard error, can be written as

$$\text{s. e. }(F_{new}) = \text{s. e.}\left(\hat{\beta}\right) = \frac{s_y}{s_x} \frac{\sqrt{1 - r_{xy}^2}}{\sqrt{n_{eff} - 2}} = \frac{\hat{\beta}}{\sqrt{n_{eff} - 2}} \sqrt{\frac{1}{r_{xy}^2} - 1} \quad , \tag{11}$$





where $s_x$ and $s_y$ are the standard deviations of $x$ and $y$, $r_{xy}$ is the correlation between them, and $n_{\text{eff}}$ is the effective sample size, which can be adjusted to account for serial correlation in the residuals (Bayley and Hammersley, 1946; Brooks and Carruthers, 1953; Matalas and Langbein, 1962):

$$n_{\text{eff}} \approx n_{\text{xy}} \left[ \frac{1 + r_{\text{sc}}}{1 - r_{\text{sc}}} - \frac{2}{n_{\text{xy}}} \frac{r_{\text{sc}}(1 - r_{\text{sc}}^n)}{(1 - r_{\text{sc}})^2} \right]^{-1} \quad , \tag{12}$$

where $n_{\text{xy}}$ is the number of pairs of $x_j$ and $y_j$, and $r_{\text{sc}}$ is the lag-1 serial correlation in the regression residuals $y_j - \hat{\beta} x_j - \alpha$. For large $n_{\text{xy}}$, Eq. (12) can be approximated as (Mitchell et al., 1966)

$$n_{\text{eff}} \approx n_{\text{xy}} \left[ \frac{1 - r_{\text{sc}}}{1 + r_{\text{sc}}} \right] \quad , \tag{13}$$

where for all positive $r_{\text{sc}}$, Eq. (13) is conservative (it underestimates $n_{\text{eff}}$ from Eq. 12), and for $r_{\text{sc}} = 0.5$, and $n_{\text{xy}} > 50$, for example, Eqs. (12) and (13) differ by less than 3%. If the scatterplot of $y_j = C_{Q_j} - C_{Q_{j-1}}$ versus $x_j = C_{\text{new}_j} - C_{Q_{j-1}}$

contains outliers, a robust fitting technique such as Iteratively Reweighted Least Squares (IRLS) may yield more reliable estimates of $F_{\text{new}}$ than ordinary least squares regression. However, the analyses presented here are based on outlier-free synthetic data generated from a benchmark model (see Sect. 3), so in this paper I have used conventional least squares (Eqs. 10-11) instead.

### 2.3 New water fraction for time steps with precipitation

The meaning of the new water fraction $F_{\text{new}}$ depends on how the new water and streamwater are sampled. For example, if the new water concentrations $C_{\text{new}}$ are measured in daily bulk precipitation samples and the stream water concentrations $C_Q$ are measured in instantaneous grab samples taken at the end of each 24-hour precipitation sampling period, then $F_{\text{new}}$ will estimate the average fraction of streamflow that is composed of precipitation from the preceding 24 hours. If the sampling interval is weekly instead of daily, then $F_{\text{new}}$ will estimate the average fraction of streamflow that consists of precipitation

from the preceding week. This will generally be larger than the $F_{\text{new}}$ calculated from daily sampling, for the obvious reason that on average more precipitation will have fallen during the previous week than during the previous 24 hours, so this precipitation will comprise a larger fraction of streamflow. Also, if the weekly streamflow concentrations are measured in integrated composite samples rather than instantaneous grab samples, then $F_{\text{new}}$ will estimate the fraction of same-week precipitation in average weekly streamflow rather than in the instantaneous end-of-week streamflow. The general rule is:

$F_{\text{new}}$ should generally estimate whatever new water has been sampled as $C_{\text{new}}$, expressed as a fraction of whatever streamflow has been sampled as $C_Q$.





In all of these cases, $\hat{\beta}$ from Eq. (10) estimates the average fraction of new water in streamflow during time steps with precipitation, because time steps without precipitation lack a "new water" tracer concentration $C_{\text{new}_j}$ and thus must be left out from the regression in Eq. (9). Using Qp to denote discharge during periods with precipitation, we can represent this "event new water fraction" as $^{\text{Qp}}F_{\text{new}}$.

### 5  2.4 New water fraction for all time steps

Periods without precipitation will inherently lack same-day (or same-week) precipitation in streamflow. Thus we can calculate the average fraction of new water in streamflow during all time steps, including those without precipitation, as

$$^{Q}F_{\text{new}} = \; ^{\text{Qp}}F_{\text{new}} \frac{n_{\text{p}}}{n} = \; \hat{\beta}\, \frac{n_{\text{p}}}{n} \quad , \tag{14}$$

where $^{Q}F_{\text{new}}$ is the new water fraction of all discharge, $^{\text{Qp}}F_{\text{new}}$ is the new water fraction of discharge during time steps with precipitation (as estimated by the regression slope $\hat{\beta}$, from Eq. 10), and $n_{\text{p}}/n$ is the fraction of time steps that have precipitation. The ratio $n_{\text{p}}/n$ in Equation (14) accounts for the fact that during time steps without rain, the new-water contribution to stream flow is inherently zero. The same ratio is also used to estimate the uncertainty in $^{Q}F_{\text{new}}$:

$$\text{s. e.}\left( ^{Q}F_{\text{new}} \right) = \frac{n_{\text{p}}}{n} \; \text{s. e.}\left( \hat{\beta} \right) = \frac{^{Q}F_{\text{new}}}{\sqrt{n_{\text{eff}} - 2}} \sqrt{\frac{1}{r_{xy}^2} - 1} \quad . \tag{15}$$

### 2.5 Volume-weighted new water fractions

The regression derived through Eqs. (4)-(9) gives each time interval $j$ equal weight. As a result, $\hat{\beta}$ from Eq. (10) can be interpreted as estimating the time-weighted average new water fraction. To estimate the volume-weighted new water fraction instead, one simply weights the regression equation by discharge. To do so, one weights $x_j$ and $y_j$ in Eqs. (9)-(10) by $\sqrt{Q_j}$ :

$$y_j = \beta^* x_j + \alpha + \varepsilon_j \quad , \quad y_j = \sqrt{Q_j}\left(C_{Q_j} - C_{Q_{j-1}}\right) \quad , \quad x_j = \sqrt{Q_j}\left(C_{\text{new}_j} - C_{Q_{j-1}}\right) \quad . \tag{16}$$

Weighting $x_j$ and $y_j$ by $\sqrt{Q_j}$ has the effect of weighting the residual sum of squares (which is minimized in least-squares regression) by $Q_j$ (equivalently, it also weights the covariance and variance in Eq. 10 by $Q_j$). We can denote the resulting regression slope $\hat{\beta}^*$ as $^{\text{Qp}}F_{\text{new}}^*$, the volume-weighted new water fraction of time intervals with precipitation, where (following von Freyberg et al., manuscript in review) the asterisk indicates volume-weighting.



If, instead, one wants to estimate the new water fraction in all discharge (during periods with and without precipitation), following the approach in Sect. 2.4 one simply rescales this regression slope by the sum of discharge during time steps with precipitation, divided by total discharge:

$$^{Q}F_{new}^{*} = {}^{Qp}F_{new}^{*} \frac{\bar{Q}_{p}}{\bar{Q}} \frac{n_{p}}{n} = \hat{\beta}^{*} \frac{\bar{Q}_{p}}{\bar{Q}} \frac{n_{p}}{n} \quad , \tag{17}$$

where $^{Q}F_{new}^{*}$ is the volume-weighted new water fraction of all discharge, $^{Qp}F_{new}^{*}$ is the fitted regression slope $\hat{\beta}$ from Eq. (16), $\bar{Q}_{p}$ is the average discharge for time steps with precipitation, $\bar{Q}$ is the average discharge for all time steps (including during rainless periods), and $n_{p}/n$ is the fraction of time steps with rain.

Because the volume-weighting will typically be uneven, the effective sample size will typically be smaller than $n$; for
example, in the extreme case that one sample had nearly all the weight and the other samples had nearly none, the effective sample size would be roughly 1 instead of $n_{xy}$. Thus, uncertainty estimates for these volume-weighted new water fractions should take account of the unevenness of the weighting. One can account for uneven weighting by calculating the effective sample size, following Kish (1995), as:

$$n_{eff} = \frac{\left(\sum Q_{j(xy)}\right)^{2}}{\sum\left(Q_{j(xy)}^{2}\right)} \quad , \tag{18}$$

where the notation $Q_{j(xy)}$ indicates discharge at time steps $j$ for which pairs of $x_{j}$ and $y_{j}$ exist. Equation (18) evaluates to $n_{xy}$ (as it should) in the case of evenly weighted samples, and declines toward 1 (as it should) if a single sample has much greater weight than the others. To obtain an estimate of the effective sample size that accounts for both serial correlation and uneven weighting, one can multiply the expressions in Eqs. (18) and (12) or (13). Combining these approaches, one can estimate the standard error of $^{Q}F_{new}^{*}$ as

$$\text{s.e.}\left(^{Q}F_{new}^{*}\right) = \frac{\sum Q_{p}}{\sum Q} \text{s.e.}\left(\hat{\beta}^{*}\right) = \frac{^{Q}F_{new}^{*}}{\sqrt{n_{eff}-2}}\sqrt{\frac{1}{r_{xy}^{2}}-1} \quad = \quad , \quad n_{eff} = \frac{\left(\sum Q_{j(xy)}\right)^{2}}{\sum\left(Q_{j(xy)}^{2}\right)}\left[\frac{1-r_{sc}}{1+r_{sc}}\right] \quad , \tag{19}$$

where $\hat{\beta}^{*}$ is the fitted regression slope from Eq. (16).

**2.6 New water fraction of precipitation**

One can also express the flux of new water as a fraction of precipitation rather than discharge. Recently, von Freyberg et al. (manuscript in review) have noted, in the context of conventional hydrograph separation, that expressing event water as a
proportion of precipitation rather than discharge may lead to different insights into catchment storm response. Analogously,



within the ensemble hydrograph separation framework we can estimate the "new water fraction of precipitation", denoted $^\mathrm{P}F_\mathrm{new}$, as

$$^\mathrm{P}F_\mathrm{new} = \ {}^\mathrm{Qp}F_\mathrm{new}\frac{\bar{Q}_\mathrm{p}}{\bar{P}_\mathrm{p}} \quad , \tag{20}$$

where $^\mathrm{Qp}F_\mathrm{new}$ is the new water fraction of discharge during time steps with precipitation (as estimated by the regression slope

$\hat{\beta}$, from Eq. 10), and $\bar{Q}_\mathrm{p}$ and $\bar{P}_\mathrm{p}$ are the average discharge and precipitation during these time steps. An alternative strategy is to re-cast Eq. (8) by multiplying both sides by $Q_j/P_j$, such that the $F_\mathrm{new}$ on the right-hand-side now expresses new water as a fraction of precipitation,

$$\frac{Q_j}{P_j}\left(C_{Q_j} - C_{Q_{j-1}}\right) = \left(\frac{Q_j}{P_j}\,F_{\mathrm{new}_j}\right)\left(C_{\mathrm{new}_j} - C_{Q_{j-1}}\right) = \ {}^\mathrm{P}F_{\mathrm{new}_j}\ \left(C_{\mathrm{new}_j} - C_{Q_{j-1}}\right) \quad . \tag{21}$$

This yields a linear regression similar to Eq. (9), but with $y_j$ re-scaled,

$$y_j = \beta\,x_j + \alpha + \varepsilon_j \quad , \quad y_j = \frac{Q_j}{P_j}\left(C_{Q_j} - C_{Q_{j-1}}\right) \quad , \quad x_j = \left(C_{\mathrm{new}_j} - C_{Q_{j-1}}\right) \quad , \tag{22}$$

where the regression slope $\hat{\beta}$, which can be calculated from Eq. (10) with the new values $y_j$, should approximate the average new water fraction of precipitation $^\mathrm{P}F_\mathrm{new}$.

The approaches represented by Eqs. (20) and (21)-(22) are not equivalent. Equation (20) is based on the *ad hoc* assumption

– which is verified by the benchmark tests in Sects. 3.3-3.5 – that the average of $^\mathrm{P}F_{\mathrm{new}_j}$ (new water in streamflow, as a fraction of precipitation) should approximate the average $F_{\mathrm{new}_j}$ (new water in streamflow, as a fraction of discharge), rescaled by the ratio of average discharge $Q_{\mathrm{p}_j}$ to average precipitation $P_{\mathrm{p}_j}$. This is only an approximation, of course; it relies on the approximation that appears in the middle of the following chain of expressions,

$$^\mathrm{P}F_\mathrm{new} = \ \langle\,^\mathrm{P}F_{\mathrm{new}_j}\rangle_\mathrm{p} = \langle F_{\mathrm{new}_j}\frac{Q_j}{P_j}\rangle_\mathrm{p} \approx \langle F_{\mathrm{new}_j}\rangle_\mathrm{p}\,\frac{\langle Q_j\rangle_\mathrm{p}}{\langle P_j\rangle_\mathrm{p}} = \ {}^\mathrm{Qp}F_\mathrm{new}\frac{\bar{Q}_\mathrm{p}}{\bar{P}_\mathrm{p}} \quad , \tag{23}$$

where the "p" subscripts on the angled brackets indicate averages taken only over time intervals with precipitation. Whether this is a good approximation will depend on how $P_j$, $Q_j$, and $F_{\mathrm{new}_j}$ are distributed, and how they are correlated with one another. By contrast, the approach outlined in Eqs. (21)-(22) is based on the exact substitution of $F_{\mathrm{new}_j}\,Q_j/P_j$ for $^\mathrm{P}F_{\mathrm{new}_j}$, which requires no approximations. The same substitution also leads to two other algebraically equivalent formulations of Eq. (21),



$$\left(C_{Q_j} - C_{Q_{j-1}}\right) = {}^{P}F_{\mathrm{new}_j} \frac{P_j}{Q_j}\left(C_{\mathrm{new}_j} - C_{Q_{j-1}}\right) \tag{24}$$

and

$$Q_j\left(C_{Q_j} - C_{Q_{j-1}}\right) = {}^{P}F_{\mathrm{new}_j}\, P_j\left(C_{\mathrm{new}_j} - C_{Q_{j-1}}\right) \quad . \tag{25}$$

But although Eqs. (21), (24), and (25) are algebraically equivalent, their statistical behavior is different when they are used

as regression equations to estimate the average value of ${}^{P}F_{\mathrm{new}}$. The regression estimate of ${}^{P}F_{\mathrm{new}}$ depends on the distributions of $P_j$, $Q_j$, and $F_{\mathrm{new}_j}$ and their correlations with each other, and benchmark testing shows that Eq. (21) yields reasonably accurate estimates of ${}^{P}F_{\mathrm{new}}$, but Eqs. (24) and (25) do not. One can also note that the approach outlined in Eq. (20) – the other approach that is successful in benchmark tests – represents an ad-hoc time averaging of $P_j$ and $Q_j$ in Eq. (21), because it is formally equivalent to

$$\frac{\bar{Q}_{\mathrm{p}}}{\bar{P}_{\mathrm{p}}}\left(C_{Q_j} - C_{Q_{j-1}}\right) = {}^{P}F_{\mathrm{new}_j}\left(C_{\mathrm{new}_j} - C_{Q_{j-1}}\right) \quad . \tag{26}$$

The precise interpretation of ${}^{P}F_{\mathrm{new}}$ depends on how streamflow is sampled. If the streamflow tracer concentrations come from integrated composite samples over each day or week, then ${}^{P}F_{\mathrm{new}}$ can be interpreted as the fraction of precipitation that becomes same-day or same-week streamflow. If the streamflow tracer concentrations instead come from instantaneous grab

samples (as is more typical), then ${}^{P}F_{\mathrm{new}}$ can be interpreted as the rate of new water discharge at that time (typically the end of the precipitation sampling interval), as a fraction of the average rate of precipitation. Adapting terminology from the literature of transit time distributions, we can call ${}^{P}F_{\mathrm{new}}$ the "forward" new water fraction because it represents the fraction of precipitation that will exit the catchment soon (during the same time step), and call ${}^{Qp}F_{\mathrm{new}}$ and ${}^{Q}F_{\mathrm{new}}$ "backward" new water fractions because they represent the fraction of streamflow that entered the catchment a short time ago. Although the

"backward" new water fraction of discharge comes in two forms (${}^{Qp}F_{\mathrm{new}}$ or ${}^{Q}F_{\mathrm{new}}$), depending on whether one includes or excludes rainless periods, the "forward" new water fraction ${}^{P}F_{\mathrm{new}}$ can only be defined for time steps with precipitation (otherwise ${}^{P}F_{\mathrm{new}}$ represents the ratio between zero new water and zero precipitation, and thus is undefined).

### 2.7 Volume-weighted new water fraction of precipitation

The new water fraction of precipitation as estimated by Eq. (20) is a time-weighted average, in which each day with

precipitation counts equally. One may also want to estimate the volume-weighted new water fraction of precipitation, which we can denote as ${}^{P}F_{\mathrm{new}}^{*}$, in keeping with the naming conventions used above. We can estimate ${}^{P}F_{\mathrm{new}}^{*}$ at least two different ways. The first method involves recognizing that we are seeking the ratio between the total volume of "new water" – that is,





same-day precipitation reaching streamflow – and the total volume of precipitation. This will equal the volume-weighted new water fraction of *discharge* (total new water divided by total discharge, which has already been derived in Sect. 2.5 above), rescaled by the ratio of total discharge to total precipitation:

$$^{\mathrm{P}}F_{\mathrm{new}}^{*} \; = \; ^{\mathrm{Q}}F_{\mathrm{new}}^{*} \, \frac{\bar{Q}}{\bar{P}} \; = \; ^{\mathrm{Qp}}F_{\mathrm{new}}^{*} \, \frac{\bar{Q}_{\mathrm{p}}}{\bar{P}} \, \frac{n_{\mathrm{p}}}{n} \quad . \tag{27}$$

where $\bar{Q}$ and $\bar{P}$ are the average rates of discharge and precipitation (averaged over all time steps), $\bar{Q}_{\mathrm{p}}$ is the average discharge on days with rain, and $n_{\mathrm{p}}/n$ is the fraction of time steps with rain. An alternative strategy, which yields nearly equivalent results in benchmark tests, precipitation-weights the regression for $^{\mathrm{P}}F_{\mathrm{new}}$ (Eq. 21) by multiplying both sides by $\sqrt{P_j}$ . This yields a linear regression of the form of Eq. (9), but with $x_j$ and $y_j$ re-scaled:

$$y_j = \beta^* x_j + \alpha + \varepsilon_j \quad , \quad y_j = \frac{Q_j}{\sqrt{P_j}} \left( C_{\mathrm{Q}_j} - C_{\mathrm{Q}_{j-1}} \right) \quad , \quad x_j = \sqrt{P_j} \left( C_{\mathrm{new}_j} - C_{\mathrm{Q}_{j-1}} \right) \quad , \tag{28}$$

where the regression slope $\hat{\beta}^*$ should approximate the volume-weighted average new water fraction of precipitation $^{\mathrm{P}}F_{\mathrm{new}}^{*}$.

### 3 Testing ensemble hydrograph separation with a simple non-stationary benchmark model

#### 3.1 Benchmark model

To test the methods outlined in Sect. 2 above, I use synthetic data generated by a simple two-box lumped-parameter catchment model. This model is documented in greater detail in Kirchner (2016b), and will be described only briefly here.

As shown in Fig. 1a, drainage ($L$) from the upper box is a power function of the storage ($S_{\mathrm{u}}$) within the box; a fraction ($\eta$) of this drainage flows directly to streamflow, and the complementary fraction 1-$\eta$ recharges the lower box, which drains to streamflow at a rate $Q_l$ that is a power function of its storage $S_l$. The model's behavior is determined by five parameters: the equilibrium storage levels $S_{\mathrm{u,ref}}$ and $S_{l,\mathrm{ref}}$ in the upper and lower boxes, their drainage exponents $b_{\mathrm{u}}$ and $b_l$, and the drainage partitioning coefficient $\eta$. For simplicity, evapotranspiration is not simulated (alternatively, the precipitation inputs can be

considered to be effective precipitation, net of evapotranspiration losses). Discharge from both boxes is assumed to be non-age-selective, meaning that discharge is taken proportionally from each part of the age distribution. Tracer concentrations and mean ages are tracked under the assumption that the boxes are each well-mixed but also distinct from one another, so their tracer concentrations and water ages will differ. Water ages and tracer concentrations are also tracked in daily age bins up to an age of 70 days, and mean water ages are tracked in both the upper and lower boxes.

The model operates at a daily time step, with the storage evolution of the lower box calculated by a weighted combination of the partly implicit trapezoidal method (for greater accuracy) and the fully implicit backward Euler method (for guaranteed stability). Unlike in Kirchner (2016b), here the storage evolution of the upper box is calculated by forward Euler integration



at 50 sub-daily time steps of 0.02 days (roughly 30 minutes) each.  At this time step, forward Euler integration is stable across the entire parameter ranges used in this paper, and is more accurate than daily time steps of trapezoidal or backward Euler integration (which are still adequate for the lower box, where storage volumes change more slowly).  Following Kirchner (2016b), the model is driven with three different real-world daily rainfall time series, representing a range of

climatic regimes: a humid maritime climate with frequent rainfall and moderate seasonality (Plynlimon, Wales; Köppen climate zone Cfb), a Mediterranean climate marked by wet winters and very dry summers (Smith River, California, USA; Köppen climate zone Csb), and a humid temperate climate with very little seasonal variation in average rainfall (Broad River, Georgia, USA; Köppen climate zone Cfa).  Synthetic daily precipitation tracer (deuterium) concentrations are generated randomly from a normal distribution with a standard deviation of 20 per mil and a lag-one serial correlation of 0.5,

superimposed on a seasonal cycle with an amplitude of 10 per mil.  The model is initialized at the equilibrium storage levels $S_{u,ref}$ and $S_{l,ref}$, with age distributions and tracer concentrations corresponding to steady-state equilibrium values at the mean input fluxes of water and tracer.  The model is then run for a one-year spin-up period; the results reported here are from five-year simulations following this spin-up period.

For the simulations shown here, the drainage exponents $b_u$ and $b_l$ are randomly chosen from uniform distributions of logarithms spanning the range of 1-20, and the partitioning coefficient $\eta$ is randomly chosen from a uniform distribution ranging from 0.1 to 0.9.  The reference storage levels $S_{u,ref}$ and $S_{l,ref}$ are randomly chosen from a uniform distribution of logarithms spanning the ranges of 50-200 mm and 200-2000 mm, respectively.  These parameter distributions encompass a wide range of possible behaviors, including both strong and damped response to rainfall inputs.

I illustrate the behavior of the model using two particular parameter sets, one that gives damped response to precipitation ($S_{u,ref}$ =100 mm, $S_{l,ref}$ =1000 mm, $b_u$=10, $b_l$=3, and $\eta$=0.3), and one that gives a more rapid response (the same parameters, except $\eta$=0.8).  These parameter values are not preferable to others in any particular way; they simply generate strongly contrasting streamflow and tracer responses that look plausible as examples of small catchment behavior.  They can be

interpreted as the behavior of two contrasting model catchments, which for simplicity (but with some linguistic imprecision) I will call the "damped catchment" and the "flashy catchment", as shorthand for "model catchment with parameters giving more damped response" and "model catchment with parameters giving more flashy response".

The model also simulates the sampling process and its associated errors.  I assume that tracer concentrations cannot be

measured when precipitation rates are below a threshold of $P_{threshold}$=1 mm/day, such that tracer samples below this threshold will be missing.  I further assume that 5% of all other precipitation tracer measurements, and 5% of all streamflow tracer measurements, will be lost at random times due to sampling or analysis failures.  I have also added Gaussian random errors (with a standard deviation of 1 per mil) to all tracer measurements.



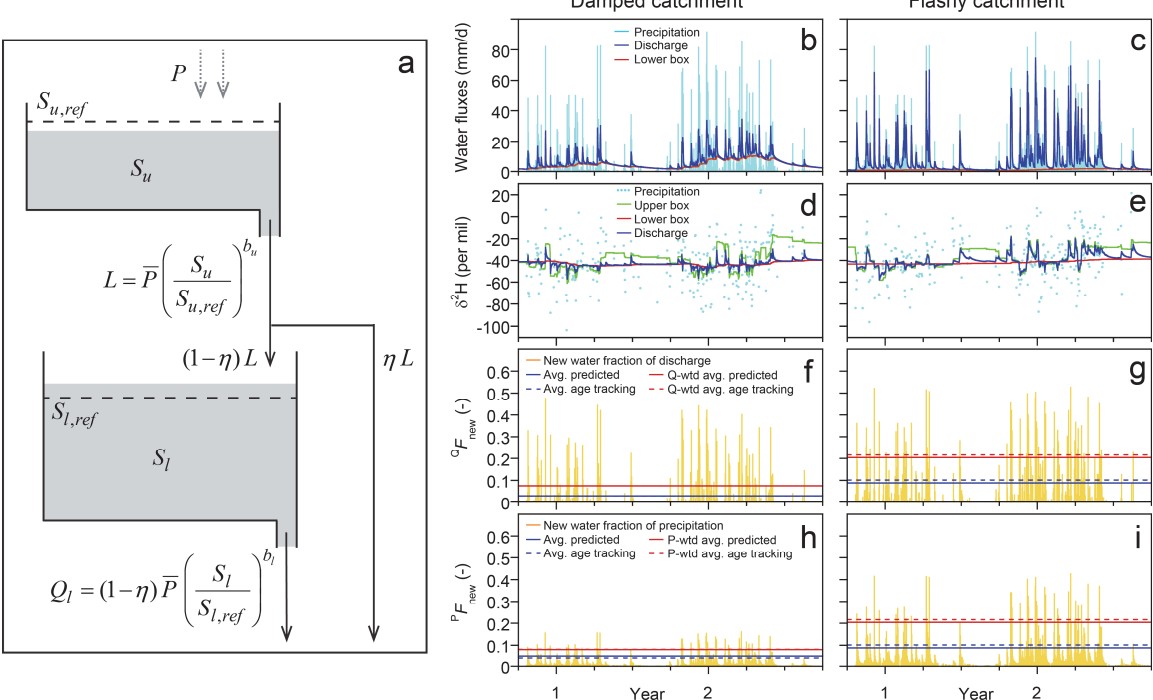

**Figure 1. Schematic diagram of the benchmark model (a), with two-year excerpts from illustrative simulations of its behavior (b-i). Model parameters for simulations of damped catchment response (left panels) are $S_{u,ref}$ = 100 mm, $S_{l,ref}$ = 1000 mm, $b_u$ = 10, and**
**$b_l$ = 3, and $\eta$ = 0.3. For simulations of flashy catchment response (right panels), all but one of the parameters are the same; only $\eta$ is changed to 0.8 and a different random realization of precipitation isotopes is used. The same daily precipitation time series (Smith River, Mediterranean climate) is used in both cases. The isotopic composition of streamflow exhibits complex dynamics over multiple time scales (blue line in panels d and e), as dominance shifts between the upper and lower boxes (green and orange lines, respectively, in panels d and e). Like the discharge and its isotopic composition, the fraction of discharge comprised of same-**
**day precipitation (the new water fraction of discharge, $^QF_{new}$, panels f and g) exhibits complex nonstationary dynamics. Nonetheless, its long-term average (dashed blue line) is well predicted by ensemble hydrograph separation (solid blue line); the same is true of the discharge-weighted average (dashed/solid red lines). The fraction of precipitation appearing in same-day discharge (the forward new water fraction, $^PF_{new}$, panels h and i) is somewhat less variable, but both its average and precipitation-weighted average are also well predicted by ensemble hydrograph separation (solid/dashed blue and red lines). In several cases**
**the dashed and solid lines cannot be distinguished because they overlap.**

### 3.2 Benchmark model behavior

Figure 1b-i shows two years of simulated daily behavior driven by the Smith River daily precipitation record applied to the

damped and flashy catchment parameter sets. The simulated stream discharge responds promptly to rainfall inputs, and

unsurprisingly the discharge response is larger in the flashy catchment (Fig. 1b-c). The streamflow isotopic response is

strongly damped in both catchments, with isotope ratios between events returning to a relatively stable baseline value





composed mostly of discharge from the lower box (Fig. 1d-e). Like the stream discharge and the isotope tracer time series, the instantaneous new water fractions (determined by age tracking within the model) also exhibit complex nonstationary dynamics (Fig. 1f-i). Despite the complexity of the modeled time-series behavior, ensemble hydrograph separation (Eqs. 14, 17, 20, and 27) accurately predicts the averages of these new water fractions, both unweighted and time-weighted, as can be

seen by comparing the dashed and solid lines (which sometimes overlap) in Fig. 1f-i.

It should be emphasized that the ensemble hydrograph separation and the benchmark model are completely independent of one another. The ensemble hydrograph separation does not know (or assume) anything about the internal workings of the benchmark model; it knows only the input and output water fluxes and their isotope signatures. This is crucial for it to work

in the real world, where any particular assumptions about the processes driving runoff could potentially be violated. Likewise, the benchmark model is not designed to conform to the assumptions underlying the ensemble hydrograph separation method. It would be relatively trivial to model a tracer time series assuming that "new water" constituted a fixed fraction of discharge, and then demonstrate that this fraction can be retrieved from the tracer behavior. What Fig. 1 demonstrates is much less obvious, and more important: that even when the new water fraction is highly dynamic and

nonstationary, an appropriate analysis of tracer behavior can accurately estimate its mean.

### 3.3 Benchmark tests: random parameter sets

This result holds not just for the two parameter sets shown in Fig. 1, but throughout the parameter ranges that are tested in the benchmark model. The scatterplots shown in Fig. 2 show new water fractions estimated by ensemble hydrograph separation, compared to the true average new water fractions determined by age tracking in the benchmark model, for 1000

random parameter sets spanning the parameter ranges described in Sect. 3.1. Figure 2 shows that ensemble hydrograph separation yields reasonably accurate estimates of average event new water fractions (Fig. 2a, b), new water fractions of discharge (Fig. 2c) and precipitation (Fig. 2d), and volume-weighted new water fractions (Fig. 2e, f). Estimates derived from single years of data (Fig. 2b) understandably exhibit greater scatter than those derived from five years of data (Fig. 2a), but in all of the plots shown in Fig. 2 there is no evidence of significant bias (the data clouds cluster around the 1:1 lines).

The scatter of the points around the 1:1 line generally agrees with the standard errors estimated from Eqs. 11, 15 and 19, suggesting that these uncertainty estimates are also reliable.



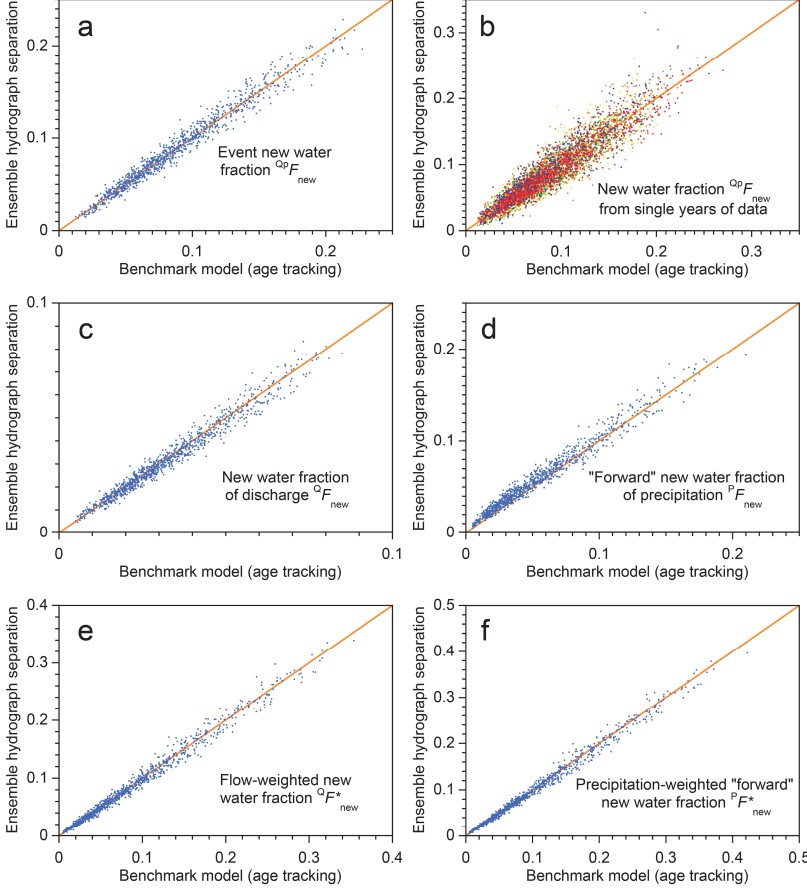

**Figure 2.** New water fractions predicted from tracer dynamics using ensemble hydrograph separation, compared to averages of time-varying new water fractions determined from age tracking in the benchmark model. Diagonal lines show perfect agreement. Each scatterplot shows 1000 points, each of which represents an individual catchment, with its own individual random set of

model parameters (i.e., catchment characteristics), randomly generated precipitation tracer time series, and random set of measurement errors and missing values (see Sect. 3.1). The daily precipitation amounts are the same (Smith River time series; Mediterranean climate) in each case. The "event new water fraction" (panels a and b) is the average fraction of new (same-day) water in streamflow during time steps with precipitation, as described in Sect. 2.3. Panel (a) shows event new water fractions estimated from five years of simulated tracer data; panel (b) shows the same quantity estimated from single years (each year is

denoted by a different color). Averaging over the five years reduces both the range and the scatter, compared to the single-year estimates. The new water fraction of discharge (panel c) is the fraction of same-day precipitation in streamflow, averaged over all time steps including rainless periods (Eq. 14, Sect. 2.4); its flow-weighted counterpart (panel e) is calculated using Eqs. (16)-(17) of Sect. 2.5. The "forward" new water fraction (the fraction of precipitation that becomes same-day streamflow; panel d) is calculated using Eq. (20), and its precipitation-weighted counterpart (panel f) is calculated using Eq. (28). In all cases there is little

evidence of bias, and the scatter around the 1:1 line is relatively small.





Mean transit times have often been estimated in the catchment hydrology literature, often under the assumption that they should also describe catchment transport and mixing on other time scales as well. This naturally leads to the question, in the context of the present study, of whether there is a systematic relationship between mean transit times and new water fractions, such that they could potentially be predicted from one another. The benchmark model allows a direct test of this

conjecture, because it tracks mean water ages as well as new water fractions. Figure 3a shows that, across the 1000 random parameter sets from Fig. 2, the relationship between new water fractions and mean transit times is a nearly perfect shotgun blast: mean transit times vary from about 40 to 400 days and new water fractions vary from nearly zero to nearly 0.1, with almost no correlation between them. Both of these quantities are estimated from age tracking in the benchmark model, so their lack of any systematic relationship does not arise from difficulties in estimating either of them from tracer data. It

instead arises because the upper tails of transit time distributions (reflecting the amounts of streamflow with very old ages) exert strong influence on mean transit times, but have no effect on new water fractions (reflecting same-day streamflow).

I have recently proposed the "young water fraction", the fraction of streamflow younger than about 2.3 months, as a more robust metric of water age than the mean transit time (Kirchner, 2016a). Fig. 3b shows that, like the mean transit time, the

young water fraction is also a poor predictor of the new water fraction, beyond the obvious constraint that new water ($\leq 1$ day old) must be a small fraction of "young" water ($\leq 69$ days old). The new water fraction will only be correlated with the young water fraction or mean transit time if the shape of the underlying transit time distribution is held constant, which is not the case for the 1000 random parameter sets considered here, and is unlikely to be true in real-world catchments either.

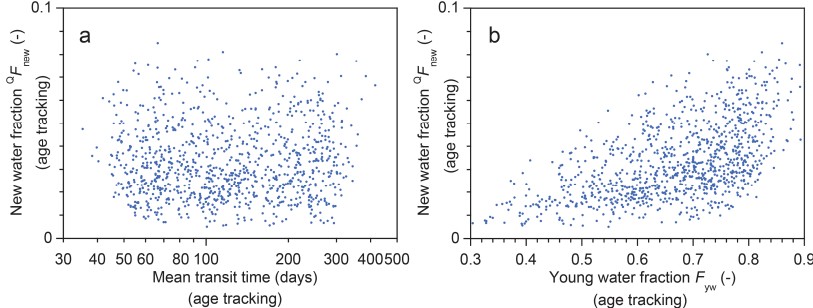

**Figure 3. Average new water fractions (same-day precipitation in streamflow) for the 1000 simulated catchments (i.e., 1000 model parameter sets) shown in Fig. 2, compared to the catchment mean transit time and the young water fraction $F_{yw}$ (the fraction of streamflow younger than 2.3 months). All values plotted here are determined from age tracking within the benchmark model, and thus are true values, without any errors associated with estimating these quantities from tracer data. Neither mean transit time**

**nor the young water fraction can reliably predict the fraction of new water in streamflow.**





### 3.4 Benchmark tests: weekly tracer sampling

Many long-term water isotope time series have been sampled at weekly intervals. Can new water fractions be estimated reliably from such sparsely sampled records? To find out, I aggregated the benchmark model's daily time series to weekly intervals, volume-weighting the isotopic composition of precipitation to simulate the effects of weekly bulk precipitation

sampling, and sub-sampling streamflow isotopes every seventh day to simulate weekly grab sampling. I then performed ensemble hydrograph separation on the aggregated weekly data, using the methods presented in Sect. 2.

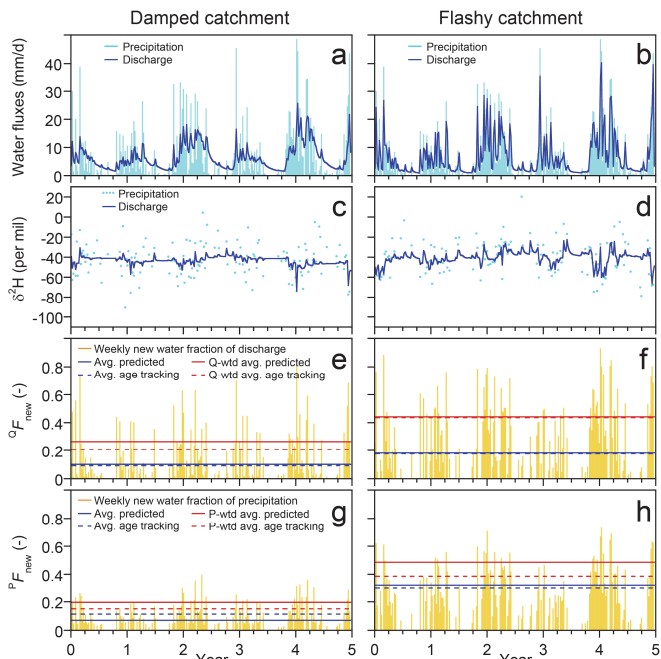

**Figure 4. Illustrative simulations of weekly water fluxes, deuterium concentrations, and new water fractions.** The benchmark
**model, precipitation forcing, and parameter values are identical to those in Fig. 1. Although the isotope tracer concentrations and
new water fractions exhibit complex nonstationary dynamics, ensemble hydrograph separation yields reasonable estimates of the
average "backward" and "forward" weekly new water fractions, as shown in panels (e-f) and (g-h), respectively. Panels (a-b)
show weekly average rates of precipitation and discharge. Panels (c-d) show the weekly volume-weighted isotopic composition of
precipitation (mimicking what would be collected in a weekly rain sample), and the instantaneous composition of discharge at the
end of each week (mimicking what would be collected in a weekly grab sample). Panels (e-f) show the fraction of discharge that is
composed of same-week precipitation (the weekly new water fraction; yellow lines), as determined from model age tracking, and
its long-term average (dashed blue line), compared to the new water fraction predicted by ensemble hydrograph separation (solid
blue line) from the weekly samples shown in panel (b). Panels (g-h) show the fraction of precipitation that becomes same-week
discharge (the weekly new water fraction of precipitation, or "forward" new water fraction, yellow lines) as determined from
model age tracking, and its long-term average (dashed blue line), compared to the new water fraction predicted by ensemble
hydrograph separation (solid blue line). Discharge-weighted and precipitation-weighted average new water fractions, and their
predicted values, are shown by red solid and dashed lines.**





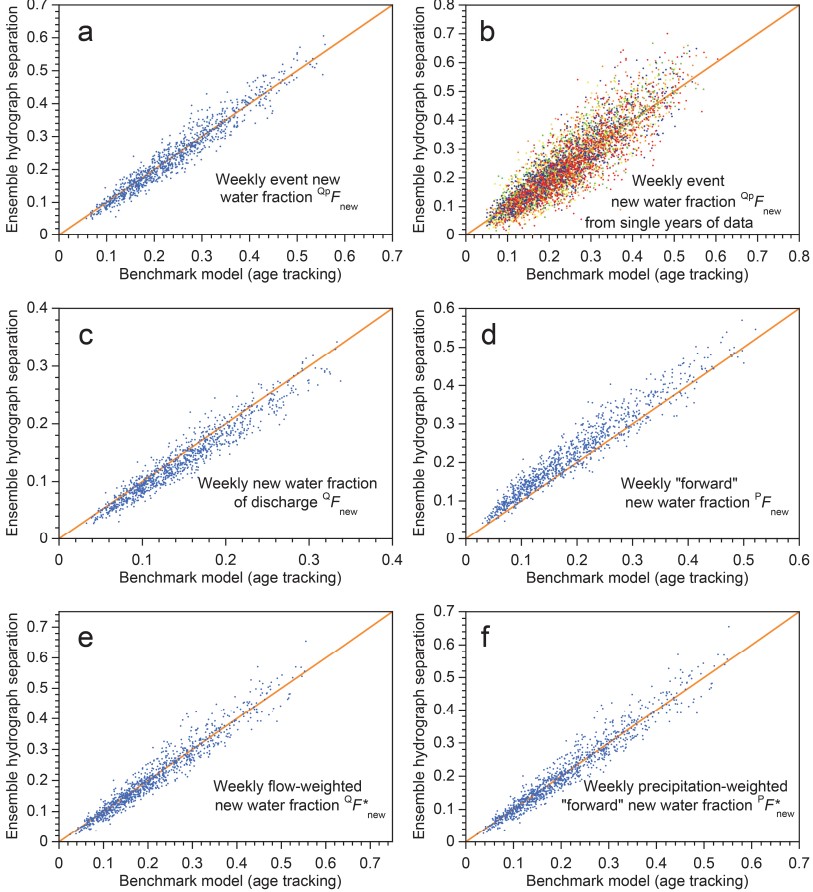

**Figure 5. New water fractions estimated from weekly tracer dynamics using ensemble hydrograph separation, compared to averages of time-varying new water fractions determined from age tracking in the benchmark model.** Plots are similar to those in Fig. 2, except here they are derived from simulated weekly sampling of tracer concentrations in precipitation and streamflow. Diagonal lines show perfect agreement. Each scatterplot shows 1000 points, each representing an individual random set of parameters, a randomly generated precipitation tracer time series, and a random set of measurement errors and missing values (see Sect. 3.1). The daily precipitation amounts are the same (Smith River time series) in each case. The "event new water fraction" (panels a and b) is the average fraction of new (same-day) water in streamflow during time steps with precipitation, as described in Sect. 2.3. Panel (a) shows event new water fractions estimated from five years of simulated weekly tracer data; panel (b) shows the same quantity estimated from single years of simulated weekly tracer data (each year is denoted by a different color). Averaging over the five years reduces scatter compared to the individual-year estimates. The new water fraction of discharge (panel c) is the fraction of same-day precipitation in streamflow, averaged over all time steps including rainless periods (Eq. 14, Sect. 2.4); its flow-weighted counterpart (panel e) is calculated using Eqs. (16)-(17) of Sect. 2.5. The "forward" new water fraction (the fraction of precipitation that becomes same-day streamflow; panel d) is calculated using Eq. (20), and its precipitation-weighted counterpart (panel f) is calculated using Eq. (27). There is only slight visual evidence of bias, and the scatter around the 1:1 line is small compared to the range spanned by the new water fractions.



Figure 4 shows the behavior of the benchmark model at weekly resolution for both the damped and flashy catchments. At
the weekly time scale, the benchmark model exhibits complex nonstationary dynamics in discharge (panels a-b), water
isotopes (panels c-d), and new water fractions (panels e-h). Nonetheless – and even though the weekly sampling timescale is
much longer than the timescales of hydrologic response in the system – ensemble hydrograph separation yields reasonable
estimates for the mean new water fractions of both precipitation and discharge (both unweighted and flow-weighted), as one
can see by comparing the dashed and solid lines in Fig. 4e-h.

A comparison of Figs. 1 and 4 shows that the isotopic signature of precipitation is less variable among the weekly samples
than among the daily samples, reflecting the fact that the weekly bulk samples of precipitation will inherently average over
the sub-weekly variability in daily rainfall. By contrast, the weekly grab samples of streamflow lose all information about
what is happening on shorter time scales. The new water fractions calculated from the weekly data are distinctly higher than
those calculated from the daily data, owing to the fact that the definition of "new" water depends on the sampling frequency:
the proportion of water ≤7 days old ("new" under weekly sampling) can never be less than the proportion ≤1 day old ("new"
under daily sampling).

Figure 5 shows scatterplots comparing new water fractions estimated by ensemble hydrograph separation and those
determined by age tracking in the benchmark model, analogous to Fig. 2 but for weekly instead of daily sampling. The
weekly new water fractions are larger than the daily ones, for the reasons described above, and exhibit more scatter because
they are based on fewer data points than their daily counterparts are. A small overestimation bias is visually evident in Fig.
2d, and an even smaller underestimation bias in Fig. 2c. These reservations notwithstanding, Fig. 5 shows that ensemble
hydrograph separation can reliably predict new water fractions of both discharge and precipitation, with and without volume-
weighting, based on weekly tracer samples.

**3.5 Variations in new water fractions with discharge, precipitation, and seasonality**

Ensemble hydrograph separation does not require continuous data as input, so it can be used to estimate $F_{new}$ values for
(potentially discontinuous) subsets of a time series that reflect conditions of particular interest. For example, if we split the
time series shown in Fig. 1 into several discharge ranges, we can see that at higher flows, tracer fluctuations in the stream are
more strongly correlated with tracer fluctuations in precipitation (Fig. 6a-b). Each of the regression slopes in Fig. 6a-b
defines the event new water fraction $^{Qp}F_{new}$ for the corresponding discharge range. Repeating this analysis for each 10-
percent interval of the discharge distribution (0-10th percentile, 10th-20th percentile, etc.), plus the 95th-100th percentile, yields
the profiles of $^{Qp}F_{new}$ as functions of discharge, as shown by the blue dots in Fig. 6c-h. The green squares show the
corresponding "forward" new water fractions $^{P}F_{new}$ for comparison. The light blue and light green lines show the
corresponding true new water fractions determined by age tracking in the benchmark model.




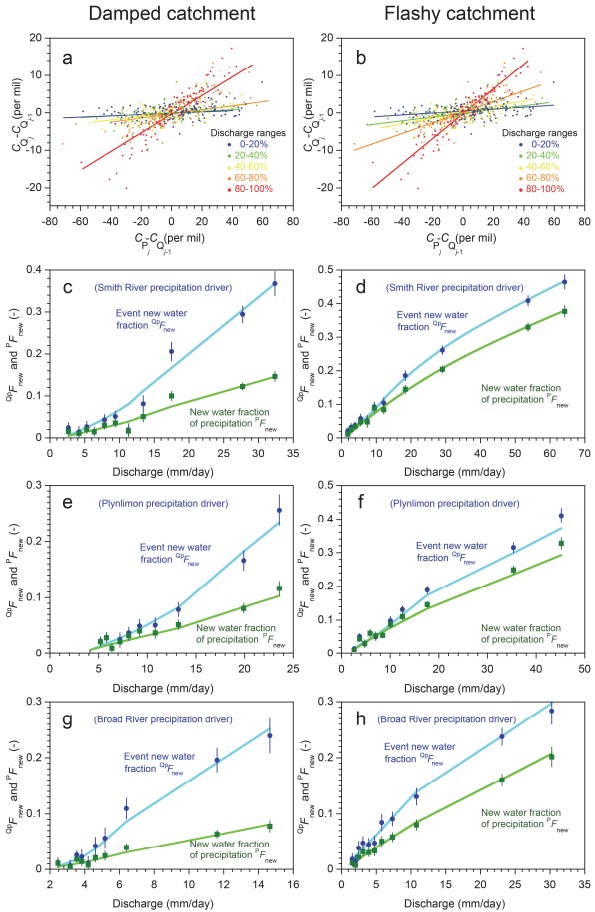

**Figure 6. Variations in new water fractions across ranges of discharge. (a,b)** Relationship between tracer concentrations in precipitation and streamflow in the benchmark model run shown in Fig. 1, stratified by percentiles of the frequency distribution of discharge, for damped and rapid response parameter sets. In these coordinates, the slopes of the regression lines through the

5  ensembles of points estimate their average event new water fractions $^{Qp}F_{new}$ (Eq. 10; Sect. 2.3). **(c-h)** Variation in new water fractions across discharge bins in the benchmark model. Dark blue and green symbols show estimates of the event new water fraction of discharge ($^{Qp}F_{new}$) and the "forward" new water fraction (fraction of precipitation appearing in same-day streamflow, $^{P}F_{new}$, Eq. 20) for each decile of the daily discharge distribution (the left-most 10 points) and the uppermost 5 percent (the right-most point). Error bars show standard errors, where these are larger than the plotting symbols. Light blue and light green lines

10  show the corresponding "true" new water fractions measured by age tracking in the benchmark model. The three rows (c-d, e-f, and g-h) show catchment response to three different precipitation climatologies (Smith River, Plynlimon, and Broad River), for both the damped response parameter set (left-hand plots c, e, and g) and the flashy response parameter set (right-hand plots d, f, and h). The new water fractions $^{Qp}F_{new}$ and $^{P}F_{new}$ vary strongly with discharge. Ensemble hydrograph separation accurately estimates both $^{Qp}F_{new}$ and $^{P}F_{new}$ across the full range of discharge for all three forcings and both parameter sets.




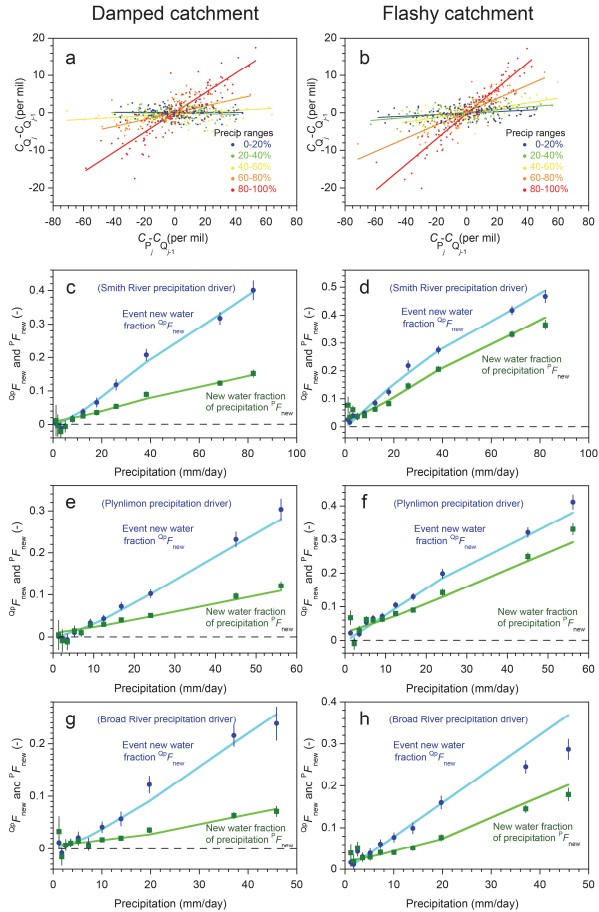

**Figure 7. Variations in new water fractions across ranges of precipitation. (a,b)** Relationship between tracer concentrations in precipitation and streamflow in the benchmark model run shown in Fig. 1, stratified by percentiles of the frequency distribution of precipitation, for damped and rapid response parameter sets. In these coordinates, the slope of the regression line through each ensemble of points estimates its average event new water fraction $^{Qp}F_{new}$ (Eq. 10; Sect. 2.3). **(c-h)** Variation in new water fractions across precipitation bins in the benchmark model. Dark blue and green symbols show estimates of the event new water fraction of discharge ($^{Qp}F_{new}$) and the "forward" new water fraction ($^{P}F_{new}$, the fraction of precipitation appearing in same-day streamflow; Eq. 20). Average $^{Qp}F_{new}$ and $^{P}F_{new}$ values are plotted for each decile of the daily precipitation distribution (the left-most 10 points) and the uppermost 5 percent (the right-most point), excluding precipitation amounts less than 1 mm/day (see text). Error bars show standard errors, where these are larger than the plotting symbols. Light blue and light green lines show the corresponding "true" new water fractions measured by age tracking in the benchmark model. The three rows (c-d, e-f, and g-h) show catchment response to three different precipitation climatologies (Smith River, Plynlimon, and Broad River), for both the damped response parameter set (left-hand plots c, e, and g) and the flashy response parameter set (right-hand plots d, f, and h). The new water fractions $^{Qp}F_{new}$ and $^{P}F_{new}$ vary strongly with daily precipitation. Ensemble hydrograph separation accurately estimates both $^{Qp}F_{new}$ and $^{P}F_{new}$ across the full range of precipitation for all three forcings and both parameter sets.



If, instead, we split the time series shown in Fig. 1 into subsets reflecting ranges of precipitation rates rather than discharge, we obtain Fig. 7. Figure 7 is a counterpart to Fig. 6, but with $^{Qp}F_{new}$ and $^{P}F_{new}$ plotted functions of rainfall rates rather than discharge. The two figures exhibit broadly similar behavior. Unsurprisingly, new water fractions are higher at higher discharges and rainfall rates, because under these conditions a higher fraction of discharge comes from the upper box, which

has younger water. "Forward" new water fractions are typically smaller than event new water fractions, because during storms the rainfall rate is higher than the streamflow rate, so the ratio between same-day streamflow and the total rainfall rate ( $^{P}F_{new}$ ) will necessarily be smaller than the ratio between same-day streamflow and the total streamflow rate ( $^{Qp}F_{new}$ ). Exceptions to this rule arise when rainfall rates are lower than discharge rates, such as during periods of light rainfall while streamflow is still undergoing recession from previous heavy rain. Thus the green and blue curves cross over one another at

the left-hand edges of Figs. 7c-h, whereas in Figs. 6c-h they do not.

Three conclusions can be drawn from Figs. 6 and 7. First, in these model catchments, new water fractions vary dramatically between low flows and high flows, and between low and high precipitation rates, with the event new water fraction $^{Qp}F_{new}$ and the forward new water fraction $^{P}F_{new}$ diverging from one another more at higher flows and higher rainfall forcing.

Second, different catchment parameters (different columns in Fig. 6) and different precipitation forcings (different rows in Fig. 6) yield different patterns in the relationships between the new water fractions $^{Qp}F_{new}$ and $^{P}F_{new}$ on the one hand, and precipitation and discharge on the other. And third, these patterns are accurately quantified by ensemble hydrograph separation, which matches the age tracking results (shown by the solid lines) within the estimated standard errors in most cases.

Thus the patterns describing how new water fractions change with precipitation and discharge may be useful as signatures of catchment transport behavior, and can be estimated directly from tracer time series using ensemble hydrograph separation. These observations raise the question of whether any of these signatures of behavior, as inferred from the patterns in these plots (if not the individual numerical values) might imply something useful about the characteristics of the catchments

themselves, ideally in a way that is not substantially confounded by precipitation climatology. A comprehensive answer is not possible within the scope of this paper, since it focuses mostly on just two parameter sets and three precipitation records. But as a first approach, one can try superimposing the results in Figs. 6 and 7 on consistent axes (note that the axes in these figures' various panels differ from one another in order to show the full range of behavior). Doing so yields Fig. 8, which overlays the age tracking results from Figs. 6c-h and 7c-h in its left- and right-hand panels, respectively. In Fig. 8,

catchments with the damped and flashy parameter sets are denoted by green and blue curves, respectively, with different levels of brightness corresponding to the three different precipitation climatologies. The key question is: are there patterns in $^{Qp}F_{new}$ or $^{P}F_{new}$ that clearly distinguish the flashy catchment from the damped catchment, regardless of the precipitation forcing? Figure 8a shows an example where this is clearly not the case; instead, the two catchments' behaviors largely





overlap in a tangle of blue and green lines. In the other three panels, however (and particularly for the trends in $^P F_{new}$ as a function of precipitation rates, as shown in Fig. 8d), the blue and green curves are relatively distinct from one another, but the different climatologies largely overlap for each catchment. This result suggests that these traces may be useful as diagnostic signatures of catchment characteristics, which are relatively insensitive to precipitation climatology. However,

Fig. 8 can only be considered a preliminary indication of what might be possible, rather than a definitive demonstration.

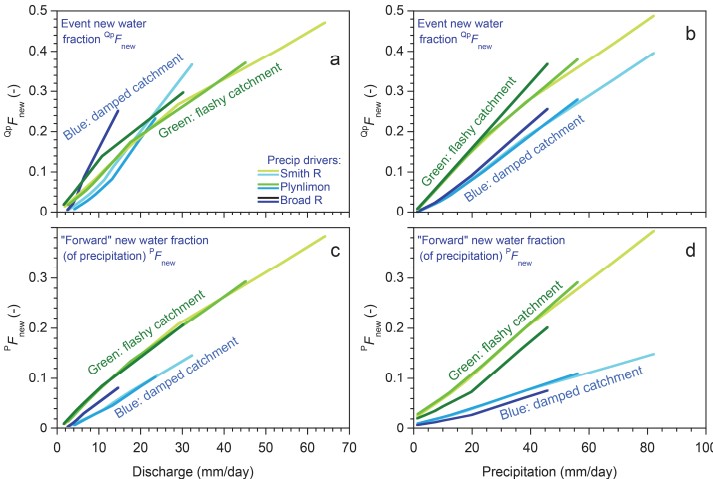

**Figure 8. Effects of precipitation climatology and catchment properties on discharge- and precipitation-dependence of new water fractions.** The lines plotted here superimpose the model age tracking results (solid lines) from Figs. 6 and 7. Left and right panels
show how event new water fractions ($^{Qp} F_{new}$, Sect. 2.3) and "forward" new water fractions ($^P F_{new}$, Sect. 2.6) vary as functions of discharge and precipitation, respectively. Green and blue lines show benchmark model behavior under the flashy and damped parameter sets, with three levels of brightness corresponding to the three different precipitation climatologies: Mediterranean climate (Smith River, lightest colors), humid maritime climate (Plynlimon, intermediate colors), and humid temperate climate (Broad River, darkest colors). When event new water fractions are plotted as functions of discharge (panel a), different
catchments and precipitation climatologies overlap. By contrast, in the other three panels (and particularly in panel d, which shows "forward" new water fractions as functions of precipitation), the lines for the flashy catchment and the damped catchment are clearly distinct from one another, regardless of precipitation climatology. This suggests that these patterns may be diagnostic of the internal workings of the catchment, but relatively insensitive to the particular rainfall forcing.

The behavior summarized in Figs. 6-8 shows that in general, new water fractions are functions of both catchment characteristics and precipitation climatology. Moreover, new water fractions will obviously depend on the sequence of precipitation events, not just on their frequency distribution, because they will depend on antecedent wetness. Thus although the ensemble hydrograph separation approach does not require continuous data, and thus can be applied to time series with data gaps, any inferred new water fractions will obviously represent only the particular time intervals that are included in the
analysis.



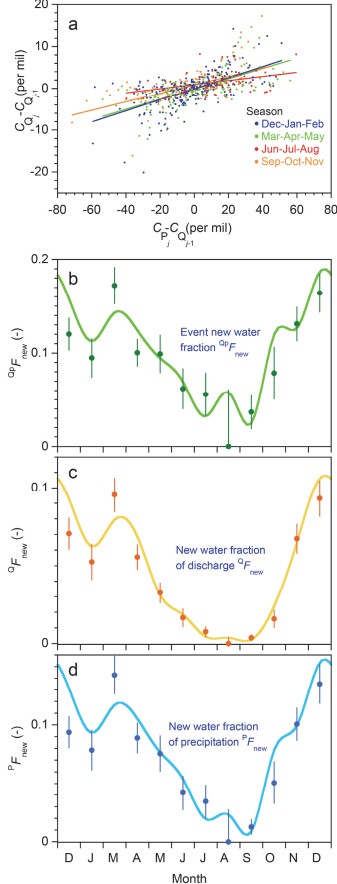

**Figure 9. Seasonality in new water fractions under Mediterranean climate precipitation forcing. (a) Relationship between tracer concentrations in precipitation and streamflow in the flashy benchmark model run shown in Fig. 1, stratified by season. Each season's event new water fraction can be estimated from the slope of the regression line fitted to the corresponding set of points.**

**(b,c,d) Average event new water fractions ($^{Qp}F_{new}$), new water fractions of discharge ($^{Q}F_{new}$), and "forward" new water fractions of precipitation ($^{P}F_{new}$) calculated from ensembles of all points within each month, across the five years of benchmark model simulations. Error bars show standard errors, where these are larger than the plotting symbols. Curves are drawn through true monthly average new water fractions, as determined by age tracking in the benchmark model. Ensemble hydrograph separation reproduces this seasonal pattern in new water fractions reasonably well. The uncertainty estimates also realistically predict the**
**average deviation of the ensemble hydrograph separation estimates from the "true" age tracking determinations. Values shown here are generated by the benchmark model with the flashy catchment parameter set and Smith River (Mediterranean climate) precipitation forcing. The new water fractions would exhibit less pronounced seasonality if the rainfall forcing were less strongly seasonal or the catchment response were less flashy.**

One implication of the forgoing considerations is that seasonal differences in storm size and frequency should also be

reflected in seasonal variations in new water fractions. Figure 9a shows a scatterplot of tracer fluctuations in streamflow and





precipitation, color-coded by season, for the "flashy" catchment simulation shown in Fig. 1. The regression lines (whose slopes define the event new water fractions $^{Qp}F_{new}$ for the corresponding seasons) show that tracer concentrations in streamflow and precipitation are more tightly coupled in winter and spring than in summer and autumn. Figures 9b–d demonstrate large variations in the event new water fraction $^{Qp}F_{new}$, the new water fraction of discharge $^{Q}F_{new}$, and the

"forward" new water fraction of precipitation $^{P}F_{new}$ from month to month, with a broad seasonal trend towards larger young water fractions in winter and spring. The month-to-month variations in the age tracking results (the smooth curves) are usually quantified by the ensemble hydrograph separation estimates (the solid dots) within their calculated uncertainties (as shown by the error bars). Thus Fig. 9 suggests that ensemble hydrograph separation can be used to quantify how catchment transport behavior is shaped by seasonal patterns in precipitation forcing.

**3.6 Effects of evaporative fractionation**

Any analysis based on water isotopes must deal with the potential effects of isotopic fractionation due to evaporation (e.g., Laudon et al., 2002; Taylor et al., 2002; Sprenger et al., 2017; Benettin et al., 2018). A detailed treatment of evaporative fractionation would necessarily be site-specific and thus beyond the scope of this paper. Nonetheless, it is possible to make a simple first estimate of how much evaporative fractionation could affect new water fractions estimated from ensemble

hydrograph separation. I first adjusted the isotope values of infiltration entering the model in Fig. 1 to mimic the effects of seasonally varying evaporative fractionation. I assumed that evaporative fractionation was a sinusoidal function of the time of year, ranging from zero in mid-winter to 20 per mil in mid-summer. Thus the assumed evaporative fractionation effectively doubled the seasonal isotopic cycle in the water entering the model catchment (but not the sampled rainfall itself, since any fractionation that occurs in both the measured precipitation and the water entering the catchment will not distort

the ensemble hydrograph separation). I then calculated new water fractions based on the time series of sampled precipitation tracer concentrations (assumed to be unaffected by evaporative fractionation) and streamflow tracer concentrations (altered by the lagged and mixed effects of evaporative fractionation), and compared these to the "true" new water fractions calculated by age tracking within the model.

The results are shown in Fig. 10, which compares 1000 Monte Carlo trials with evaporative fractionation (the blue dots) and another 1000 Monte Carlo trials without evaporative fractionation (the gray dots). One can see that in these simulations, evaporative fractionation leads to a slight tendency to underestimate new water fractions. Nonetheless, the blue and gray dots largely overlap, and both generally follow the 1:1 lines, even though the modeled fractionation effects were designed to be a worst-case scenario. Because ensemble hydrograph separation is based on patterns of fluctuations in precipitation and

streamflow tracers, any fractionation process that created a constant offset between inputs and outputs would introduce no bias. For the same reason, any fractionation process that was uncorrelated to the input isotopic signature would likewise introduce no bias; thus, for example, the modeled seasonal fractionation cycle would have had no effect if there were no



seasonal pattern in the precipitation isotopes themselves. Figure 10 thus suggests that ensemble hydrograph separation should yield realistic estimates of new water fractions, even in a worst-case scenario of confounding by evaporative fractionation.

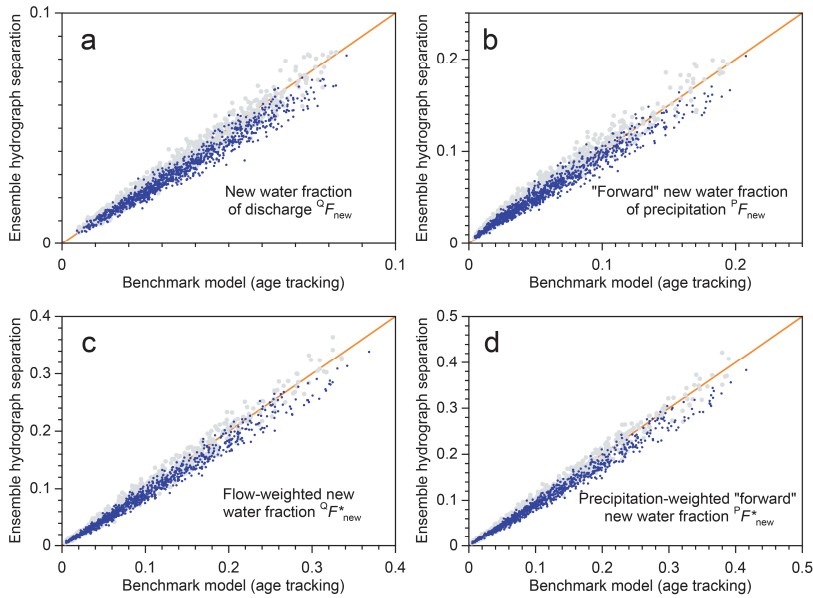

**Figure 10. Effects of seasonally varying evaporative fractionation on new water fractions estimated by ensemble hydrograph separation. Points show new water fractions predicted from tracer fluctuations in precipitation and streamflow (on the y-axis), compared to averages of time-varying new water fractions determined by age tracking in the benchmark model (on the x-axis). Blue points show 1000 model runs in which precipitation undergoes seasonally varying evaporative fractionation ranging from zero in winter to 20 per mil in summer. Gray background points show 1000 model runs without evaporative fractionation (analogous to Fig. 2). Each model run has a different random set of model parameters, measurement errors, and missing values, but the precipitation driver (Smith River daily precipitation) is the same in all cases. The blue data clouds closely follow the 1:1 line, indicating that ensemble hydrograph separation can reliably estimate new water fractions even in the presence of substantial evaporative fractionation.**

## 4 Estimating transit time distributions by ensemble hydrograph separation

A natural extension of the approach outlined in Sect. 2 would be to quantify the contributions of precipitation to streamflow over a range of lag times: to quantify, in other words, the catchment transit time distribution. In principle this should be straightforward, although in practice several challenges must be overcome. Below, I describe these issues and outline techniques for addressing them.



### 4.1 Definitions

I assume that catchment inputs and outputs are sampled at the same fixed time interval $\Delta t$, and define the time that a parcel of water enters the catchment (via rainfall, snowmelt, etc.) as $t_i$ and the time that it exits via streamflow as $t_j$. The lag interval between precipitation and streamflow is indexed as $k = j\text{-}i$. $P_i$ is the rate that precipitation or snowmelt (net of

evaporative losses) enters the catchment at time $t_i$, and $Q_j$ is the rate of discharge that exits the catchment at time $t_j$. $C_{P_i}$ and $C_{Q_j}$ are the tracer concentrations in precipitation and streamflow, respectively. The water flux that enters as precipitation at time $t_i$ and leaves as streamflow $k$ time steps later (at time $t_j = t_{i+k}$) is represented as $q_{jk}$. The sum of $q_{jk}$ over all lag times $k$ (corresponding to all previous entry times $i = j\text{-}k$) is the total discharge $Q_j$. Each of the $q_{jk}$ will be a fraction of the total precipitation falling at time $t_{i=j\text{-}k}$ and a (typically different) fraction of the total discharge at time $t_j$. The fraction of

discharge exiting at time $t_j$ that entered $k$ time steps earlier is $q_{jk}/Q_j$, and the distribution of $q_{jk}/Q_j$ over lag time $k$ is the transit time distribution conditioned on the exit time $t_j$ (also called the "backward" transit time distribution). The fraction of precipitation entering at time $t_i$ that subsequently leaves $k$ time steps later is $q_{jk}/P_{j-k} = q_{i+k,k}/P_i$, and the distribution of $q_{i+k,k}/P_i$ over lag time $k$ is the transit time distribution conditioned on the entry time $t_i$ (also called the "forward" transit time distribution). The water fluxes $P_i$, $Q_j$, and $q_{jk}$ are assumed to be in units of water depth per time (e.g., mm/day), and

thus the transit time distributions are dimensionless functions of lag time.

In practice, precipitation fluxes are typically measured as averages over discrete time intervals, and tracer concentrations in precipitation are likewise volume-averaged over discrete intervals (such as a day or a week) during which the sample is accumulated. By contrast, discharge fluxes are typically measured instantaneously, and discharge tracer concentrations are

typically measured in instantaneous grab samples. In most of what follows, I will assume that $P_i$ and $C_{P_i}$ are averages over the interval $t_{(i-1)} < t \leq t_i$, and $Q_j$ and $C_{Q_j}$ are instantaneous values at $t = t_j$. However, in a few catchment studies, discharge concentrations have instead been measured in time-integrated samples. The analysis presented below is the same, whether the discharge tracer concentrations $C_{Q_j}$ are instantaneous at $t = t_j$ or are integrated over each time interval $t_{(i-1)} < t \leq t_i$. The interpretation is slightly different, however, because the average lag time corresponding to a given lag

interval $k$ will depend on how precipitation and streamflow are sampled. Usually, streamwater samples are collected more-or-less instantaneously (grab sampling), and precipitation samples are integrated over the time interval that the sampler is open. A typical daily sampling scheme, for example, might involve collecting a precipitation sample at noon (which integrates precipitation that fell over the previous 24 hours), and also collecting a grab sample of streamflow at noon. In this case, the average lag time between a raindrop falling as precipitation and being sampled in the same day's streamflow (i.e.,

$k = 0$) would be 12 hours, assuming that on average, the probability of rainfall is independent of the time of day. Thus in this conventional sampling scheme, the average lag time will be $(k + 0.5)\Delta t$, where $\Delta t$ is the sampling interval. If, instead, the stream samples were daily composites, then (for example) the "same day" raindrops appearing the first hour's subsample





of streamflow would have an average lag time of 30 minutes, the second hour's would be 60 minutes, and so forth, and therefore the daily average lag time would be 6 hours. Thus if stream samples are time-integrated composites, the average lag time will be $(k + 0.25)\Delta t$.

I now outline the fundamentals of the ensemble hydrograph separation approach to estimating transit time distributions. Conservation of water mass requires that the discharge at time step $j$ equals the contributions from all lag times $k$ (corresponding to all previous entry times $i = j\text{-}k$):

$$Q_j = \sum_{k \geq 0} q_{jk} \qquad . \tag{29}$$

Because tracing contributions to streamflow from all previous time steps would be impractical, it will be necessary to
truncate the summation in (29) at some maximum lag, which I will denote as $m$, and to combine the unmeasured older contributions in a water flux $Q_{\text{older}_j}$:

$$Q_j = \sum_{k=0}^{m} q_{jk} + Q_{\text{older}_j} \qquad , \qquad Q_{\text{older}_j} = \sum_{k=m+1}^{\infty} q_{jk} = Q_j - \sum_{k=0}^{m} q_{jk} \qquad . \tag{30}$$

Conservation of tracer mass requires that the tracer fluxes add up similarly, again with a catch-all flux $Q_{\text{older}_j} C_{\text{older}_j}$:

$$Q_j \, C_{Q_j} = \sum_{k=0}^{m} q_{jk} \, C_{P_{j-k}} + Q_{\text{older}_j} C_{\text{older}_j} = \sum_{k=0}^{m} q_{j,k} \, C_{P_{j-k}} + \left( Q_j - \sum_{k=0}^{m} q_{jk} \right) C_{\text{older}_j} \qquad . \tag{31}$$

Dividing (31) by $Q_j$ and rearranging terms directly yields:

$$\left( C_{Q_j} - C_{\text{older}_j} \right) = \sum_{k=0}^{m} \frac{q_{jk}}{Q_j} \left( C_{P_{j-k}} - C_{\text{older}_j} \right) \qquad , \tag{32}$$

which readers will recognize as the multi-lag counterpart of Eq. (7).

Analogous to the approach in Sect. 2, here I account for the concentration of older inputs $C_{\text{older}_j}$ using the streamflow
concentration at lag $m + 1$, just beyond the longest lag $m$, with the goal of filtering out long-term patterns that could otherwise distort the correlations between $C_{P_{j-k}}$ and $C_{Q_j}$. Thus $C_{Q_{j-m-1}}$ serves as a reference level for measuring fluctuations in precipitation and streamflow tracer concentrations, analogous to $C_{Q_{j-1}}$ in Eq. (8). Adding a bias term $\alpha$ and an error term $\varepsilon_j$ yields

$$\left( C_{Q_j} - C_{Q_{j-m-1}} \right) = \sum_{k=0}^{m} \frac{q_{jk}}{Q_j} \left( C_{P_{j-k}} - C_{Q_{j-m-1}} \right) + \alpha + \varepsilon_j \qquad , \tag{33}$$





which almost looks like a conventional multiple linear regression equation,

$$y_j = \sum_{k=0}^{m} \beta_k \, x_{jk} \; + \alpha + \varepsilon_j \quad , \tag{34}$$

where

$$y_j = \left( C_{Q_j} - C_{Q_{j-m-1}} \right) \quad \text{and} \quad x_{jk} = \left( C_{P_{j-k}} - C_{Q_{j-m-1}} \right) \quad , \tag{35}$$

with the difference that the coefficients $\beta_k$ in (34) are constant over all exit times $j$ and differ only as a function of the lag time $k$, whereas the $q_{jk}/Q_j$ terms in (33) can differ among both lag times $k$ and exit times $j$. Nonetheless, by analogy with the mathematical arguments in Appendix A and those at the end of Appendix B, one can expect that $\beta_k$ will closely approximate the average of the time-varying contributions $q_{jk}/Q_j$ to streamflow over the ensemble of exit times $j$ (please note that this is not the same as assuming that the transit time distribution is time-invariant!). Substituting $\beta_k$ as an ensemble

estimate of $q_{jk}/Q_j$, one obtains the ensemble hydrograph separation equation for estimating transit time distributions,

$$\left( C_{Q_j} - C_{Q_{j-m-1}} \right) = \sum_{k=0}^{m} \beta_k \left( C_{P_{j-k}} - C_{Q_{j-m-1}} \right) \; + \alpha + \varepsilon_j \quad . \tag{36}$$

When appropriately re-scaled as described in Sects. 4.5-4.7 below, the coefficients $\beta_k$ in Eq. (36) – or more precisely, their regression estimates $\hat{\beta}_k$ – can be used to estimate the time-averaged (also sometimes called the "marginal") transit time distribution.

**4.2 Solution method**

Using $\boldsymbol{Y}$ to represent the vector of reference-corrected streamflow tracer concentrations $y_j = C_{Q_j} - C_{Q_{j-m-1}}$ and $\mathbf{X}$ to represent the matrix of reference-corrected input tracer concentrations $x_{jk} = C_{P_{j-k}} - C_{-}Q_{j-m-1}$, we can rewrite (36) in the array form of a multiple regression equation:

$$\boldsymbol{Y} = \sum_{k=0}^{m} \beta_k \, \boldsymbol{X}_k \; + \alpha + \boldsymbol{\varepsilon} \quad , \tag{37}$$

where $\boldsymbol{X}_k$ is the $k^{\text{th}}$ column vector of $\mathbf{X}$, and $\boldsymbol{\varepsilon}$ is the vector of the errors $\varepsilon_j$. The least-squares solution for multiple regressions like (37) can be expressed in matrix form as





$$
\begin{pmatrix} \hat{\beta}_0 \\ \hat{\beta}_1 \\ \hat{\beta}_2 \\ \vdots \\ \hat{\beta}_m \end{pmatrix} = \begin{pmatrix} \mathrm{cov}(X_0,X_0) & \mathrm{cov}(X_0,X_1) & \mathrm{cov}(X_0,X_2) & \cdots & \mathrm{cov}(X_0,X_m) \\ \mathrm{cov}(X_1,X_0) & \mathrm{cov}(X_1,X_1) & \mathrm{cov}(X_1,X_2) & \cdots & \mathrm{cov}(X_1,X_m) \\ \mathrm{cov}(X_2,X_0) & \mathrm{cov}(X_2,X_1) & \mathrm{cov}(X_2,X_2) & \cdots & \mathrm{cov}(X_2,X_m) \\ \vdots & \vdots & \vdots & \ddots & \vdots \\ \mathrm{cov}(X_m,X_0) & \mathrm{cov}(X_m,X_1) & \mathrm{cov}(X_m,X_2) & \cdots & \mathrm{cov}(X_m,X_m) \end{pmatrix}^{-1} \begin{pmatrix} \mathrm{cov}(X_0,Y) \\ \mathrm{cov}(X_1,Y) \\ \mathrm{cov}(X_2,Y) \\ \vdots \\ \mathrm{cov}(X_m,Y) \end{pmatrix} , \qquad (38)
$$

where the regression coefficients $\hat{\beta}_k$ are the least-squares estimators of the true (but unknowable) coefficients $\beta_k$. Equation (38) is the multi-dimensional counterpart to Eq. (10). The first term on the right-hand side of (38) is the inverse of the matrix of the covariances of the $X_k$ at each lag with each other lag, and the second term is a vector of the covariances

between $Y$ and the $X_k$ at each lag. Equation (38) is equivalent to the more widely known "normal equation" for solving multiple regressions,

$$
\widehat{\boldsymbol{\beta}} = (\mathbf{X}^{\mathrm{T}}\mathbf{X})^{-1}\,\mathbf{X}^{\mathrm{T}}Y \quad , \qquad (39)
$$

if one first normalizes $Y$ and each of the $X_k$ by subtracting their respective means; doing so has no effect on the estimates of the regression coefficients $\hat{\beta}_k$. (The elements of the square matrix $\mathbf{X}^{\mathrm{T}}\mathbf{X}$ are the covariances between the $X_k$'s at each pair of

lags, multiplied by the number of samples; likewise the elements of the column matrix $\mathbf{X}^{\mathrm{T}}Y$ are the covariances between each of the $X_k$'s and $Y$, multiplied by the number of samples.)

Astute readers will immediately notice a fundamental problem with applying Eqs. (38) or (39) in practice, namely that they require precipitation tracer concentrations $C_{P_{j-k}}$ for all time steps $j$ and lags $k$. In every practical case, many precipitation

tracer concentrations will be missing, for two reasons. Some tracer concentrations will be missing due to sampling or measurement failures, and many more will be inherently missing because precipitation tracer concentrations cannot exist for time steps without precipitation. As we will see shortly, missing measurements that arise for these two different reasons must be handled in two different ways. But regardless of its origins, each missing tracer concentration $C_{P_i}$ at time step $i$ will create a diagonal line of missing values in the matrix $x_{jk}$, causing a missing value in the first column ($k = 0$) at $j = i$, and

another in the second column ($k = 1$) at $j = i + 1$, and so on up to the last column ($k = m$) at $j = i + m$.

So-called "missing data problems" arise frequently in the statistical literature, and several approaches have been proposed for handling them (Little, 1992). One approach, termed "listwise deletion" or "complete-case analysis", involves discarding all cases (meaning all rows $j$ in the matrix $x_{jk}$) in which any variables are missing, and analyzing only the remaining (complete)

cases. In our situation, this would mean analyzing only exit times $t_j$ that are preceded by unbroken series of rainy periods, up to the maximum lag $m$ for which we want to estimate the coefficients $\hat{\beta}_k$. Such ensembles of points would be mathematically convenient, but they would also be very strongly biased in a hydrological sense, because they would represent periods of unusually consistent rainfall (and thus unusually wet catchment conditions). Furthermore, if the





maximum lag $m$ is sufficiently long, records with continuous rainfall over all $m + 1$ lags ($k = 0 \dots m$) will become impossible to find. For these reasons, complete case analysis is not a feasible approach to our problem.

A second class of approaches to the missing data problem involves imputing values to the missing data (Little, 1992). In our
case, however, many of the missing data are not simply unmeasured, but cannot exist at all (because rainless days have no rainfall concentrations), so it is not obvious how to impute the missing values.

A third approach, termed "pairwise deletion" or "available-case analysis", first proposed by Glasser (1964), entails evaluating each of the covariances in (38) using any cases for which the necessary pairs of observations exist. Thus the
covariances in (38) are replaced by

$$\text{cov}(\boldsymbol{X}_k, \boldsymbol{X}_\ell)_{(k\ell)} = \frac{1}{n_{(k\ell)} - 1} \sum_{j \in (k\ell)} \left( x_{jk} - \bar{x}_{k(k\ell)} \right) \left( x_{j\ell} - \bar{x}_{\ell(k\ell)} \right) \tag{40}$$

and

$$\text{cov}(\boldsymbol{X}_k, \boldsymbol{Y})_{(ky)} = \frac{1}{n_{(ky)} - 1} \sum_{j \in (ky)} \left( x_{jk} - \bar{x}_{k(ky)} \right) \left( y_j - \bar{y}_{(ky)} \right) \quad , \tag{41}$$

where the notation $(k\ell)$ indicates terms that are evaluated over all cases $j$ for which both $x_{jk}$ and $x_{j\ell}$ exist (e.g., $\bar{x}_{k(k\ell)}$ is the mean of the column vector $\boldsymbol{X}_k$ for rows $j$ where neither $x_{jk}$ nor $x_{j\ell}$ is missing, and $n_{(k\ell)}$ is the number of such cases), and $(ky)$ indicates terms that are evaluated over all cases $j$ for which $x_{jk}$ and $y_j$ exist.

Glasser's approach can potentially handle the problem of tracer measurements that are missing at random due to sampling or
analysis failures. However, it will not correctly handle the problem of tracer concentrations that are missing due to a lack of sufficient precipitation, because it assumes that the missing values occur randomly and therefore that Eqs. (40)-(41) are unbiased estimators of the covariances that one would obtain if no samples were missing. But when little or no precipitation falls on the catchment, it delivers little or no tracer to subsequent streamflow, and thus its contribution to the covariance between precipitation and streamflow concentrations will be nearly zero. Therefore different handling is required for
precipitation tracer concentrations that are missing because they were not measured, versus those that are missing because they never existed at all (because no rain fell). As shown in Appendix B, periods without precipitation must be taken into account with weighting factors on the off-diagonal elements of the covariance matrix (because the tracer covariances will be less strongly coupled to one another, the less frequently precipitation falls). When the approach outlined in Appendix B is combined with Glasser's method for estimating each of the covariances, the end result is




$$
\begin{pmatrix} \hat{\beta}_0 \\ \hat{\beta}_1 \\ \hat{\beta}_2 \\ \vdots \\ \hat{\beta}_m \end{pmatrix} = \begin{pmatrix} \text{cov}(X_0,X_0)_{(0,0)} & \frac{n_{x_0 x_1}}{n_{x_0}}\text{cov}(X_0,X_1)_{(0,1)} & \frac{n_{x_0 x_2}}{n_{x_0}}\text{cov}(X_0,X_2)_{(0,2)} & \cdots & \frac{n_{x_0 x_m}}{n_{x_0}}\text{cov}(X_0,X_m)_{(0,m)} \\ \frac{n_{x_1 x_0}}{n_{x_1}}\text{cov}(X_1,X_0)_{(1,0)} & \text{cov}(X_1,X_1)_{(1,1)} & \frac{n_{x_1 x_2}}{n_{x_1}}\text{cov}(X_1,X_2)_{(1,2)} & \cdots & \frac{n_{x_1 x_m}}{n_{x_1}}\text{cov}(X_1,X_m)_{(1,m)} \\ \frac{n_{x_2 x_0}}{n_{x_2}}\text{cov}(X_2,X_0)_{(2,0)} & \frac{n_{x_2 x_1}}{n_{x_2}}\text{cov}(X_2,X_1)_{(2,1)} & \text{cov}(X_2,X_2)_{(2,2)} & \cdots & \frac{n_{x_2 x_m}}{n_{x_2}}\text{cov}(X_2,X_m)_{(2,m)} \\ \vdots & \vdots & \vdots & \ddots & \vdots \\ \frac{n_{x_m x_0}}{n_{x_m}}\text{cov}(X_m,X_0)_{(m,0)} & \frac{n_{x_m x_1}}{n_{x_m}}\text{cov}(X_m,X_1)_{(m,1)} & \frac{n_{x_m x_2}}{n_{x_m}}\text{cov}(X_m,X_2)_{(m,2)} & \cdots & \text{cov}(X_m,X_m)_{(m,m)} \end{pmatrix}^{-1} \begin{pmatrix} \text{cov}(X_0,Y)_{(0,y)} \\ \text{cov}(X_1,Y)_{(1,y)} \\ \text{cov}(X_2,Y)_{(2,y)} \\ \vdots \\ \text{cov}(X_m,Y)_{(m,y)} \end{pmatrix},
$$

(42)

where the covariance terms are defined by Eqs. (40)-(41), $n_{x_k}$ is the number of time steps $j$ for which precipitation fell at time $i = j - k$ (whether or not that precipitation was sampled and analyzed), and $n_{x_k x_\ell}$ is the number of time steps $j$ for which precipitation fell at both $j - k$ and $j - \ell$ (again, whether or not those precipitation events were sampled and analyzed).

As explained in Appendix B, the estimated coefficients $\hat{\beta}_k$ will closely approximate the average of the time-varying coefficients $\beta_{j,k} = q_{jk}/Q_j$, averaged over times $j$ for which precipitation fell at times $i = j - k$ (but not over rainless periods, from which no streamflow can originate and thus $\beta_{j,k} = q_{jk}/Q_j$ must be zero). In practice, a single droplet of mist does not make a rainstorm, so there will be some threshold rate of precipitation (here I have used 1 mm/day) below which there will be too little water to have any detectable effect on streamflow (and too little water to analyze). Thus $n_{x_k}$ and $n_{x_k x_\ell}$ will be determined by counting the time steps that exceed this precipitation threshold:

$$
n_{x_k} = \sum_{j=1}^{n} \begin{cases} 1 : P_{j-k} \geq P_{\text{threshold}} \\ 0 : P_{j-k} < P_{\text{threshold}} \end{cases}
$$

$$
n_{x_k x_\ell} = \sum_{j=1}^{n} \begin{cases} 1 : P_{j-k} \geq P_{\text{threshold}} \ \text{ and } \ P_{j-\ell} \geq P_{\text{threshold}} \\ 0 : P_{j-k} < P_{\text{threshold}} \ \text{ or } \ P_{j-\ell} < P_{\text{threshold}} \end{cases},
$$

(43)

Note that some measurements will usually also be missing due to sampling or measurement failures in addition to precipitation intermittency. Thus $n_{(ky)}$ and $n_{(k\ell)}$ in Eqs. (40)-(41), which account for both types of missing data, will typically be smaller than $n_{x_k}$ and $n_{x_k x_\ell}$ in Eq. (42).

## 4.3 Tikhonov-Phillips regularization

Gaps in the underlying data imply that, unlike covariance matrices in conventional multiple regressions, the covariance matrix in (38) is not guaranteed to be positive definite (and thus may not be invertible). Even when the covariance matrix is invertible, it may be ill-conditioned, making its inversion unstable. This issue arises frequently in inversion problems whenever different combinations of lagged inputs will have nearly equivalent effects on the output, making it difficult for the inversion to decide among them (this is the multi-dimensional analogue to nearly dividing by zero in Eq. 10). In minimizing the sum of squared deviations from the observations, inversions like Eq. (38) can potentially yield wildly oscillating solutions, with huge negative values of $\hat{\beta}_k$ at some lags delicately balancing huge positive values at other lags. Such results




are not just unrealistic; they are also unstable, with tiny differences in the underlying data potentially having huge effects on the $\hat{\beta}_k$ estimates.

A standard therapy for this disease is Tikhonov-Phillips regularization (Phillips, 1962; Tikhonov, 1963). This technique

(also known by many other names, including Tikhonov regularization, Tikhonov-Miller regularization, and the Phillips-Twomey method) is commonly used to solve ill-conditioned geophysical inversion problems (Zhadanov, 2015) but is less widely known in hydrology. Whereas conventional least-squares inversion finds the set of parameters $\hat{\beta}_k$ that will minimize the misfit between the predicted and observed $y_j$, no matter how strange those $\hat{\beta}_k$ values may be, Tikhonov-Phillips regularization adds a second criterion that quantifies the strangeness of the $\hat{\beta}_k$ values themselves, and finds the set of

parameters $\hat{\beta}_k$ that will minimize the sum of both criteria. Phillips (1962) first showed how this joint minimization could be formulated as a simple extension of the normal matrix approach to solving linear inversion problems. This formulation, applied to our problem, is:

$$\begin{pmatrix} \hat{\beta}_k \end{pmatrix} = \begin{pmatrix} \mathbf{C} + \lambda\mathbf{H} \end{pmatrix}^{-1} \begin{pmatrix} \mathrm{cov}(\boldsymbol{X}_k, \boldsymbol{Y})_{(ky)} \end{pmatrix} \quad , \tag{44}$$

where $\mathbf{C}$ is the matrix of covariance terms in Eq. (42), and the parameter $\lambda$ controls the relative weight given to the two

criteria, namely the mean squared deviations of the predicted and observed $y_j$ values (controlled by the covariance matrix $\mathbf{C}$) and the deviations from ideal behavior of the $\hat{\beta}_k$ values (controlled by the matrix $\mathbf{H}$).

The form of $\mathbf{H}$ is determined by the criterion of reasonableness that is applied to the $\hat{\beta}_k$. One possible criterion (among many that can be found in the literature) can be called *parsimony*: minimize the mean square of the $\hat{\beta}_k$, thus penalizing solutions

with large $\hat{\beta}_k$ values. Minimizing the functional $\langle \hat{\beta}_k^2 \rangle$ yields the identity matrix for $\mathbf{H}$ (Tikhonov, 1963):

$$\mathbf{H} = \begin{pmatrix} 1 & 0 & 0 & \cdots & 0 \\ 0 & 1 & 0 & \cdots & 0 \\ \vdots & & \ddots & & \vdots \\ 0 & \cdots & 0 & 1 & 0 \\ 0 & \cdots & 0 & 0 & 1 \end{pmatrix} \quad . \tag{45}$$

This approach, also called "ridge regression" because it adds a "ridge" of extra weight along the diagonal of the covariance matrix, was Tikhonov's original regularization criterion and is widely used in geophysical inversions (including unit hydrograph estimation). In our case, however, it would have the undesirable effect of creating a systematic underestimation

bias in our estimates of recent contributions to streamflow, by always making the $\hat{\beta}_k$ smaller than they would be otherwise.



A second possible criterion is *consistency*: minimize the variance of the $\hat{\beta}_k$, thus penalizing solutions with individual $\hat{\beta}_k$ values that differ greatly from the mean of all the $\hat{\beta}_k$. Minimizing the functional $\langle(\hat{\beta}_k - \langle\hat{\beta}_k\rangle)^2\rangle$, where angled brackets indicate averages from $k = 1$ to $k = m$, leads to an **H** matrix of the form (Press et al., 1992),

$$\mathbf{H} = \begin{pmatrix} 1 & -1 & 0 & 0 & 0 & \cdots & 0 \\ -1 & 2 & -1 & 0 & 0 & \cdots & 0 \\ 0 & -1 & 2 & -1 & 0 & \cdots & 0 \\ \vdots & & & \ddots & & & \vdots \\ 0 & \cdots & 0 & -1 & 2 & -1 & 0 \\ 0 & \cdots & 0 & 0 & -1 & 2 & -1 \\ 0 & \cdots & 0 & 0 & 0 & -1 & 1 \end{pmatrix} \quad . \tag{46}$$

5   Like (45), this minimum-variance criterion is also widely used, and has the advantage that, unlike (45), it does not lead to systematic biases in the average $\hat{\beta}_k$ values. However, if the transit time distribution is strongly skewed, with large contributions to streamflow at short lags, minimizing the variance of the $\hat{\beta}_k$ will tend to suppress this short-lag peak in the transit time distribution. This distortion of the transit time distribution is undesirable when one seeks to quantify recent contributions to streamflow.

A third possible criterion is *smoothness*: minimize the mean square of the second derivatives of the $\hat{\beta}_k$, thus penalizing $\hat{\beta}_k$ values that deviate greatly from their neighbors. Minimizing the second derivative functional $\langle(\hat{\beta}_{k-1} - 2\hat{\beta}_k + \hat{\beta}_{k+1})^2\rangle$, where the angled brackets indicate an average from $k = 1$ to $k = m$-1, leads to an **H** matrix of the form (Phillips, 1962; Press et al., 1992),

$$\mathbf{H} = \begin{pmatrix} 1 & -2 & 1 & 0 & 0 & 0 & 0 & \cdots & 0 \\ -2 & 5 & -4 & 1 & 0 & 0 & 0 & \cdots & 0 \\ 1 & -4 & 6 & -4 & 1 & 0 & 0 & \cdots & 0 \\ 0 & 1 & -4 & 6 & -4 & 1 & 0 & \cdots & 0 \\ \vdots & & & & \ddots & & & & \vdots \\ 0 & \cdots & 0 & 1 & -4 & 6 & -4 & 1 & 0 \\ 0 & \cdots & 0 & 0 & 1 & -4 & 6 & -4 & 1 \\ 0 & \cdots & 0 & 0 & 0 & 1 & -4 & 5 & -2 \\ 0 & \cdots & 0 & 0 & 0 & 0 & 1 & -2 & 1 \end{pmatrix} \quad . \tag{47}$$

This criterion, first used by Phillips (1962) has the advantage of strongly suppressing rapid oscillations in the $\hat{\beta}_k$ while barely affecting the larger-scale structure of the inferred transit time distribution. Therefore this will be the regularization criterion employed here.

20   The solution to Eq. (44) will obviously depend on the value of the parameter $\lambda$, which determines the relative weight given to the regularization criterion versus the goodness-of-fit criterion. How should the value of $\lambda$ be chosen? One can first note that for $\lambda\mathbf{H}$ to be dimensionally consistent with the covariance matrix, $\lambda$ must have the same dimensions as the variance of $\mathbf{X}_k$. The second point to note is that the regularization criterion and the goodness-of-fit criterion will have roughly equal weight in determining the $\hat{\beta}_k$ if the trace of $\lambda\mathbf{H}$ equals the trace of the covariance matrix **C** (Press et al., 1992). Combining





these two considerations, we can define a dimensionless parameter $v$ that ranges between 0 and 1 and expresses the fractional weight given to the regularization criterion, and then calculate the corresponding value of $\lambda$ as,

$$\lambda = \frac{v}{1-v} \frac{\mathrm{Tr}(\mathbf{C})}{\mathrm{Tr}(\mathbf{H})} \quad . \tag{48}$$

As one can see from Eq. (48), when $v = 0.5$, the trace of $\lambda\mathbf{H}$ will equal the trace of the covariance matrix $\mathbf{C}$, and the two

criteria will have roughly equal weight in determining the $\hat{\beta}_k$. As $v$ grows toward 1, the solution will be increasingly dominated by the regularization criterion; conversely, if $v = 0$ the regularization criterion will be ignored, and Eq. (44) will become equivalent to Eq. (38).

The question remains as to what the most appropriate value of $v$ (or $\lambda$) would be for any particular situation. An appropriate

degree of regularization will prevent the predicted values of $y_j$ from fitting the data more closely than they should (that is, it will prevent "fitting the noise" with unrealistic values of $\hat{\beta}_k$). Thus a theoretically optimal value of $v$ or $\lambda$ would be one that makes the variance of the prediction errors of the $y_j$ similar to the expected variance of the $\varepsilon_j$ (Press et al., 1992). This approach will not work for our problem, for three reasons. First, the variance of the $\varepsilon_j$ is not known a priori. Second, directly calculating the predicted $y_j$, and thus the prediction errors, is impossible if many values of $x_{jk}$ are missing, as will

usually be the case. Third, and perhaps most importantly, equation (37) is, strictly speaking, structurally incorrect for our system, because $\hat{\beta}_k$ is only an approximation to the time-varying $q_{jk}/Q_j$. Therefore in our case a more pragmatic approach (which is also taken in many geophysical applications of regularization methods) is to follow the advice of Phillips (1962) that "in practice several values... should be tried and the best value should be the one that appears to take out the oscillation without appreciably smoothing the [solution]", while keeping in mind that an element of subjectivity is inevitably

introduced. In the analyses presented here, unless otherwise noted, $v = 0.5$ and thus the regularization criterion and the least-squares criterion have roughly equal weight in determining the values of the $\hat{\beta}_k$. Regularization usually has little effect on the estimated transit time distributions presented below, but it can serve as a safeguard against obtaining wildly unrealistic results, particularly with large fractions of missing measurements.

### 4.4 Uncertainties

In conventional multiple regression analysis, calculating the uncertainties in the $\hat{\beta}_k$ requires estimating the variance $s_\varepsilon^2$ of the prediction errors $\varepsilon_j$,

$$s_\varepsilon^2 = \frac{n-1}{n-(m+1)-1} \mathrm{var}(\varepsilon_j)$$

$$\varepsilon_j = y_j - \sum_{k=0}^m \hat{\beta}_k x_{jk} - \alpha = \left(C_{Q_j} - C_{Q_{j-m-1}}\right) - \sum_{k=0}^m \hat{\beta}_k \left(C_{P_{j-k}} - C_{Q_{j-m-1}}\right) - \alpha \quad . \tag{49}$$





It may seem that calculating (49) is impossible in our case, because values of $C_{\mathrm{P}_{j-k}}$ are missing for all days $i = j\text{-}k$ without rain. However, as noted in Sect. 4.2 above, for those points the true value of $\beta_{j,k}$ is known to be zero, so the rainless terms can simply be ignored because they will have no effect on the predicted $y_j$. Thus if sampling and measurement failures account for only a small fraction of the missing tracer concentrations, Eq. (49) may yield adequate estimates of $s_\varepsilon^2$. Where

there are many sampling and measurement failures, we can use the error variance formula of Glasser (1964), adapted to our problem as

$$s_\varepsilon^2 = \frac{n-1}{n-(m+1)-1}\left( s_y^2 - \sum_{k=0}^{m} \hat{\beta}_k \left(\frac{n_{x_k}}{n}\right) \mathrm{cov}(X_k, Y)_{(ky)} \right) \quad , \tag{50}$$

which is the mean square error of the estimated $y_j$ values. The factor $n_{x_k}/n$ accounts for the fact that there are $n$ values of $y_j$, but only $n_{x_k}$ of them are affected by $\hat{\beta}_k$; for the other $n - n_{x_k}$, $x_{jk}$ is missing and $\hat{\beta}_k$ has no influence on $y_j$. In both (49)

and (50), the factor $\frac{n-1}{n-(m+1)-1}$ corrects for degrees of freedom. If one removes this degree-of-freedom correction, one gets the "population" mean square error (i.e., the error variance of the fit to these particular data). With the degree-of-freedom correction, one gets the "sample" mean square error (i.e., an estimate of the prediction error for data drawn from the same population, but not used to fit the model in the first place). When applied to complete data sets (without missing values, and without regularization), Eq. (50) equals the conventional error variance for multiple regression, and it usually works

reasonably well with missing values and with unbiased regularization, e.g., with the consistency criterion of Eq. (46) or the smoothness criterion of Eq. (47). However, unlike in conventional multiple regression, there is no absolute guarantee that the variance of the predicted values (the summation in Eq. 50) will be smaller than the variance of the observed values of $y_j$. Users should therefore be aware that Eq. 50 could potentially yield nonsensical negative values (or unrealistically small positive values) for the error variance in particular cases.

In conventional multiple regression, the covariance matrix of the coefficients $\hat{\beta}_k$ equals the inverse of the covariance matrix $\mathbf{C}$, scaled by the error variance $s_\varepsilon^2$ divided by the sample size $n$. This approach must be adapted to account for the effects of regularization, yielding the following expression for the covariances of the $\hat{\beta}_k$:

$$\left( \mathrm{cov}(\hat{\beta}_k, \hat{\beta}_\ell) \right) = \frac{s_\varepsilon^2}{n_{\mathrm{eff}}} \left( \mathbf{C} + \lambda\mathbf{H} \right)^{-1} \left( \mathbf{C} \right) \left( \mathbf{C} + \lambda\mathbf{H} \right)^{-1} , \tag{51}$$

where $s_\varepsilon^2$ is the error variance as estimated in Eq. (49) or (50), and $n_{\mathrm{eff}}$ is the sample size $n$, adjusted to account for serial correlation in the residuals using Eq. (13). (Where there are so many measurement or analysis failures that residuals cannot be calculated reliably, it is better to guess a reasonable value for their serial correlation than to assume it is zero, which will typically lead to overestimates of $n_{\mathrm{eff}}$ and thus underestimates of the associated uncertainties.) The standard errors of the $\hat{\beta}_k$ will be the square roots of the diagonal elements of the matrix defined by Eq. (51),





$$\mathrm{s.\,e.}\left(\hat{\beta}_k\right) = \frac{s_\varepsilon}{\sqrt{n_{\mathrm{eff}}}}\sqrt{\left[\left(\quad \mathbf{C} + \lambda\mathbf{H}\quad\right)^{-1}\left(\quad \mathbf{C}\quad\right)\left(\quad \mathbf{C} + \lambda\mathbf{H}\quad\right)^{-1}\right]_{kk}} \quad, \qquad (52)$$

Benchmark data sets verify that Eqs. (51) and (52) perform as they should: the root-mean-square averages of the calculated s. e. $\left(\hat{\beta}_k\right)$ are close to the root-mean-square averages, over many replicate data sets, of the deviation of the fitted coefficients $\hat{\beta}_k$ from the true $\beta_k$ used to generate the synthetic data. This result holds both with and without substantial fractions of

missing values, strong correlations among the $\boldsymbol{X}_k$, and substantial additive noise.

There is one important caveat to this generalization, however: it holds only if the assumptions underlying the regularization criterion *are actually true*. For example, if the true $\beta_k$ vary smoothly with $k$, then regularization using Eq. (47) will retrieve a set of smoothly varying coefficients $\hat{\beta}_k$ that deviate from the true $\beta_k$ by amounts that are well approximated by the

calculated standard errors s. e. $\left(\hat{\beta}_k\right)$. But if (say) the true $\beta_k$ oscillate wildly from one $k$ to the next, regularization using Eq. (47) will generate a smoothly varying set of $\hat{\beta}_k$ which will deviate from the true (wildly oscillating) $\beta_k$ by much more than the calculated standard errors s. e. $\left(\hat{\beta}_k\right)$ as calculated from Eq. (52). Regularization methods are forced to assume that the $\beta_k$ obey the regularization criterion (with the strength of this assumption determined by the parameter $\lambda$), and thus they cannot be used to test whether this assumption is true. Thus what the calculated standard errors tell us is that *if* the true $\beta_k$ vary

smoothly over $k$, then the estimation errors of the $\hat{\beta}_k$ should be on the order of s. e. $\left(\hat{\beta}_k\right)$.

### 4.5 Transit time distribution of discharge

The coefficients $\hat{\beta}_k$ determined by Eqs. (38)-(52) estimate the average contribution to discharge $Q_j$ that originated as precipitation $k$ time steps earlier; that is, they estimate the average of $q_{jk}/Q_j$ for combinations of times $j$ and $k$ for which precipitation occurred at $i = j - k$. They do not account for times $i$ when no precipitation occurred, and thus for which

$q_{jk} = 0$ at the corresponding time steps $j = i + k$.

To estimate the average contribution $q_{jk}/Q_j$ of precipitation to discharge across all time steps, both with and without precipitation, we need to include values of $q_{jk} = 0$ for times without precipitation (and thus without any contribution of precipitation to discharge). This is done by re-scaling the coefficients $\hat{\beta}_k$ and their uncertainties s. e. $\left(\hat{\beta}_k\right)$ by $n_{x_k}/n$, the ratio

of "event" time steps (those with precipitation) to all time steps; doing so yields the transit time distribution of discharge ${}^Q TTD_k$ (also termed the "backward" transit time distribution, or the transit time distribution conditioned on exit time):

$$^Q TTD_k = \hat{\beta}_k\,\frac{n_{x_k}}{n} \quad, \quad \mathrm{s.\,e.}\left({}^Q TTD_k\right) = \mathrm{s.\,e.}\left(\hat{\beta}_k\right)\frac{n_{x_k}}{n} \quad. \qquad (53)$$

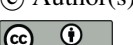



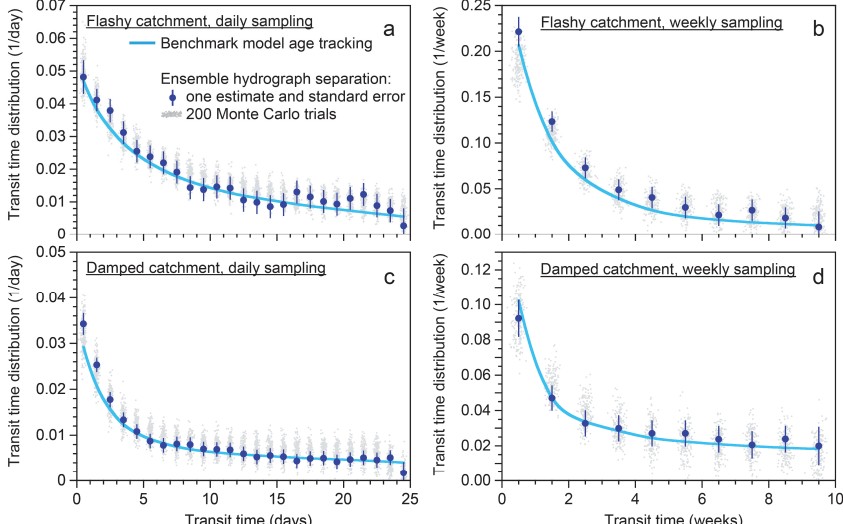

**Figure 11. Transit time distributions of discharge estimated by ensemble hydrograph separation based on both daily and weekly tracer sampling, versus true transit time distributions determined by benchmark model age tracking (light blue curves). Upper and lower rows show TTD's for the modeled flashy and damped catchments, both driven by Smith River (Mediterranean climate)**
5 **precipitation. Dark blue symbols show transit time distributions estimated from one time series. Data clouds (light gray) show 200 different realizations of random precipitation tracer values, random missing data, and random measurement errors. Ensemble hydrograph separation correctly reveals the shapes of the transit time distributions and also quantifies their values, within the calculated uncertainties, at most lags. It can clearly distinguish the transit time distributions of the two catchments under either daily or weekly tracer sampling.**

These transit time distributions can be tested by comparing them to time-averaged streamwater age distributions calculated by age tracking in the benchmark model (Sect. 3.1). Figure 11 shows the results of several such tests, using both daily and weekly tracer data as input (left and right columns, respectively). The light blue curves indicate the true time-averaged transit time distribution (determined from age tracking in the benchmark model), the dark blue symbols show transit time

15 distributions estimated from one tracer time series, and the gray data clouds show 200 more transit time distributions from the same model with different realizations of the random inputs. The weekly TTD's are larger, in absolute terms, than the daily TTD's, because streamflow will always contain at least as much water that originated as precipitation during the previous week as during the previous day (for the simple reason that the previous day is part of the previous week). Figure 11 shows that ensemble hydrograph separation correctly estimates the general shapes of the TTD's, and their quantitative

20 values. Furthermore, the gray data clouds show that no TTD estimates deviate too wildly from the age tracking curves.



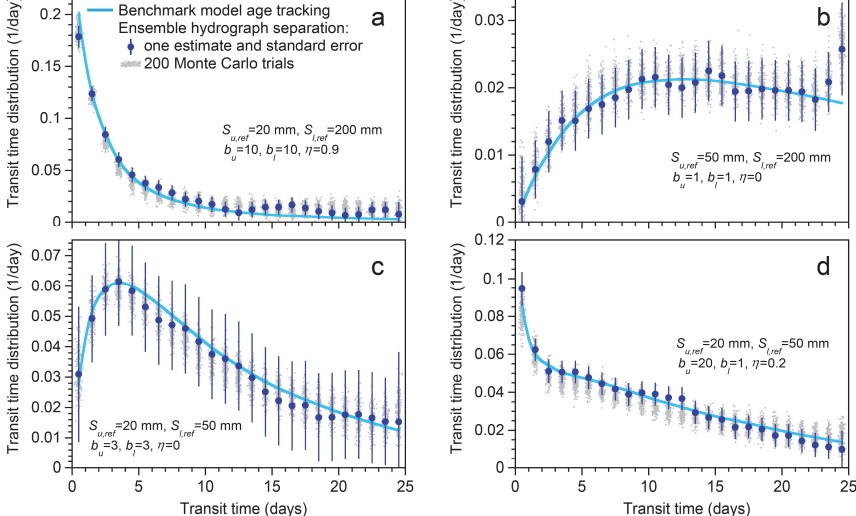

**Figure 12. Transit time distributions (TTD's) of discharge estimated by ensemble hydrograph separation based on daily sampling, compared to true TTD's determined by benchmark model age tracking (light blue curves), for four model parameter sets yielding diverse patterns of transport behaviour. Dark blue symbols show transit time distributions estimated from one time series. Data**
**clouds (light gray) show 200 different realizations of random precipitation tracer values, random missing data, and random measurement errors. Vertical axis scales differ greatly. Ensemble hydrograph separation correctly reveals the shapes of the TTD's and also quantifies their values at most lags. However, panels b and c show that standard errors are overestimated for TTD's that result in strong serial correlation in the modelled time series (see text).**

Real-world transit time distributions could potentially have different shapes from those shown in Fig. 10. To test whether ensemble hydrograph separation can correctly estimate transit time distributions with more widely varying shapes, I explored the benchmark model's parameter space, in some cases venturing beyond the nominal parameter ranges outlined in Sect. 3.1. As Fig. 12 illustrates, widely differing time-averaged (or "marginal") transit time distributions generated by the benchmark model (solid lines) are well approximated by the ensemble hydrograph separation estimates (blue dots) calculated from the

tracer time series. The standard errors are overestimated for humped TTD's, which generate strongly autocorrelated time series. The reason appears to be that when the benchmark model's parameters generate a strongly autocorrelated tracer time series, the residuals will also be strongly autocorrelated; thus the effective sample size $n_{\text{eff}}$ will be small (Eq. 13) and the resulting uncertainties s. e. $\left(^{Q}TTD_k\right)$ will be correspondingly large (Eqs. 52-53). Also, in some TTD's the last few lags can exhibit substantial deviations from the age tracking results (e.g., Fig. 12b). This may be an aliasing effect that arises when

$C_{Q_{j-m-1}}$ does not adequately capture the effects of the unmeasured older fluxes (see Eqs. 30-33). One would expect this aliasing effect to be strongest in time series for which the TTD does not approach zero at the longest measured lags (such as in Fig. 12b). Pragmatic solutions to this problem could include ignoring the last few lags of the estimated TTD, and/or estimating the TTD over a wider range of lags so that it converges to nearly zero. These caveats notwithstanding, Figs. 11



and 12 demonstrate that ensemble hydrograph separation can reliably quantify transit time distributions with widely varying shapes.

**4.6 Volume-weighted transit time distribution**

The transit time distributions defined in Eq. (53) are ensemble averages in which each day counts equally; that is, for a given

lag $k$, $^{Q}TTD_k$ estimates the average of the ratio $q_{jk}/Q_j$ across all time steps, including zeroes at time steps for which there was no precipitation at the corresponding time step $i = j - k$. Thus Eq. (53) estimates *time-weighted* average TTD's, the distribution of temporal origins of an average *day* of discharge.

For many purposes, it would be useful to estimate the temporal origins of an average *liter* of discharge instead, that is, the

*volume-weighted* TTD, which we can denote $^{Q}TTD_k^*$ (where, following the convention in Sect. 2, the asterisk indicates volume-weighting). Instead of estimating the average of the ratio $q_{jk}/Q_j$ (the time-weighted average), a volume-weighted TTD approximates the ratio of the average $q_{jk}$ to the average $Q_j$ across all time steps (the ratio of the averages rather than the average of the ratios). This is the multi-dimensional analogue to the volume-weighted new water fraction presented in Sect. 2.4, and is handled similarly. The multiple regression in Eq. (34) can be volume-weighted by multiplying the $y_j$ and

$x_{jk}$ by $\sqrt{Q_j}$:

$$y_j = \sum_{k=0}^{m} \beta_k^* \, x_{jk} \; + \alpha + \varepsilon_j \quad , \quad y_j = \sqrt{Q_j} \left( C_{Q_j} - C_{Q_{j-m-1}} \right) \quad , \quad x_{jk} = \sqrt{Q_j} \left( C_{P_{j-k}} - C_{Q_{j-m-1}} \right) \quad . \quad (54)$$

This volume-weighted regression can be solved by the same procedures described in Sects. 4.2-4.4, yielding volume-weighted estimates of the coefficients $\hat{\beta}_k^*$ (where, as above, the asterisk indicates volume-weighting). Following the approach of Sect. 2.5, one should account for the unevenness of the weighting when calculating the effective sample size $n_{\text{eff}}$

to be used in estimating the uncertainties in the $\hat{\beta}_k^*$,

$$n_{\text{eff}_k} = \frac{\left( \sum Q_{j(ky)} \right)^2}{\sum \left( Q_{j(ky)}^2 \right)} \quad , \quad (55)$$

where $n_{\text{eff}_k}$ is the effective sample size at lag $k$, and $Q_{j(ky)}$ denotes discharge during time steps $j$ for which pairs of $y_j$ and $x_{jk}$ exist (for a given lag $k$).

To estimate the volume-weighted TTD, we must average over all discharge (including discharge after time steps with no precipitation). Thus the coefficients $\hat{\beta}_k^*$ and their uncertainties should be re-scaled, following the approach in Sect. 2.5, as follows:





$$^{Q}TTD_k^* = \frac{\bar{Q}_{x_k}}{\bar{Q}} \frac{n_{x_k}}{n} \hat{\beta}_k^* \quad , \quad \text{s.e.}\left(^{Q}TTD_k^*\right) = \frac{\bar{Q}_{x_k}}{\bar{Q}} \frac{n_{x_k}}{n} \text{ s.e.}\left(\hat{\beta}_k^*\right) \tag{56}$$

where $\bar{Q}_{x_k}$ is the average discharge during the $n_{x_k}$ time steps $j$ for which precipitation fell at $i = j - k$, $\bar{Q}$ is the average

discharge over all time steps (including rainless periods), $n_{x_k}/n$ is the fraction of time steps with precipitation, and $\hat{\beta}_k^*$ and

s.e. $\left(\hat{\beta}_k^*\right)$ are estimated from the multiple regression in Eq. (54), with the effective sample size $n_{\text{eff}_k}$ defined in Eq. (55). The

ratio $\bar{Q}_{x_k}/\bar{Q}$ corrects for any differences in average discharge during sampled and un-sampled time steps, and the ratio $n_{x_k}/n$

corrects for rain-free periods, which contribute no "new" water to streamflow.

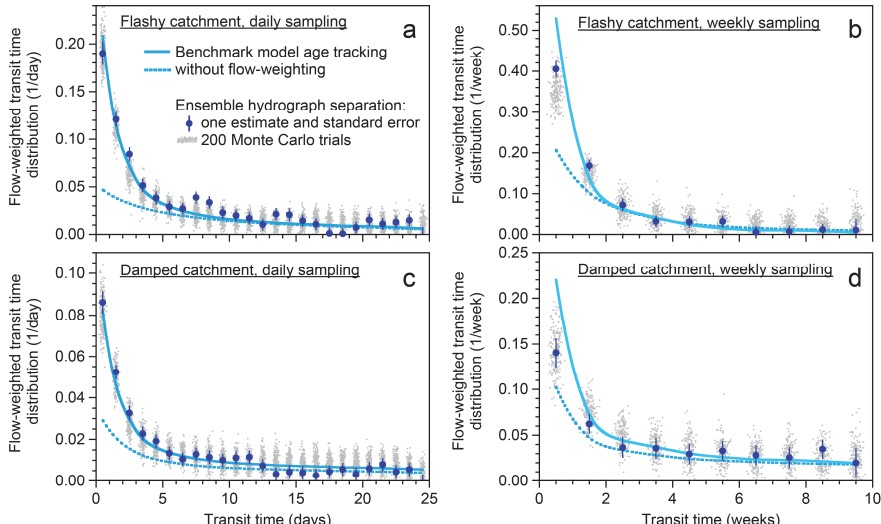

**Figure 13. Volume-weighted transit time distributions (TTD's) of discharge estimated by ensemble hydrograph separation (Eq.**
**56) compared to benchmark model age tracking. Upper and lower panels show TTD's for rapid and damped response**
**parameters, respectively; model is driven by Smith River precipitation in both cases. Ensemble hydrograph separation estimates**
**from tracer fluctuations (dark blue symbols) are broadly consistent with true TTD from age tracking in benchmark model (solid**
**curve). Data clouds show ensemble hydrograph separation results (slightly jittered on x-axis) from 200 different realizations of**
**random precipitation tracer values, random missing data, and random measurement errors. Dashed curve is unweighted**
**benchmark model TTD from Fig. 11 for comparison.**

**4.7 Forward transit time distribution**

In addition to the "backward" transit time distributions $q_{jk}/Q_j$, which estimate the fraction of streamflow that originated as

precipitation $k$ time steps earlier, it may also be useful to estimate "forward" transit time distributions $q_{jk}/P_{j-k} = q_{i+k,k}/P_i$,

which estimate the fraction of precipitation that becomes streamflow $k$ time steps later. Instantaneous, time-varying forward

and backward transit time distributions can differ markedly at any point in time. For example, today's backward transit time




distribution strongly depends on the timing and magnitude of previous precipitation supplying today's streamflow, whereas the forward transit time distribution strongly depends on how future precipitation mobilizes water stored from today's rainfall. These individual differences become less prominent when averaged over a large ensemble of events. Systematic differences nonetheless persist, because forward transit time distributions are defined only during periods with precipitation

(otherwise both $q_{jk}$ and $P_{j-k}$ are both zero and their ratio is undefined), and during these periods precipitation must be higher, on average, than discharge (otherwise there can be no recharge of storage to supply discharge during rainless periods.

Forward transit time distributions are less straightforward to estimate from tracers than backward distributions are, for the simple reason that although streamflow is a mixture of contributions from previous precipitation events, the converse does

not hold: that is, precipitation cannot be expressed as a mixture of subsequent streamflows. Although it is algebraically straightforward to rewrite Eq. (33) as either

$$\left( C_{Q_j} - C_{Q_{j-m-1}} \right) = \sum_{k=0}^{m} \frac{q_{jk}}{P_{j-k}} \left( \frac{P_{j-k}}{Q_j} \left( C_{P_{j-k}} - C_{Q_{j-m-1}} \right) \right) + \alpha + \varepsilon_j \tag{57}$$

or

$$Q_j \left( C_{Q_j} - C_{Q_{j-m-1}} \right) = \sum_{k=0}^{m} \frac{q_{jk}}{P_{j-k}} \left( P_{j-k} \left( C_{P_{j-k}} - C_{Q_{j-m-1}} \right) \right) + \alpha + \varepsilon_j \quad , \tag{58}$$

regressions based on these equations do not reliably predict the average of $q_{jk}/P_{j-k}$ when applied to synthetic data from the benchmark model. (Note that these are the multi-dimensional counterparts to Eqs. (24) and (25), which likewise fail benchmark tests.)

Instead, by analogy Eq. (20), we can estimate the forward transit time distribution from the regression coefficients $\hat{\beta}_k$ of Eq.

(42), rescaled as

$$^{P}TTD_k = \hat{\beta}_k \ \frac{\bar{Q}_{x_k}}{\bar{P}_{x_k}} \quad , \quad \text{s.e.} \left( ^{P}TTD_k \right) = \ \text{s.e.} \left( \hat{\beta}_k \right) \frac{\bar{Q}_{x_k}}{\bar{P}_{x_k}} \tag{59}$$

where

$$\bar{P}_{x_k} = \frac{1}{n_{x_k}} \sum_{j=1}^{n} \left\{ \begin{array}{ll} P_{j-k} & : P_{j-k} \geq P_{\text{threshold}} \\ 0 & : P_{j-k} < P_{\text{threshold}} \end{array} \right. \tag{60}$$

is the average precipitation rate during time steps with precipitation, and




$$\bar{Q}_{x_k} = \frac{1}{n_{x_k}} \sum_{j=1}^{n} \begin{cases} Q_j & : P_{j-k} \geq P_{\text{threshold}} \\ 0 & : P_{j-k} < P_{\text{threshold}} \end{cases} \tag{61}$$

is the average of the discharges that occur $k$ time steps after each of these precipitation intervals. Figure 14 shows that

forward transit time distributions estimated with Eq. (59) are broadly consistent with true forward TTD's calculated by age

tracking in the benchmark model.

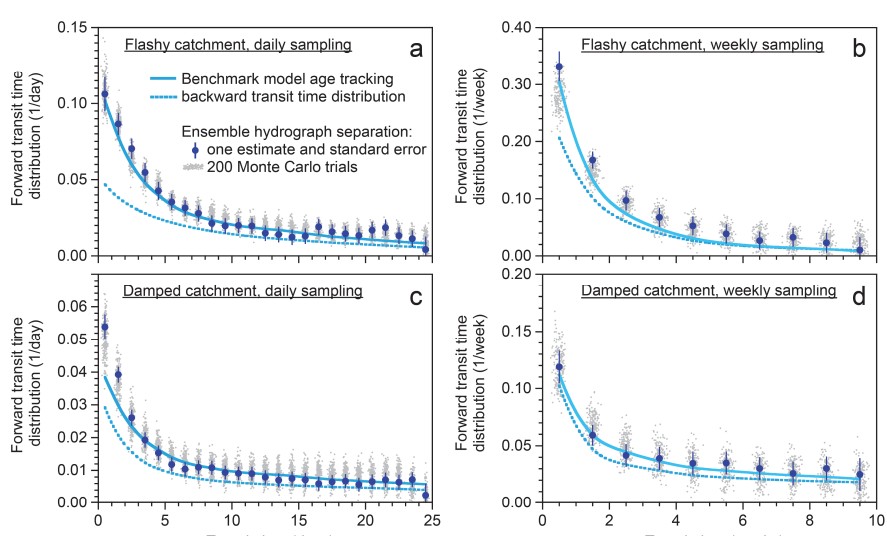

**Figure 14. Forward transit time distributions (the fraction of precipitation that leaves the catchment within one time step, two time steps, and so on) estimated by ensemble hydrograph separation (Eq. 59) compared to benchmark model age tracking. Upper and lower panels show TTD's for flashy and damped catchments, respectively; the model is driven by Smith River (Mediterranean climate) precipitation in both cases. Ensemble hydrograph separation estimates from tracer fluctuations (dark blue symbols) are broadly consistent with true TTD's from age tracking in the benchmark model (solid curve). Data clouds show ensemble hydrograph separation results (slightly jittered on x-axis) from 200 different realizations of random precipitation tracer values, random missing data, and random measurement errors. Dashed curve is the benchmark model backward TTD from Fig. 11 for comparison.**

The volume-weighted forward transit time distribution $^{\text{P}}TTD_k^*$ can similarly be calculated by rescaling arguments, analogous

to the approach in Sect. 2.7. The key is to recognize that we are seeking the ratio between the total volume of precipitation

that will leave the catchment $k$ days after falling as precipitation (the sum of $q_{jk}$ over all $j$), and the total volume of

precipitation that fell on the catchment during the corresponding rainy days. The numerator of this ratio is identical to the

20   numerator of the volume-weighted "backward" transit time distribution $^{\text{Q}}TTD_k^*$, but the denominator is total precipitation

rather than total discharge. Thus the precipitation-weighted forward transit time distribution can be estimated as



$$^{\mathrm{P}}TTD_k^* = \frac{\bar{Q}}{\bar{P}} \, ^{\mathrm{Q}}TTD_k^* = \frac{\bar{Q}_{x_k}}{\bar{P}} \frac{n_{x_k}}{n} \hat{\beta}_k^* \quad , \quad \text{s.e.} \left( ^{\mathrm{P}}TTD_k^* \right) = \frac{\bar{Q}}{\bar{P}} \, \text{s.e.} \left( ^{\mathrm{P}}TTD_k^* \right) = \frac{\bar{Q}_{x_k}}{\bar{P}} \frac{n_{x_k}}{n} \, \text{s.e.} \left( \hat{\beta}_k^* \right) \quad . \quad (62)$$

Because the benchmark model in Fig. 1 has no evaporative losses and thus $\bar{Q} = \bar{P}$, benchmark tests of the precipitation-weighted forward TTD ( $^{\mathrm{P}}TTD_k^*$) and the discharge-weighted backward TTD ( $^{\mathrm{Q}}TTD_k^*$) will yield identical results; thus the benchmark test of $^{\mathrm{Q}}TTD_k^*$ (Fig. 13) will not be repeated here as a test of $^{\mathrm{P}}TTD_k^*$ .

**4.8 Variations in transit time distributions with discharge, precipitation, antecedent moisture, and seasonality**

Like the new water fraction $F_{\mathrm{new}}$, estimating the transit time distribution $^{\mathrm{Q}}TTD_k$ does not require unbroken time series. Thus, using approaches similar to those outlined in Sect. 3.5, one can estimate transit time distributions for subsets (including discontinuous subsets) of the precipitation and streamflow time series that reflect conditions of particular interest. In the case of new water fractions, subdividing the source data is relatively simple, because new water fractions are

estimated from precipitation and streamflow tracers at the same time steps; thus when one subdivides the streamflow time series one also subdivides the precipitation time series, and vice versa.  Transit time distributions are not so simple, because each discharge time $j$ is potentially affected by $k = 1 \dots m$ precipitation time steps $i = j - k$; thus the precipitation and streamflow time series can be subdivided differently, according to different criteria.

For example, we can choose to subdivide the data set according to the discharge time $j$, thus evaluating Eq. (34) only for time steps $j$ that meet particular criteria (for example, to analyze time steps with high or low flows separately).  Doing so has the effect of creating blank rows in the vector $\mathbf{Y}$ and matrix $\mathbf{X}$ in Eq. (37) for each excluded value of $j$. Figure 15 shows the results of estimating transit time distributions $^{\mathrm{Q}}TTD_k$ using only the highest 20% of discharges (the corresponding $^{\mathrm{Q}}TTD_k$'s calculated from the entire time series are also shown for comparison).  Because large inputs of recent precipitation are likely

to result in high flows, one would intuitively expect that high flows should contain larger contributions from recent precipitation.  But how much larger?  As Fig. 15 shows, this question can be answered, at least on average, by examining the transit time distributions of high-flow discharges.  Figure 15 shows that ensemble hydrograph separation can accurately estimate the transit time distributions of both high flows and normal flows, and thus can accurately quantify how transport behavior is different under high-flow conditions.





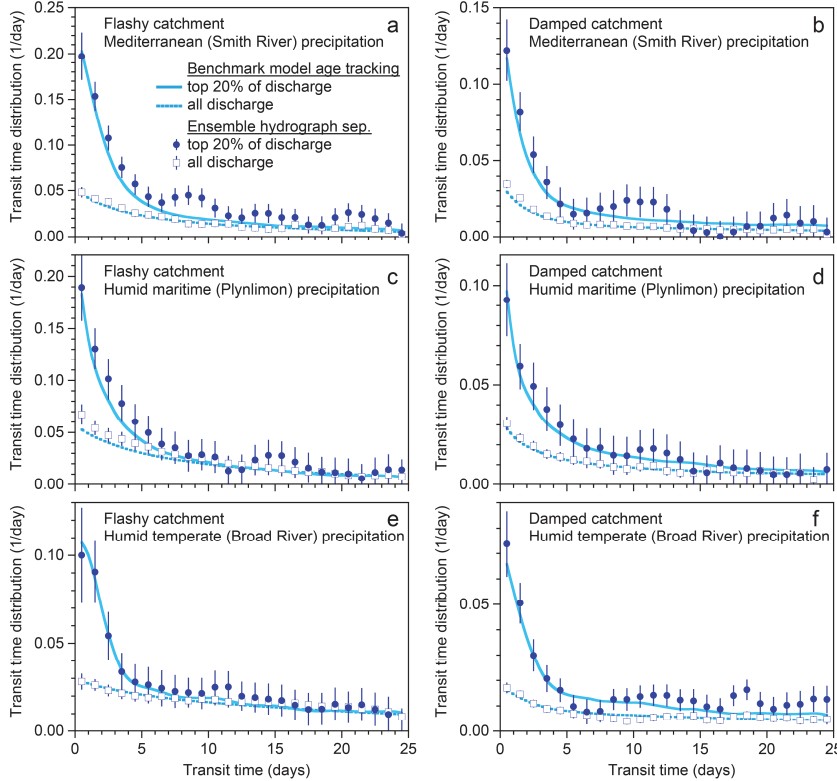

**Figure 15. Transit time distributions $^QTTD_k$ for high flows (the highest 20% of daily discharges; solid curve and solid circles), compared to transit time distributions for all flows (dashed curve and open squares). Solid circles and open squares show $^QTTD_k$ estimates from ensemble hydrograph separation (Eq. 53); solid and dashed curves show true $^QTTD_k$ determined by age tracking in the benchmark model. Left and right panels show TTD's for flashy and damped catchments, respectively; the three rows of panels represent three different precipitation drivers. Note that vertical axis scales differ substantially. High flows have much larger contributions of recent precipitation than average flows do. Ensemble hydrograph separation quantitatively captures this behavior across flashy and damped model catchments with all three precipitation drivers.**

In a Mediterranean climate (as depicted by, for example, the Smith River precipitation record shown in Fig. 1), one would intuitively expect rainy-season streamflow to have larger contributions from recent precipitation. Conversely, one would expect that dry-season streamflow will have much smaller contributions from recent rainfall (because there is so little of it, among other reasons). But how big are the differences between rainy-season and dry-season transit time distributions? As an illustration of what may be possible with real-world data, I took the five-year daily and weekly time series for the benchmark model driven by the Mediterranean climate (Smith River) precipitation record, and separated them into summer (dry) and winter (wet) seasons, and analyzed the two seasons separately. Figure 16 shows that, as expected, the



contributions of recent precipitation to streamflow are much larger during the wet season than the dry season. But more

importantly, Figure 16 also shows that these differences can be accurately quantified, directly from data.

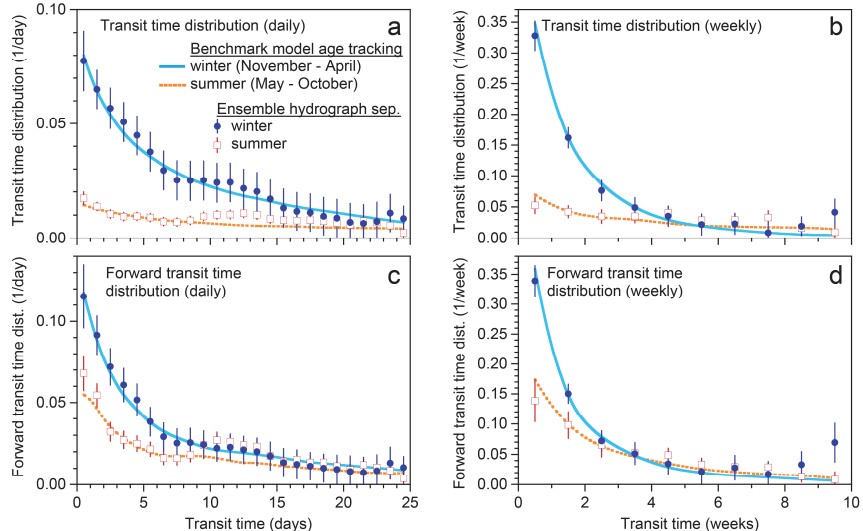

**Figure 16. Backward and forward transit time distributions (upper and lower panels, respectively) compared for summer (May-October) and winter (November-April) months, from benchmark model with Mediterranean (Smith River) precipitation climatology and flashy catchment parameters. Solid circles and open squares show estimates from ensemble hydrograph separation (Eqs. 53 and 59); solid and dashed curves show the true TTD's determined by age tracking in the benchmark model. Left and right panels show TTD's estimated from daily and weekly sampling, respectively. Owing to larger and more frequent**
**rainfalls during winter (see Fig. 1), transit time distributions calculated for the winter months show a much larger contribution of recent rainfall to current streamflow (upper panels), and a much larger fraction of current precipitation becoming streamflow in the near future (lower panels). Ensemble hydrograph separation quantitatively captures the seasonal differences in the benchmark model's transit time distributions.**

The examples above are based on subdividing the data set according to the discharge time $j$. It is also possible to subdivide

the data according to precipitation times $i = j - k$ that meet particular criteria (for example, to analyze time steps with large

and small rainstorms separately). Doing so has the effect of creating diagonal stripes of blanks in the matrix **X** in Eq. (37) at

$j = i + k$ for each excluded value of $i$. These are in addition to the diagonal stripes of missing values that arise because of

sampling and measurement failures, or more commonly because no rain fell. Thus they pose no new mathematical

challenges, and can be handled by the methods outlined in Sect. 4.2.

One question that can be explored by subdividing the time series according to precipitation is whether larger rainfall events

propagate faster through catchments. Intuition suggests that intense rainfall should lead to larger contributions to streamflow

from faster flowpaths. But how much larger? Figure 17 illustrates how this kind of question could potentially be explored.



In Fig. 17, the forward transit time distributions of the highest 20% of precipitation are compared to the average transit time distributions of all precipitation events, for the damped and flashy parameter sets and all three precipitation climatologies. One can see that large rain events are associated with much larger amounts of water reaching the stream quickly, but this effect largely disappears after about 2-3 days. Moreover, the magnitude and timing of this effect are nearly the same in the

estimates derived from ensemble hydrograph separation and benchmark model age tracking, suggesting that they could also be reliably estimated from real-world data.

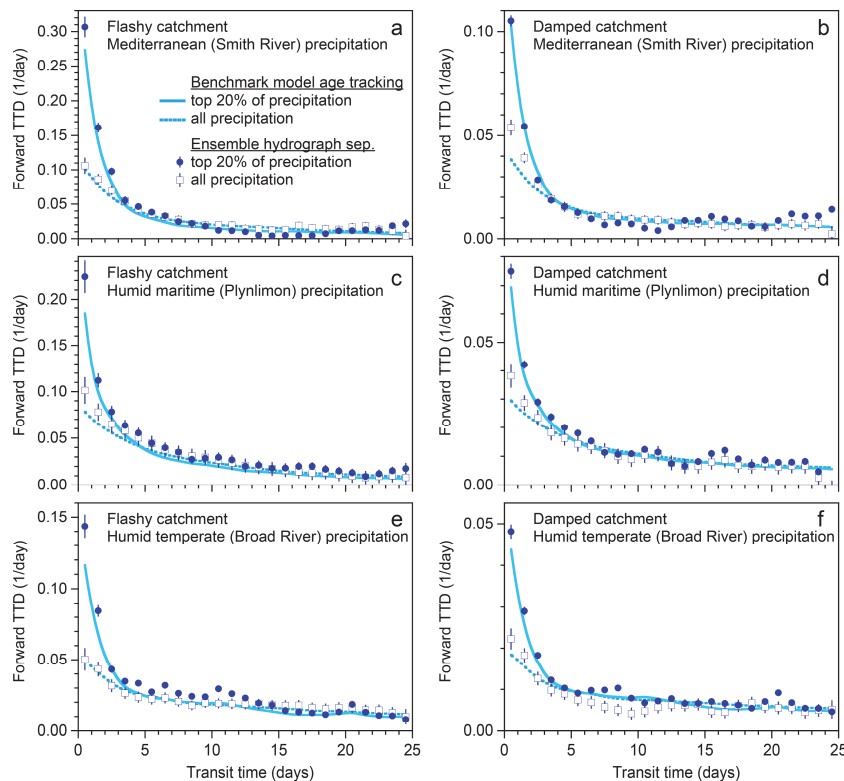

**Figure 17. Forward transit time distributions $^\mathrm{P}TTD_k$ for intense precipitation (the highest 20% of daily precipitation totals; solid**
**curve and solid circles), compared to forward transit time distributions for all precipitation (dashed curve and open squares). Solid circles and open squares show $^\mathrm{P}TTD_k$ estimates from ensemble hydrograph separation (Eq. 59); solid and dashed curves show the true $^\mathrm{P}TTD_k$ determined by age tracking in the benchmark model. Left and right panels show TTD's for flashy and damped catchments, respectively; the three rows of panels represent three different precipitation drivers. Note that vertical axis scales differ greatly. Despite a tendency for ensemble hydrograph separation to over-predict $^\mathrm{P}TTD_k$ for short lag times, the**
**differences between the ensemble hydrograph separation estimates for intense precipitation and normal precipitation (open squares and solid circles) closely mirror the differences between the solid and dashed curves. Thus ensemble hydrograph separation can estimate the relative effect of intense precipitation on forward transit times, across widely differing precipitation drivers and catchment characteristics.**




Antecedent wetness has been recognized as a controlling factor in catchment storm response (e.g., Detty and McGuire, 2010; Merz et al., 2006; Penna et al., 2011), but its effects on solute transport at the catchment scale have not been widely explored. To assess whether ensemble hydrograph separation might be helpful in exploring this question, I binned the

benchmark model time series into ranges of antecedent moisture (as measured by the upper box storage values $S_u$ at the end of the previous day), and estimated the new water fractions $^QF_{new}$ and $^PF_{new}$ using ensemble hydrograph separation. I used the upper box storage as a proxy for measurements of soil moisture or shallow groundwater levels, which are commonly used as indicators of antecedent wetness in catchment studies (one could use antecedent discharge as a proxy instead; this would yield nearly equivalent results). As Figs. 18a and c show, ensemble hydrograph separation accurately predicts how

both "backward" and "forward" new water fractions increase as functions of antecedent moisture.

To visualize how high antecedent moisture affects transit time distributions, I isolated the discharge times $t_j$ associated with the highest 10% of antecedent moisture values, and calculated the corresponding backward transit time distribution $^QTTD_k$ (Fig. 18b). This TTD shows that high antecedent moisture is associated with large contributions of recent rainfall to

streamflow, up to lags of about 3-4 days. The peak of the transit time distribution does not come at the shortest possible lag (same-day precipitation), but instead at a lag of 1.5 days (i.e., previous-day precipitation). This is the inevitable result of selecting points with high previous-day moisture, which are likely to be associated with high previous-day precipitation (and thus high contributions of that previous-day precipitation to current streamflow). Storms typically last about 2-3 days in the Smith River precipitation record underlying the simulations in Fig. 18, so much of the backward TTD could potentially just

reflect the pattern of precipitation, combined with the fact that points with high antecedent moisture have been selected.

One can even question why one would expect a *backward* TTD to help in understanding the effects of antecedent moisture at all, given that the backward TTD will mostly reflect precipitation inputs that came *before* (and, in some cases, created) the antecedent moisture conditions themselves. A *forward* TTD, on the other hand, might help in quantifying how antecedent

moisture affects the transmission of *future* precipitation to streamflow. I therefore isolated the precipitation times $t_i = t_{j-k}$ associated with the highest 10% of antecedent moisture values (thus, as explained above, filtering the matrix **X** in Eq. (37) along diagonal lines), and calculated the corresponding forward transit time distribution $^PTTD_k$ (Fig. 18d). As Fig. 18d shows, in this model system, high antecedent moisture roughly doubles the proportion of precipitation that reaches the stream, but only out to lags of approximately two days, beyond which there is no clearly detectable effect. Naturally, these

inferences pertain only to the model system, and do not tell us how real-world catchments might behave. However, because Fig. 18 shows that new water fractions and transit time distributions can be accurately quantified across a range of antecedent moisture conditions using ensemble hydrograph separation, it illustrates how this technique could be used to explore the effects of antecedent moisture in real-world catchments.





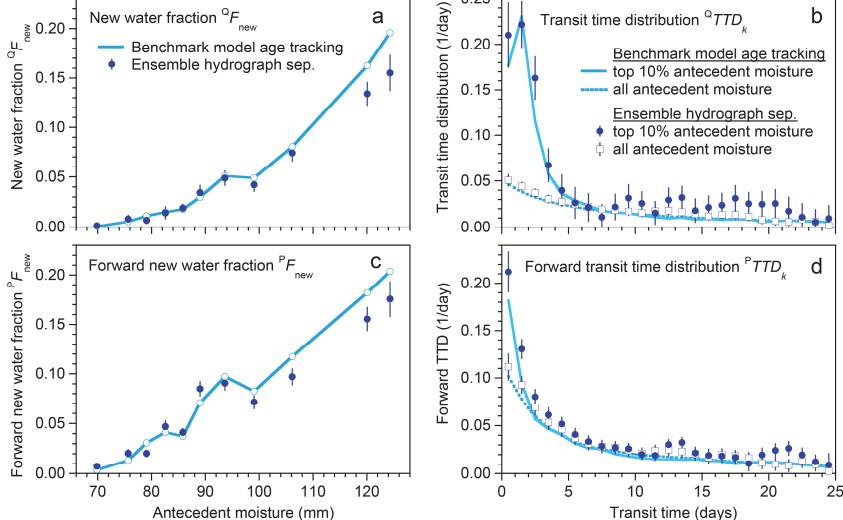

**Figure 18. Effects of antecedent moisture on new water fractions and transit time distributions (upper panels) and their "forward" counterparts (lower panels). Left-hand panels show new water fractions from benchmark model age tracking (solid curves) and ensemble hydrograph separation (solid circles) stratified by percentiles of antecedent moisture (previous-day storage $S_u$ in the benchmark model's upper box). Right-hand plots show transit time distributions for high antecedent moisture conditions (the highest 10% of previous-day storage levels in the upper box of the benchmark model; solid curve and solid circles), compared to transit time distributions for all antecedent moisture levels (dashed curve and open squares). All panels are derived from simulations with flashy catchment parameter set driven by Smith River (Mediterranean climate) precipitation time series. Error bars are one standard error.**

## 5 Discussion

Over 20 years ago, Rodhe et al. (1996) wrote that transit times, despite their importance to modeling discharge, were "impractical to determine experimentally except in rare manipulative experiments where catchment inputs can be adequately controlled." Despite over two decades of effort, including increasingly elaborate theoretical discussions of transit time distributions, the problem identified by Rodhe et al. remains: how can we measure transit times, and transit time distributions, of real-world catchments under real-world conditions? And how can we verify whether the estimates we get are realistic ones? The theory and benchmark tests presented in Sects. 2-4 aim to provide a partial answer.

### 5.1 Comparisons with other approaches

Particularly because their names are similar, it is important to recognize how *ensemble* hydrograph separation contrasts with conventional hydrograph separation. Although one could view Eq. (9) as an algebraic re-arrangement of the conventional

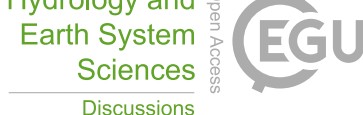



hydrograph separation equation (Eq. 3), with both sides multiplied by ($C_\text{new}$-$C_\text{old}$) and $C_{Q_{j-1}}$ substituted in place of $C_\text{old}$,

there are important differences between the two approaches:

1. Conventional hydrograph separation estimates the *time-varying* new water fraction $F_{\text{new}_j}$ at each individual point in time. By contrast, ensemble hydrograph separation estimates the *average* new water fraction $F_\text{new}$ over an *ensemble* of points (hence the name).

2. Conventional hydrograph separation assumes that the end-member tracer signatures are *constant*, but ensemble hydrograph separation assumes them to be *time-varying*; indeed, it exploits their variability through time as its main source of information.

3. Conventional hydrograph separation requires that all end-members that contribute to streamflow must be identified, sampled, and measured. Ensemble hydrograph separation, by contrast, requires tracer measurements only from streamflow and any end-members whose contributions to streamflow are to be estimated. There is no need to assume that all end-members have been identified and measured, just that tracer fluctuations in any un-measured end-members are not strongly correlated with those in measured end-members and in streamflow.

4. Conventional hydrograph separation requires that the end-members' tracer concentrations are distinct from one another; otherwise the solution to Eq. (3) becomes unstable because the denominator is nearly zero. By contrast, ensemble hydrograph separation estimates the new water fraction by regression, and points where the new water and old water concentrations overlap will have almost no leverage on the regression slope (they correspond to points near zero on the x-axes of Figs. 6a-b, 7a-b, 9a, or A1d, for example).

5. Conventional hydrograph separation is vulnerable to biases in tracer measurements, such as could arise from isotopic evaporative fractionation. By contrast, these same biases should have relatively little effect on ensemble hydrograph separation (e.g., Sect. 3.6), because it is based on regressions between tracer fluctuations, and regression slopes are unaffected by constant offsets on either the x or y axes.

It is also useful to contrast ensemble hydrograph separation with other methods for estimating transit time distributions from tracer data. As reviewed by McGuire and McDonnell (2006), these approaches typically convolve the precipitation tracer time series with an assumed transit time distribution, and then adjust the parameters of that distribution to achieve a best fit with the streamflow tracer time series. This convolution approach differs from ensemble hydrograph separation in several important respects:

1. In the convolution approach, the functional form of the transit time distribution must be assumed (although shape parameters often allow the shape of the TTD to be fitted, within a given family of distributions). By contrast, the ensemble hydrograph separation approach makes no assumption about the shape of the distribution; instead, the TTD values at each lag $k$ are estimated directly from data.





2. The convolution approach is based on convolution *integrals*, and thus errors in the input terms accumulate over time. By contrast, the ensemble hydrograph separation approach is based on local *derivatives* of the stream tracer concentrations, and their covariances with fluctuations in the input tracer concentrations at various lags; as a result, errors in the input terms do not accumulate.

3. Missing input data pose a fundamental problem for convolution integrals, whereas they can be readily accommodated in the covariances that underlie the ensemble hydrograph separation approach (Sect. 4.2).

These considerations also generally apply to approaches that use tracer concentrations in rainfall and streamflow to calibrate time-invariant storage selection (SAS) functions, instead of time-invariant transit time distributions (e.g., van der Velde et al., 2012; Harman, 2015).

Yet another approach that is coming into more frequent use is to calibrate a conceptual or physically based model to reproduce, as closely as possible, the observed hydrograph and streamflow tracer time series, and then infer the catchment transit time distribution from particle tracking within the model (e.g., Benettin et al., 2013; Benettin et al., 2015; Remondi et al., 2018). The validity of these inferences requires that the model is not just a good predictor of the calibration data, but

also that the underlying processes are the correct ones – in other words, that the model gets the right answers for the right reasons. It will generally be difficult to verify whether this is the case. Thus it will be difficult to know how much the inferred transit times are determined by the tracer data or by the structural assumptions of the underlying model. Nor does a good fit to the observational data verify the correctness of the model and the inferences drawn from it, because a good fit can imply either that the model is doing everything correctly, or that it is doing multiple things wrong, in offsetting ways.

Of course one can argue that every data analysis approach also implies some underlying model, and one might argue that ensemble hydrograph separation is based on the (implausible) assumption that the transit time distribution is time-invariant. Such an argument would be mistaken. As I have shown, ensemble hydrograph separation neither assumes nor requires that the transit time distribution is stationary (see Appendices A and B). Instead, ensemble hydrograph separation quantifies the

ensemble average of a catchment's *time-varying* transit time distribution, even when that distribution is highly dynamic.

### 5.2 Benchmark testing

Considerable effort has been devoted to benchmark tests of the methods proposed in Sects. 2 and 4. One may naturally ask: why bother? Why not just describe how ensemble hydrograph separation works, and apply it to several field data sets, and see whether it gives reasonable results? One answer is that whether the results seem reasonable only reflects whether they

agree with our preconceptions, not whether they (or our preconceptions) are correct. A second answer is that only through properly designed benchmark tests can we assess how accurate the method is, and what factors might affect its accuracy. Yet another answer is that the benchmark model gives the analysis method a precise target to hit, thus better revealing its strengths and weaknesses.





Benchmark tests also have a role to play in the day-to-day application of data analysis methods like those proposed here. Users may wonder: will this approach work with data from my catchment? Given the data I have, how accurately can I estimate the ensemble average transit time distribution? What kinds of tracer data will be needed to distinguish between two

different conceptualizations of catchment-scale storage and transport? Carefully designed benchmark tests with synthetic data can be helpful in addressing questions such as these.

It should be emphasized that in the tests presented here, the benchmark model knows nothing about how the analysis method works; in fact, its nonlinearity and nonstationarity rather badly violate the assumptions that would make the equations

underlying the analysis exactly true. Conversely, the analysis method knows nothing about the inner workings of the benchmark model. It knows the model inputs and outputs (the water fluxes and tracer concentrations in rainfall and streamflow), but it does not know – and, importantly, it does not need to know – how those outputs were generated. This is important because, for ensemble hydrograph separation to be useful in real-world catchments, its validity must not depend on the particular mechanisms that regulate flow and transport at the catchment scale.

Likewise, its validity must not depend on having unrealistically accurate or complete data. For this reason, the benchmark tests include substantial measurement errors, and substantial numbers of missing values (Sect. 3.1).

Thus these benchmark tests are much stricter than many in the literature. Some benchmark tests, for example, generate the

benchmark data set using exactly the same assumptions that underlie the analysis method itself. That kind of test is guaranteed to succeed (barring math errors), since it is performing the same calculation twice (first forwards, then backwards). At the same time, such tests are not helpful, because they would only be relevant to real-world cases where all of the assumptions underlying the analysis method were exactly true. Such cases are unlikely to exist.

One could argue that the benchmark model presented here would be more realistic if it were (for example) a spatially-distributed three-dimensional model based on Richards' equation, calibrated to a particular research watershed. However, the benchmark model's purpose is to generate a wide variety of targets for the analysis method to hit, with each target precisely defined, rather than to realistically mimic any particular catchment. All that is essential is that it must generate realistically complex patterns of behavior, and exactly compute the "true" new water fractions and transit time distributions

by age tracking. The relatively simple two-box conceptual model that has been used here was chosen because it fulfills both criteria. These criteria did not include consistency with a particular mechanistic view of flow and transport. Likewise, consistency with the assumptions underlying ensemble hydrograph separation was not one of the criteria, nor should it be.





For the same reason, it should be clear that real-world catchments may not necessarily exhibit similar patterns of behavior to those of the benchmark model, as shown in Figs. 6-9 and 15-18. Thus the analyses presented here do not show, for example, that new water fractions are roughly linear functions of discharge (Fig. 6), precipitation (Fig. 7), or antecedent moisture (Fig. 18). These patterns of behavior are properties of the benchmark model and its precipitation forcing. Whether real-world

catchments behave similarly or differently is an open question, and the analysis presented here shows how such questions could potentially be answered.

### 5.3 Errors, biases and uncertainties

The analysis methods outlined in Sects. 2 and 4 include explicit procedures for estimating the uncertainties (as quantified by standard errors) in both new water fractions (Eqs. 11, 15, and 19) and transit time distributions (Eqs. 52, 53, 56, 69, and 62).

These uncertainties are generally realistic predictors of how much the ensemble hydrograph separation estimates deviate from the "true" benchmark values determined from age tracking: the scatter in Figs. 2 and 5, for example, is consistent with the estimated standard errors, and the error bars in Figs. 6, 7, 9, and 11-18 (one standard error in all cases) are usually reasonable estimates of the deviations from the benchmark values (exceptions include the "humped" transit time distributions in Fig. 12, where the uncertainties are overestimated).

Unsurprisingly, the standard errors scale with the scatter (error variance) in the data, and inversely with the square root of the effective number of degrees of freedom. Thus the uncertainties will be larger when the data set is sparse and noisy, and when the new water fraction and/or transit time distribution explains only a small fraction of its variance. It should also be noted that the *relative* standard error can be large, for example when the TTD is small at long lags.

Because ensemble hydrograph separation does not require continuous input data, it can facilitate comparisons among various subsets of a catchment time series, as demonstrated in Sects. 3.5 and 4.7. However, it should be kept in mind that as one cuts the data set into more (and thus smaller) pieces, the statistical sampling variability among the data points remaining in each piece will become more and more influential, and the inferences drawn on each piece will become correspondingly more

uncertain. Thus there will be practical limits to the granularity of the subsampling that can be applied in real-world cases.

In some TTD's, the last few lags exhibit unusually large deviations from the "true" TTD curves derived from age tracking (e.g. Figs. 12b, 13a and c, 14c, 16b and d, and 17b and d; in several of these cases the last point is below zero and thus does not appear on the plot). As noted in Sect. 4.5, I suspect (but cannot prove) that this is an aliasing effect that arises when the

effects of fluxes beyond the longest measured lag are not adequately accounted for by the reference concentration $C_{\text{older}_j} = C_{Q_{j-m-1}}$. In practice this means that TTD values for the last few lags should not be taken too literally, particularly if they deviate from the trend in the previous lags.





Because ensemble hydrograph separation is based on correlations among tracer fluctuations, it is relatively insensitive to systematic biases that produce persistent offsets in the underlying data. For example, isotope ratios in precipitation often vary with altitude, leading to potential biases in precipitation tracer samples (depending on the sampling location). To the

extent that these biases are constant, however, they should not alter regression slopes between tracer fluctuations in precipitation and streamflow (e.g., Figs. A1d, 6a-b, and 7a-b), or their multidimensional counterparts that determine the TTD. The same applies to randomly fluctuating precipitation tracer biases, unless they are large compared to the standard deviation of the tracer concentrations themselves – i.e., unless the fluctuating biases account for most of the variability in the precipitation tracer measurements. Likewise, confounding by any un-measured end-members should be small, unless the un-

measured end-members are correlated with the measured ones, and have a strong influence on stream tracer concentrations.

The uncertainties calculated here, like all error propagation results, depend on the assumptions underlying the analysis (in this case, ensemble hydrograph separation). Under different assumptions, the errors in estimating the average $F_{\text{new}}$ by regression could be larger. For example, if the means of $C_{\text{old}_j}$ and $C_{\text{new}_j}$ differed by much more than their pooled standard

deviations, then variations in $C_{Q_j}$ would mostly be driven by variations in $F_{\text{new}}$ rather than variations in $C_{\text{new}_j}$. This highlights the important contrast between conventional and ensemble methods of hydrograph separation. Conventional hydrograph separation is based on comparing stream tracer values to constant end-members, and therefore works best when the end-members have widely separated means and small variability. By contrast, ensemble hydrograph separation works best when the variations in the end-members are large compared to the differences among their means, because it relies on

correlating tracer fluctuations in streamflow with fluctuations in measured end-members.

### 5.4 Potential applications and extensions

The techniques proposed here quantify the timescales over which catchments store and transport water, and quantify how those timescales change with precipitation, discharge, and antecedent moisture. Such descriptive methods are often grouped under the heading of "catchment characterization". One should keep in mind, however, that a catchment's storage and

transport behavior also depends on its external forcing. If its precipitation climatology were wetter (or drier), for example, its timescales of storage and transport would decrease (or increase) accordingly. Thus transport and storage timescales are not characteristics of the catchment alone, but rather of the catchment and its particular precipitation climatology. By mapping out how a catchment's storage and transport behavior changes with hydrologic forcing (e.g., Figs. 6, 7, 15, 17, and 18), however, ensemble hydrograph separation can contribute to a more complete picture of catchment response.

Alternatively, these patterns of response to hydrologic forcing can be considered as catchment characteristics in their own right.


Because new water fractions and transit time distributions from ensemble hydrograph separation closely match benchmark model age tracking, one might consider using them as a model for catchment transport processes. This will usually be a bad idea. One must remember that ensemble hydrograph separation quantifies ensemble *averages*, which will not be good models of catchment processes unless the real-world transit time distribution is approximately time-invariant. That is

unlikely to be the case.

This observation raises an important point. Ensemble hydrograph separation yields inferences that are phenomenological, not mechanistic. It quantifies *how catchments behave*, but does not, by itself, explain *how they work*. It can nonetheless contribute to mechanistic understanding by precisely quantifying catchment transport behavior, and thus facilitating more

incisive comparisons with models. Examples of possible comparisons include:

-    Do the model and the real-world catchment have similar new water fractions, and "forward" new water fractions (Figs. 2 and 5)?
-    Do these new water fractions change similarly as functions of precipitation and discharge (Figs. 6 and 7)?
-    Do they exhibit similar seasonal patterns (Fig. 9)?

-    Do the model and the real-world catchment have similar transit time distributions, including "forward" transit time distributions (Figs. 11-14)?
-    Do these transit time distributions change similarly as functions of precipitation, discharge, antecedent moisture, and seasonality (Figs. 15-18)?

In this approach to hypothesis testing, key signatures of behavior are extracted from both the model and the data before they

are compared (Kirchner et al., 1996; Kirchner, 2006). This approach stands in contrast to the conventional model-testing paradigm in which model predictions are compared with observational time series through standard goodness-of-fit statistics. The conventional approach ignores the important question of *in what ways* the model predictions deviate from the data. Exploring this question requires diagnostic signatures of catchment behavior like those presented here, and is essential to improving models of catchment processes.

The analysis methods developed here can potentially be extended in several ways. For example, these methods could potentially be applied to infer transit times in other catchment fluxes, such as groundwater seepage or evapotranspiration. They could also be applied to other systems where transit times could be inferred from the propagation of fluctuating tracer inputs; potential examples include not only lakes, oceans, and aquifers, but also the atmosphere and perhaps even organisms.

The multiple regression analysis presented in Sect. 4 demonstrates that one can quantify the contributions of multiple end-members using a single conservative tracer. This is not possible in conventional end-member mixing analysis, which assumes that the end-members are constant and consequently requires that the number of end-members cannot exceed the number of tracers plus one. But because ensemble hydrograph separation is based on correlations of tracer fluctuations, one



tracer can potentially identify many end-members as long as their fluctuations are not too tightly correlated. This is potentially useful, because hydrologists typically have very few truly conservative tracers to work with (arguably only one, in the case of stable isotopes, because $^{18}$O and $^{2}$H are strongly dependent on one another). In the analysis in Sect. 4, the TTD's can be considered to represent 25 different end-members (which are all precipitation, at different lags). However the

5 same approach could be used to analyze (for example) precipitation and snowmelt as sources of streamflow, if tracer time series are available in both candidate end-members and they are not too strongly correlated with one another. Similarly, in large river basins one could potentially quantify the contributions (and transit time distributions) of waters sourced from precipitation in different parts of the catchment – if, again, tracer time series are available for these multiple precipitation sources, and are not too strongly correlated with one another.

Last but not least, the approach presented here can also, with some modifications, be applied to rainfall and streamflow rates in order to quantify the time lags in catchments' hydraulic response to precipitation (reflecting the celerity of hydraulic potentials, as distinct from the velocity of water transport). A follow-up paper describing this "ensemble unit hydrograph" analysis is currently in preparation.

*Competing interests*. The author declares that he has no conflict of interest.

20 *Acknowledgments*. This work was motivated by discussions with Chris Soulsby and Doerthe Tetzlaff during long walks in the Scottish countryside. I also thank Jana von Freyberg, Andrea Rücker, Julia Knapp, Wouter Berghuijs, Paolo Benettin, and Greg Quenell for helpful discussions, and Melissa Heyer for proofreading assistance.



**Table 1.** Definition of symbols (with defining equation, or equation of first use, in parentheses)

| Symbol | Definition |
| --- | --- |
| | |

*Indices / subscripts*

| Symbol | Definition |
| --- | --- |
| $i$ | index for precipitation time steps |
| $j$ | index for discharge time steps |
| $k = j - i$ | index for lags between precipitation and discharge |
| $\ell$ | second index for lags between precipitation and discharge |
| $(xy)$ | parentheses indicate that analysis applies to cases $j$ where neither $x_j$ nor $y_j$ is missing |
| $(ky)$ | parentheses indicate that analysis applies to cases $j$ where neither $x_{jk}$ nor $y_j$ is missing |
| $(k\ell)$ | parentheses indicate that analysis applies to cases $j$ where neither $x_{jk}$ nor $x_{j\ell}$ is missing |

*Benchmark model variables and parameters*

| Symbol | Definition |
| --- | --- |
| $b_\mathrm{u}$ , $b_l$ | upper- and lower-box drainage exponents in benchmark model (Fig. 1) |
| $\eta$ | partitioning coefficient for upper-box drainage in benchmark model (Fig. 1) |
| $L$ | drainage rate from upper box in benchmark model (Fig. 1) |
| $Q_l$ | drainage rate from lower box in benchmark model (Fig. 1) |
| $S_\mathrm{u}$ , $S_l$ | upper- and lower-box storage in benchmark model (Fig. 1) |
| $S_{\mathrm{u,ref}}$ , $S_{l,\mathrm{ref}}$ | reference storage levels in upper and lower boxes of benchmark model (Fig. 1) |

*Other symbols*

| Symbol | Definition |
| --- | --- |
| $\alpha$ | regression intercept (9) |
| $\beta$ , $\hat{\beta}$ | true regression slope (9), and its regression estimate (10) |
| $\beta^*$ , $\hat{\beta}^*$ | true discharge-weighted regression slope (16), and its regression estimate (17) |
| $\beta_k$ , $\hat{\beta}_k$ | true multiple regression slope as function of lag time $k$ (34), and its regression estimate (38) |
| $\hat{\boldsymbol{\beta}}$ | vector of regression estimates $\hat{\beta}_k$ (39) |
| $\varepsilon_j$ | regression error term (9) |
| $\boldsymbol{\varepsilon}$ | vector of regression errors $\varepsilon_j$ (37) |
| $C_{Q_j}$ | tracer concentration in stream discharge at time step $j$ (1) |
| $C_\mathrm{new}$ , $C_\mathrm{old}$ | tracer concentration in new and old water (2) |
| $C_{\mathrm{new}_j}$ , $C_{\mathrm{old}_j}$ | time-varying tracer concentration in new and old water (5) |
| $C_{P_{j-k}}$ | tracer concentration in precipitation at time step $i = j - k$ (31) |
| $C_{\mathrm{older}_j}$ | concentration effects of older tracer inputs, beyond maximum lag $m$ (31) |
| **C** | covariance matrix (44) |
| $F_{\mathrm{new}_j}$ | fraction of new water in streamflow at time step $j$ (3) |
| $F_\mathrm{new}$ | ensemble average of $F_{\mathrm{new}_j}$ (10) |
| $^{Qp}F_\mathrm{new} = \hat{\beta}$ | ensemble average of new water fraction in discharge during time steps with rain (Sect. 2.3) |
| $^{Q}F_\mathrm{new}$ | ensemble average of new water fraction in discharge, including rainless time steps (14) |
| $^{Qp}F_\mathrm{new}^* = \hat{\beta}^*$ | volume-weighted new water fraction in discharge during time steps with rain (17) |
| $^{Q}F_\mathrm{new}^*$ | volume-weighted new water fraction in discharge, including rainless time steps (17) |
| $^{P}F_\mathrm{new}$ | new water fraction of precipitation (20, 26) |
| $^{P}F_\mathrm{new}^*$ | volume-weighted new water fraction in precipitation (27) |
| **H** | Tikhonov-Philips regularization matrix (44) |





| | | |
|---|---|---|
| | $\lambda$ | regularization parameter (44) |
| | $\nu$ | dimensionless regularization parameter (48) |
| | $m$ | maximum lag in transit time distribution |
| | $n$ | number of discharge time steps |
| 5 | $n_{\text{eff}}$ | effective sample size, adjusted for serial correlation and/or uneven weighting (12, 18, 19) |
| | $n_{\text{p}}$ | number of time steps with precipitation (14) |
| | $n_{\text{xy}}$ | number of pairs of $x_j$ and $y_j$ (12) |
| | $n_{x_k}$ | number time steps $j$ with above-threshold precipitation at time step $i = j - k$ (43) |
| | $n_{x_k x_\ell}$ | number time steps $j$ with above-threshold precipitation at both $i = j - k$ and $i = j - \ell$ (43) |
| 10 | $P_j$ | precipitation rate during time step j (21) |
| | $\bar{P}_{\text{p}}$ | average precipitation rate excluding rainless periods (21) |
| | $P_{\text{threshold}}$ | threshold precipitation rate below which tracer inputs are ignored (Sect. 3.1; Eq. 43) |
| | $\bar{P}_{x_k}$ | average precipitation rate during time steps $i = j - k$ with above-threshold precipitation (60) |
| | $Q_j$ | stream discharge at time step $j$ (1) |
| 15 | $Q_{\text{new}_j}, Q_{\text{old}_j}$ | new-water and old-water components of stream discharge (1) |
| | $\bar{Q}$ | average stream discharge (17) |
| | $\bar{Q}_{\text{p}}$ | average stream discharge during time steps with precipitation (17) |
| | $\bar{Q}_{x_k}$ | average stream discharge during time steps $j$ with above-threshold precipitation at step $i = j - k$ (61) |
| | $Q_{j(xy)}$ | stream discharge during time steps $j$ for which neither $x_j$ nor $y_j$ is missing (13) |
| 20 | $Q_{\text{older}_j}$ | unmeasured fluxes from older precipitation inputs, beyond maximum lag $m$ (30) |
| | $q_{jk}$ | volume of water entering as precipitation in time step $i = j - k$ and exiting in time step $j$ (29) |
| | $r_{xy}$ | correlation between $x_j$ and $y_j$ (11) |
| | $r_{\text{sc}}$ | lag-1 serial correlation in regression residuals (12) |
| | $s.e.(\,)$ | standard error (11) |
| 25 | $s_\varepsilon^2$ | variance of regression prediction errors (49, 50) |
| | $^Q TTD_k$ | "backward" transit time distribution of discharge, conditioned on exit time (53) |
| | $^Q TTD_k^*$ | discharge-weighted "backward" transit time distribution (56) |
| | $^P TTD_k$ | "forward" transit time distribution of precipitation, conditioned on entry time (59) |
| | $^P TTD_k^*$ | volume-weighted "forward" transit time distribution (62) |
| 30 | $x_j$ | explanatory variable in linear regression (9) |
| | $x_{jk}$ | explanatory variable in multiple linear regression (34) |
| | $\mathbf{X}$ | matrix of reference-corrected input tracer concentrations $x_{jk}$ (37) |
| | $y_j$ | response variable in linear regression (9, 34) |
| | $\mathbf{Y}$ | vector of reference-corrected streamflow tracer concentrations $y_j$ (37) |





## Appendix A: Estimating non-constant "constants" via regression

A conventional linear regression equation has the form

$$y_j = \beta\, x_j + \alpha + \varepsilon_j \quad , \tag{A1}$$

where $y_j$ and $x_j$ are response and explanatory variables, respectively, measured for individual cases $j$, where $\alpha$ and $\beta$ are (unknown) constants, and where $\varepsilon_j$ is a random (and unknown) additive error term with mean of zero (alternatively, one can consider $\alpha + \varepsilon_j$ to represent all of the unmeasured factors that influence $y_j$). Under the assumption that these unmeasured factors are uncorrelated with $x_j$, linear regression obtains unbiased estimates of $\beta$ from any of several functionally equivalent formulas, including

$$\hat{\beta} \;=\; \frac{\mathrm{cov}(y,x)}{\mathrm{var}(x)} \;=\; \frac{\frac{1}{n-1}\sum_{j=1}^{n}(y_j - \bar{y})(x_j - \bar{x})}{\frac{1}{n-1}\sum_{j=1}^{n}(x_j - \bar{x})^2} \;=\; \frac{\langle y_j\, x_j\rangle - \langle y_j\rangle\langle x_j\rangle}{\langle x_j^2\rangle - \langle x_j\rangle^2} \;=\; \frac{\langle y_j'\, x_j'\rangle}{\langle x_j'^2\rangle} \quad , \tag{A2}$$

where $\hat{\beta}$ denotes the conventional least-squares estimator of $\beta$, primes denote deviations from means, and means over all $j$ may be denoted by either angled brackets or overbars, depending on context.

In many practical situations, the unknown constant $\beta$ may not in fact be constant, but instead may differ among the cases $j$.
In such situations, the true relationship among the variables is not Eq. (A1), but instead

$$y_j = \beta_j\, x_j + \alpha + \varepsilon_j \quad , \tag{A3}$$

where the small but important difference between (A1) and (A3) is the subscript $j$ on $\beta$. It may be unclear *a priori* whether $\beta$ is a constant or not, and therefore whether (A1) or (A3) applies. In other words, (A1) represents a special case of the more general relationship represented by (A3), and it may be unclear whether we are dealing with the special case or the general
one.

Thus, in environmental work, regression equations are often used to estimate "constants" that are not known to be constant, or, even more pointedly, "constants" that we _know are not_ constant. Regression equations are nonetheless used, under the assumption that the result will provide a useful estimate of some central tendency of the non-constant "constant". The basis
for this assumption and its range of validity are unclear.

The problem at hand can be stated like this: if the unknown coefficient $\beta_j$ differs among the cases $j$, as in (A3), but one nonetheless calculates a conventional least squares estimator $\hat{\beta}$ using (A2), how will the calculated value of $\hat{\beta}$ depend on the



properties of the (unknown) $\beta_j$, including their possible relationships with the values $x_j$ of the explanatory variable? The answer can be obtained straightforwardly by substituting $y_j$ from (A3) into (A2) and solving for $\hat{\beta}$. The math is streamlined somewhat if one separates $x, y, \beta,$ and $\varepsilon$ into their (sample) means and deviations (replacing $x_j$ with $\bar{x} + x_j'$, and similarly for $y, \beta,$ and $\varepsilon$), where primed quantities indicate deviations from means. One can begin by expressing (A3) in terms of

5    deviations from means,

$$y_j' = y_j - \bar{y} = \beta_j\, x_j - \langle\beta_j\, x_j\rangle + (\varepsilon_j - \bar{\varepsilon}) = (\bar{\beta} + \beta_j')(\bar{x} + x_j') - \langle(\bar{\beta} + \beta_j')(\bar{x} + x_j')\rangle + \varepsilon_j' \quad, \tag{A4}$$

and then by multiplying the terms in parentheses, yielding

$$y_j' = \underline{\bar{\beta}\,\bar{x}} + \bar{\beta}\, x_j' + \bar{x}\, \beta_j' + \beta_j'\, x_j' - \underline{\langle\bar{\beta}\,\bar{x}\rangle} - \underline{\underline{\langle\bar{\beta}\, x_j'\rangle}} - \underline{\underline{\langle\bar{x}\, \beta_j'\rangle}} - \langle\beta_j'\, x_j'\rangle + \varepsilon_j' \quad, \tag{A5}$$

The single-underlined terms in (A5) cancel each other, and the double-underlined terms are zero because primed quantities

10    will always average to zero (although products of two or more primed quantities usually will not). Removing all underlined terms, multiplying by $x_j' = x_j - \bar{x}$, averaging over all $j$, and dividing by the variance of $x$ yields directly:

$$\hat{\beta} = \frac{\mathrm{cov}(y,x)}{\mathrm{var}(x)} = \frac{\langle y_j'\, x_j'\rangle}{\langle x_j'\, x_j'\rangle} = \frac{\bar{\beta}\,\langle x_j'\, x_j'\rangle + \bar{x}\,\langle\beta_j'\, x_j'\rangle + \langle\beta_j'\, x_j'\, x_j'\rangle + \underline{\underline{\langle\langle\beta_j'\, x_j'\rangle\, x_j'\rangle}} + \langle\varepsilon_j'\, x_j'\rangle}{\langle x_j'\, x_j'\rangle} \quad. \tag{A6}$$

The double-underlined term in the numerator of (A6) is zero, because the inner average is a constant and therefore just re-scales $x_j'$, which in turn averages to zero. Simplifying the remaining terms, one obtains

$$\hat{\beta} = \bar{\beta} + \bar{x}\,\frac{\mathrm{cov}(\beta,x)}{\mathrm{var}(x)} + \frac{\langle\beta_j'\, x_j'^2\rangle}{\mathrm{var}(x)} + \frac{\mathrm{cov}(\varepsilon,x)}{\mathrm{var}(x)} \quad. \tag{A7}$$

Equation (A7) cannot be evaluated in practice, of course, because the true coefficients $\beta_j$ and the errors $\varepsilon_j$ will not be known. Nonetheless it can be useful to understand how their properties influence $\hat{\beta}$, so that regression results can be properly interpreted. In this regard, each of the four terms of (A7) has a story to tell. The first term of (A7) says that the linear regression coefficient $\hat{\beta}$ will be a good approximation to the (sample) mean of the $\beta_j$, if the other terms are negligible.

The second term says that the linear regression coefficient $\hat{\beta}$ can also be affected by correlations between $\beta_j$ and $x_j$. The magnitude of this effect will be the average value of $x$, multiplied by the regression slope of the relationship between $x$ and $\beta$. This second term will vanish if $\bar{x}$ is zero or if there is no correlation between $x_j$ and $\beta_j$.

25    The third term can be viewed as a weighted average of the deviations of the $\beta_j$ from their mean, where the weighting factors are the squared deviations of the $x_j$ from their mean (in statistical terms, these weighting factors are called leverages). Thus the third term of (A7) expresses the effect of a cup-shaped relationship between $\beta_j$ and $x_j$; for example, if $x_j$ values with greater leverage on $\hat{\beta}$ (because they lie farther from $\bar{x}$) are also associated with higher values of $\beta_j$ (and thus a steeper



relationship between $x_j$ and $y_j$), the estimate of $\hat{\beta}$ will be biased upward. Note in particular that the third term could be non-zero even if the correlation between $\beta_j$ and $x_j$ is zero (that is, the relationship between $\beta_j$ and $x_j$ could be cup-shaped even if it has a slope of zero overall). Conversely, the third term is insensitive to linear correlations (even strong ones) between $\beta_j$ and $x_j$.

The fourth term says that $\hat{\beta}$ could also be biased by correlations between the error term and the explanatory variable; the magnitude of this possible bias equals the regression slope of $\varepsilon_j$ as a function of $x_j$. This is the well-known problem of artifactual correlation (also called the "third variable problem" or "hidden variable problem"): if hidden (unmeasured) variables are correlated with the measured explanatory ($x$) variable, their effects on the response ($y$) variable will be falsely

attributed to the $x$ variable instead, distorting its regression coefficient.

It should be noted that the means, variances, and covariances in (A2)-(A7) are sample statistics calculated over the sample cases $j$, which may differ from the true means, variances, and covariances of the underlying distributions. Thus there will be additional uncertainty resulting from sampling variability (in addition to the biases quantified by the second, third, and fourth

terms in Eq. A7), if one interprets the regression slope as an estimate of the true mean of $\beta$ rather than the sample mean of the $\beta_j$ for the particular cases $j$ that have been sampled.

To illustrate the analysis outlined above, I conducted a simple numerical experiment based on ensemble hydrograph separation. I created a synthetic data set based on the mixing equation

$$C_{Q_j} = F_{\text{new}_j} C_{\text{new}_j} + \left(1 - F_{\text{new}_j}\right) - C_{Q_{j-1}} \quad , \tag{A8}$$

where $C_{Q_j}$, the concentration in the stream, is a volume-weighted average of the (measured) new-water concentration $C_{\text{new}_j}$ and the old-water concentration $C_{Q_{j-1}}$ from the previous time step, weighted by the new water fraction $F_{\text{new}_j}$ and its complement $1 - F_{\text{new}_j}$. Values of $F_{\text{new}_j}$ for each time step $j$ are randomly chosen from a beta distribution,

$$\text{Beta}(\alpha, \beta) = \frac{x^{\alpha-1} (1-x)^{\beta-1}}{\text{B}(\alpha, \beta)} \quad , \tag{A9}$$

where $x$ is a random variable that, appropriately for a fraction, ranges from 0 to 1, the beta function $\text{B}(\alpha, \beta) = \Gamma(\alpha)\Gamma(\beta)/\Gamma(\alpha + \beta)$ is a normalization constant that ensures that the cumulative probability is 1, and $\alpha$ and $\beta$ are shape parameters that are related to the mean ($\mu$) by $\mu = 1/(1 + \beta/\alpha)$, or equivalently $\beta = \alpha[(1-\mu)/\mu]$. In the simulations shown here (Fig. A1a-e), the $\alpha$ parameter is fixed at 1.



Values of $C_{\mathrm{new}_j}$ for each point in time $j$ are randomly chosen from a normal distribution with a standard deviation of 10 (Fig. A1b). Values of $C_{\mathrm{Q}_j}$ are calculated for the whole time series using Eq. (A8), and measurement errors (normally distributed, with a standard deviation of 1) are added to both $C_{\mathrm{new}}$ and $C_{\mathrm{Q}}$. Then an ensemble estimate of the average $F_{\mathrm{new}}$ is obtained by linear regression of $y_j = C_{\mathrm{Q}_j} - C_{\mathrm{Q}_{j-1}}$ on $x_j = C_{\mathrm{new}_j} - C_{\mathrm{Q}_{j-1}}$, following Eq. (9) in the main text. A plot of such a

regression is shown in Fig. A1d. In this particular ensemble, the individual $F_{\mathrm{new}_j}$ values for each time step varied between 0.0001 and 0.71, with a mean of 0.20 and a standard deviation of 0.16. The ensemble hydrograph separation estimate of the average $F_{\mathrm{new}}$ was 0.205±0.009 (mean ± standard error), deviating from the true mean value by roughly its standard error, as one would expect. This analysis was repeated 1000 times for mean $F_{\mathrm{new}}$ values randomly chosen between 0.025 and 0.975. The results are summarized in Fig. A1e, which compares the regression estimates of the average $F_{\mathrm{new}}$ against the true means

of the $F_{\mathrm{new}}$ values in each sample. Although the individual $F_{\mathrm{new}_j}$ values that make up each mean vary widely (as indicated by the horizontal width of the shading in Fig. A1e), the regression estimates of the average $F_{\mathrm{new}}$ cluster tightly around the 1:1 line, with a root-mean-square deviation of less than 0.02 across the full range of average $F_{\mathrm{new}}$ (this root-mean-square deviation scales, as one would expect, inversely with the square root of the number of data points in the simulated time series).

In the simulations shown in Fig. A1, $F_{\mathrm{new}_j}$ is independent of $C_{\mathrm{new}_j}$, $C_{\mathrm{Q}_{j-1}}$, and the measurement errors; therefore the biases quantified in Eq. A7 are expected to be small. Nonetheless, one should be aware that in the specific case of Eq. A8 there could be two additional sources of bias that Eq. A7 does not account for. Large measurement errors in $C_{\mathrm{new}}$ (meaning measurement errors that are not small compared to the standard deviation of $C_{\mathrm{new}}$ itself) could potentially create negative

biases in estimates of the average $F_{\mathrm{new}}$, because they would add spurious variation to the $x$ axis of regressions like Fig. A1d. Conversely, large measurement errors in $C_{\mathrm{Q}}$ – which again means errors that are not small compared to the standard deviation of $C_{\mathrm{new}}$ (not $C_{\mathrm{Q}}$) – could potentially create positive biases in estimates of the average $F_{\mathrm{new}}$, because $C_{\mathrm{Q}_{j-1}}$ appears on both axes of the regression in Fig. A1d, so large errors in $C_{\mathrm{Q}_{j-1}}$ would spuriously increase the correlation between the $x$ and $y$ axes of regressions like Fig. A1d. Both of these biases should be negligible in real-world cases, however, because the

measurement uncertainties in $C_{\mathrm{new}}$ and $C_{\mathrm{Q}}$ are typically much smaller than the variability in $C_{\mathrm{new}}$.



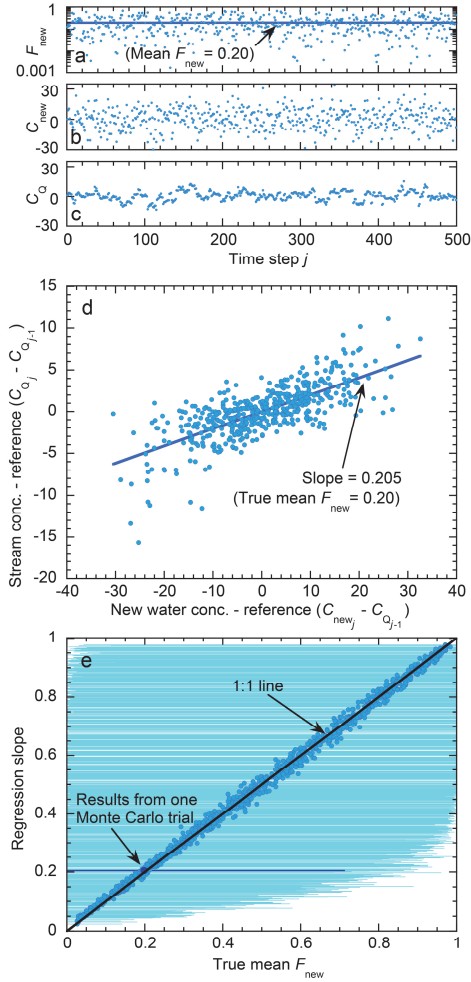

**Figure A1.** Benchmark test of regression estimates of mean new water fractions, using data from a simple two-component mixing model. In that mixing model (Eq. A8), a randomly varying new water fraction $F_{new}$ (a) determines the relative proportions of new and old water ($C_{new_j}$ and $C_{Q_{j-1}}$, respectively) which are combined to yield a mixture with concentration $C_{Q_j}$ (c). Among the 500-point time series shown in panels (a)-(c), the new water fraction $F_{new}$ varies between 0.0001 and 0.71, with a mean of 0.20 and a standard deviation of 0.15. Plotting the concentration of the mixture in the stream as a function of the concentration in the new-water end-member (e) yields a regression slope of 0.205±0.009, which agrees within error with the true average of $F_{new}$ of $\mu$=0.20. Repeating this analysis 1000 times, with mean values of $F_{new}$ ranging from nearly zero to nearly one, yields regression slopes that agree with the means of the $F_{new}$ for each Monte Carlo trial with an RMS error of only 0.02 (panel e). In panel (e), the circles show the regression slopes and mean $F_{new}$, and the horizontal light blue lines show the range of $F_{new}$, for each Monte Carlo trial. The dark circle and dark line show the results for the individual Monte Carlo trial shown in panels (a)-(d).





**Appendix B: Accounting for rain-free periods, and estimating non-constant "constants" by multiple regression**

Assume a multiple linear regression equation with non-constant unknown coefficients,

$$y_j = \sum_{k=0}^{m} \beta_{j,k}\, x_{j,k} + \alpha + \varepsilon_j \quad , \tag{B1}$$

which can be more explicitly represented, for a series of sampling times $j = 1 \ldots n$ as

$$
\begin{aligned}
y_1 &= \beta_{1,0}\, x_{1,0} + \beta_{1,1}\, x_{1,1} + \beta_{1,2}\, x_{1,2} + \beta_{1,3}\, x_{1,3} \ldots + \beta_{1,m}\, x_{1,m} + \alpha + \varepsilon_1 \\
y_2 &= \beta_{2,0}\, x_{2,0} + \beta_{2,1}\, x_{2,1} + \beta_{2,2}\, x_{2,2} + \beta_{2,3}\, x_{2,3} \ldots + \beta_{2,m}\, x_{2,m} + \alpha + \varepsilon_2 \\
y_3 &= \beta_{3,0}\, x_{3,0} + \beta_{3,1}\, x_{3,1} + \beta_{3,2}\, x_{3,2} + \beta_{3,3}\, x_{3,3} \ldots + \beta_{3,m}\, x_{3,m} + \alpha + \varepsilon_3 \\
y_4 &= \beta_{4,0}\, x_{4,0} + \beta_{4,1}\, x_{4,1} + \beta_{4,2}\, x_{4,2} + \beta_{4,3}\, x_{4,3} \ldots + \beta_{4,m}\, x_{4,m} + \alpha + \varepsilon_4 \\
&\quad\quad\quad\quad\quad\quad\quad\quad\quad\ddots \\
y_n &= \beta_{n,0}\, x_{n,0} + \beta_{n,1}\, x_{n,1} + \beta_{n,2}\, x_{n,2} + \beta_{n,3}\, x_{n,3} \ldots + \beta_{n,m}\, x_{n,m} + \alpha + \varepsilon_n \quad .
\end{aligned} \tag{B2}
$$

For simplicity, and without loss of generality, assume that the $y_j$'s and $x_{j,k}$'s have means of zero. Assume further that for each $k$, the coefficients $\beta_{j,k}$ either have a constant value of $\beta_k$ (when precipitation is present at time step $i = j - k$), or have a value of 0 (when precipitation in missing at time step $i = j - k$). In the latter case the value of $x_{j,k}$ will be undefined, but it will also be irrelevant because it is multiplied by zero. The resulting system of equations will then have the following form, with missing values along diagonal stripes (this illustration shows just one possible set of missing values):

$$
\begin{aligned}
y_1 &= \beta_0\, x_{1,0} + \quad\quad\quad + \beta_2\, x_{1,2} + \beta_3\, x_{1,3} \ldots + \beta_m\, x_{1,m} + \alpha + \varepsilon_1 \\
y_2 &= \quad\quad\quad + \beta_1\, x_{2,1} + \quad\quad + \beta_3\, x_{2,3} \ldots + \beta_m\, x_{2,m} + \alpha + \varepsilon_2 \\
y_3 &= \beta_0\, x_{3,0} + \quad\quad\quad + \beta_2\, x_{3,2} + \quad\quad \ldots + \beta_m\, x_{3,m} + \alpha + \varepsilon_3 \\
y_4 &= \beta_0\, x_{4,0} + \beta_1\, x_{4,1} + \quad\quad + \beta_3\, x_{4,3} \ldots + \quad\quad + \alpha + \varepsilon_4 \\
&\quad\quad\quad\quad\quad\quad\quad\quad\quad\ddots \\
y_n &= \beta_0\, x_{n,0} + \beta_1\, x_{n,1} + \beta_2\, x_{n,2} + \quad\quad \ldots + \beta_m\, x_{n,m} + \alpha + \varepsilon_n \quad .
\end{aligned} \tag{B3}
$$

Multiplying the left and right sides of Eq. (B3) by the transpose of $\boldsymbol{X}_0$ (to take one example) yields

$$
\begin{aligned}
\sum_{j=1}^{n} \left( y_j\, x_{j,0} \right)_{P_{j-k}>0} &= \beta_0 \sum_{j=1}^{n} \left( x_{j,0}\, x_{j,0} \right)_{P_{j-0}>0} + \beta_1 \sum_{j=1}^{n} \left( x_{j,1}\, x_{j,0} \right)_{P_{j-1}>0,\, P_{j-0}>0} \\
&+ \beta_2 \sum_{j=1}^{n} \left( x_{j,2}\, x_{j,0} \right)_{P_{j-2}>0,\, P_{j-0}>0} + \ldots + \beta_m \sum_{j=1}^{n} \left( x_{j,m}\, x_{j,0} \right)_{P_{j-m}>0,\, P_{j-0}>0} \quad ,
\end{aligned} \tag{B4}
$$

where the constant $\alpha$ and the error term $\varepsilon_j$ drop out because their sums of cross-products with the $x_{j,0}$ are zero. One can see that each of the summations of cross-products equals the covariance of the respective vectors, multiplied by the number of points in the summation. In contrast to a typical multiple regression, these numbers of points are not the same. For the left-hand side of Eq. (B4), the summation is taken over the non-missing members of $\boldsymbol{X}_0$; if we use $n_{x_0}$ to express the number of such members, this summation equals $n_{x_0} \mathrm{cov}(\boldsymbol{X}_0, \boldsymbol{Y})$. The first term on the right hand side can also be evaluated for all $n_{x_0}$ non-missing members of $\boldsymbol{X}_0$; there are $n_{x_0}$ of these, so this term becomes $\beta_0\, n_{x_0} \mathrm{cov}(\boldsymbol{X}_0, \boldsymbol{X}_0) = \beta_0\, n_{x_0} \mathrm{var}(\boldsymbol{X}_0)$. The



second term on the left-hand side, on the other hand, can only be evaluated when both $X_0$ and $X_1$ are non-missing. If we denote the number of such cases as $n_{x_0 x_1}$, the second term equals $\beta_1 \, n_{x_0 x_1} \, \mathrm{cov}(X_0, X_1)$, the third term equals $\beta_2 \, n_{x_0 x_2} \, \mathrm{cov}(X_0, X_2)$, and so forth. Thus when re-expressed as covariances, Eq. (B3) becomes

$$n_{x_0} \mathrm{cov}(X_0, Y) = \beta_0 \; n_{x_0} \; \mathrm{cov}(X_0, X_0) + \beta_1 \, n_{x_0 x_1} \mathrm{cov}(X_0, X_1) + \beta_2 \, n_{x_0 x_2} \mathrm{cov}(X_0, X_2) + \, \ldots \, \beta_m \, n_{x_0 x_m} \mathrm{cov}(X_0, X_m)$$
$$n_{x_1} \mathrm{cov}(X_1, Y) = \beta_0 \, n_{x_1 x_0} \mathrm{cov}(X_1, X_0) + \beta_1 \; n_{x_1} \; \mathrm{cov}(X_1, X_1) + \beta_2 \, n_{x_1 x_2} \mathrm{cov}(X_1, X_2) + \, \ldots \, \beta_m \, n_{x_1 x_m} \mathrm{cov}(X_1, X_m)$$
$$n_{x_2} \mathrm{cov}(X_2, Y) = \beta_0 \, n_{x_2 x_0} \mathrm{cov}(X_2, X_0) + \beta_1 \, n_{x_2 x_1} \mathrm{cov}(X_2, X_1) + \beta_2 \; n_{x_2} \; \mathrm{cov}(X_2, X_2) + \, \ldots \, \beta_m \, n_{x_2 x_m} \mathrm{cov}(X_2, X_m)$$
$$\ddots$$
$$n_{x_m} \mathrm{cov}(X_m, Y) = \beta_0 \, n_{x_m x_0} \mathrm{cov}(X_m, X_0) + \beta_1 \, n_{x_m x_1} \mathrm{cov}(X_m, X_1) + \beta_2 \, n_{x_m x_2} \mathrm{cov}(X_m, X_2) + \, \ldots \, \beta_m \, n_{x_m} \mathrm{cov}(X_m, X_m) \quad . \quad \text{(B5)}$$

Dividing through by the $n_{x_k}$ terms on the left-hand side, one directly obtains the following system of $m$ equations in $m$ unknowns,

$$\mathrm{cov}(X_0, Y) = \beta_0 \mathrm{cov}(X_0, X_0) + \beta_1 \frac{n_{x_0 x_1}}{n_{x_0}} \mathrm{cov}(X_0, X_1) + \beta_2 \frac{n_{x_0 x_2}}{n_{x_0}} \mathrm{cov}(X_0, X_2) + \, \ldots \, \beta_m \frac{n_{x_0 x_m}}{n_{x_0}} \mathrm{cov}(X_0, X_m)$$
$$\mathrm{cov}(X_1, Y) = \beta_0 \frac{n_{x_1 x_0}}{n_{x_1}} \mathrm{cov}(X_1, X_0) + \beta_1 \mathrm{cov}(X_1, X_1) + \beta_2 \frac{n_{x_1 x_2}}{n_{x_1}} \mathrm{cov}(X_1, X_2) + \, \ldots \, \beta_m \frac{n_{x_1 x_m}}{n_{x_1}} \mathrm{cov}(X_1, X_m)$$
$$\mathrm{cov}(X_2, Y) = \beta_0 \frac{n_{x_2 x_0}}{n_{x_2}} \mathrm{cov}(X_2, X_0) + \beta_1 \frac{n_{x_2 x_1}}{n_{x_2}} \mathrm{cov}(X_2, X_1) + \beta_2 \mathrm{cov}(X_2, X_2) + \, \ldots \, \beta_m \frac{n_{x_2 x_m}}{n_{x_2}} \mathrm{cov}(X_2, X_m)$$
$$\ddots$$
$$\mathrm{cov}(X_m, Y) = \beta_0 \frac{n_{x_m x_0}}{n_{x_m}} \mathrm{cov}(X_m, X_0) + \beta_1 \frac{n_{x_m x_1}}{n_{x_m}} \mathrm{cov}(X_m, X_1) + \beta_2 \frac{n_{x_m x_2}}{n_{x_m}} \mathrm{cov}(X_m, X_2) + \, \ldots \, \beta_m \mathrm{cov}(X_m, X_m) \quad , \quad \text{(B6)}$$

which can be solved by the usual matrix inversion approach, yielding:

$$\begin{pmatrix} \hat\beta_0 \\ \hat\beta_1 \\ \hat\beta_2 \\ \vdots \\ \hat\beta_m \end{pmatrix} = \begin{pmatrix} \mathrm{cov}(X_0, X_0) & \frac{n_{x_0 x_1}}{n_{x_0}} \mathrm{cov}(X_0, X_1) & \frac{n_{x_0 x_2}}{n_{x_0}} \mathrm{cov}(X_0, X_2) & \cdots & \frac{n_{x_0 x_m}}{n_{x_0}} \mathrm{cov}(X_0, X_m) \\ \frac{n_{x_1 x_0}}{n_{x_1}} \mathrm{cov}(X_1, X_0) & \mathrm{cov}(X_1, X_1) & \frac{n_{x_1 x_2}}{n_{x_1}} \mathrm{cov}(X_1, X_2) & \cdots & \frac{n_{x_1 x_m}}{n_{x_1}} \mathrm{cov}(X_1, X_m) \\ \frac{n_{x_2 x_0}}{n_{x_2}} \mathrm{cov}(X_2, X_0) & \frac{n_{x_2 x_1}}{n_{x_2}} \mathrm{cov}(X_2, X_1) & \mathrm{cov}(X_2, X_2) & \cdots & \frac{n_{x_2 x_m}}{n_{x_2}} \mathrm{cov}(X_2, X_m) \\ \vdots & \vdots & \vdots & \ddots & \vdots \\ \frac{n_{x_m x_0}}{n_{x_m}} \mathrm{cov}(X_m, X_0) & \frac{n_{x_m x_1}}{n_{x_m}} \mathrm{cov}(X_m, X_1) & \frac{n_{x_m x_2}}{n_{x_m}} \mathrm{cov}(X_m, X_2) & \cdots & \mathrm{cov}(X_m, X_m) \end{pmatrix}^{-1} \begin{pmatrix} \mathrm{cov}(X_0, Y) \\ \mathrm{cov}(X_1, Y) \\ \mathrm{cov}(X_2, Y) \\ \vdots \\ \mathrm{cov}(X_m, Y) \end{pmatrix} .$$

(B7)

One can see that Eq. (B7) is identical in form to Eq. (38), with the addition of weighting factors on the off-diagonal elements of the covariance matrix. One consequence of these leading terms is that the weighted covariance matrix will usually not be completely symmetrical, because (for example) $n_{x_2 x_0}/n_{x_2}$ will often differ from $n_{x_2 x_0}/n_{x_0}$.

It bears emphasis that Eq. (B7) accounts for gaps in precipitation, but not for precipitation or streamflow samples that are
missing due to sampling and measurement failures. A gap in precipitation means that the corresponding tracer values never existed at all, and had no effect on streamflow, whereas tracer values that are missing due to sampling and measurement failures actually did affect streamflow, but are unknown. Equation (B7) accounts for the fact that the tracer covariances will necessarily be less strongly coupled to one another, the less frequently precipitation falls. Glasser's method, by contrast, estimates the covariances themselves from all available pairs of observations, but says nothing about how they are related to





one another. Therefore we can account for both kinds of missing data using Eq. (B7), with the covariances between pairs of variables estimated using Glasser's method (Eqs. 40-41). That approach results in Eq. (42).

Astute readers may notice that Eq. (B3) is equivalent to the normal equations of conventional multiple regression, with the
cases of missing precipitation replaced by $x_{j,k} = 0$ (instead of $\beta_{j,k} = 0$). This provides a simple procedure for estimating the $\beta_k$ if tracer values are only missing due to lack of precipitation, with no sampling or measurement failures. This method proceeds as follows:

1. Normalize $\mathbf{Y}$ and each of the $\mathbf{X}_k$ to zero by subtracting the mean from each vector (excluding any missing values from these means).
2. Replace any values that are missing due to lack of precipitation with zeroes.

3. Solve for the $\beta_k$ using conventional multiple regression.

4. Multiply the standard errors of the $\beta_k$ (not the $\beta_k$ themselves!) by $\sqrt{n/n_{(k)}}$ to account for the fact that the zeroes that have been used to in-fill the missing values are not measured values, and thus do not contribute information to constrain the $\beta_k$.

This method is unlikely to be useful in most practical cases, in which occasional sampling and measurement failures are virtually guaranteed. However, it can provide a useful consistency check for implementations of the more complex approach developed here (Eqs. B7 and 42).

There remains one last important detail. In transitioning from Eq. (B2) to Eq. (B3), I made the simplifying assumption that
all of the coefficients $\beta_{j,k}$ for a given $k$ were either equal to zero or had a constant value of $\beta_k$ instead. The same assumption is made in the derivation presented in Sect. 4. One could naturally ask what happens if the $\beta_{j,k}$ vary individually across the time steps $j$. Is there is a (nearly) equivalent constant $\beta_k$, and if so, how does it relate to the values of the $\beta_{j,k}$?

If we have a variable $\beta_{j,k}$ rather than a constant $\beta_k$, each of the terms of Eq. (B4) will be of the form $\sum \beta_{j,k} \, x_{j,k} \, x_{j,\ell}$ instead of
$\beta_k \sum x_{j,k} \, x_{j,\ell}$, and each of the terms of Eq. (B5) will be of the form $n_{x_k x_\ell} \, \mathrm{cov}\big(\beta_{j,k} x_{j,k} \, , x_{j,\ell}\big)$ instead of $\beta_k \, n_{x_k x_\ell} \, \mathrm{cov}\big(x_{j,k} \, , x_{j,\ell}\big)$. Thus the effect of a variable vs. constant $\beta$ depends on how $\mathrm{cov}\big(\beta_{j,k} x_{j,k} \, , x_{j,\ell}\big)$ differs from $\beta_k \, \mathrm{cov}\big(x_{j,k} \, , x_{j,\ell}\big)$. Following the approach in Appendix A, I begin by expanding the three variables into their means and deviations, replacing $\beta_{j,k}$ with $\bar{\beta}_k + \beta'_{j,k}$, $x_{j,k}$ with $\bar{x}_k + x'_{j,k}$, and $x_{j,\ell}$ with $\bar{x}_k + x'_{j,\ell}$. Each covariance on the right-hand side of Eq. (B5) would thus become instead

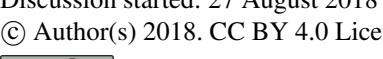


$$\text{cov}(\beta_{j,k} x_{j,k}, x_{j,\ell}) = \langle\left((\bar{\beta}_k + \beta'_{j,k})(\bar{x}_k + x'_{j,k}) - \overline{(\bar{\beta}_k + \beta'_{j,k})(\bar{x}_k + x'_{j,k})}\right)\left((\bar{x}_k + x'_{j,\ell}) - \overline{(\bar{x}_k + x'_{j,\ell})}\right)\rangle$$

$$= \langle\left((\bar{\beta}_k + \beta'_{j,k})(\bar{x}_k + x'_{j,k}) - \bar{\beta}_k \bar{x}_k - \overline{\beta'_{j,k} x'_{j,k}}\right)\left((\bar{x}_k + x'_{j,\ell}) - \bar{x}_k\right)\rangle$$

$$= \langle\left(\bar{\beta}_k x'_{j,k} + \bar{x}_k \beta'_{j,k} + \beta'_{j,k} x'_{j,k} - \overline{\beta'_{j,k} x'_{j,k}}\right)(x'_{j,\ell})\rangle$$

$$= \langle\bar{\beta}_k x'_{j,k} x'_{j,\ell} + \bar{x}_k \beta'_{j,k} x'_{j,\ell} + \beta'_{j,k} x'_{j,k} x'_{j,\ell} - \overline{\beta'_{j,k} x'_{j,k}} x'_{j,\ell}\rangle$$

$$= \bar{\beta}_k \langle x'_{j,k} x'_{j,\ell}\rangle + \bar{x}_k \langle\beta'_{j,k} x'_{j,\ell}\rangle + \langle\beta'_{j,k} x'_{j,k} x'_{j,\ell}\rangle \qquad , \tag{B8}$$

where angled brackets and overbars indicate averages over $j$. The final result can thus be written as

$$\text{cov}(\beta_{j,k} x_{j,k}, x_{j,\ell}) = \bar{\beta}_k \text{cov}(x_{j,k}, x_{j,\ell}) + \bar{x}_k \text{cov}(\beta_{j,k}, x_{j,\ell}) + \langle\beta'_{j,k} x'_{j,k} x'_{j,\ell}\rangle \qquad , \tag{B9}$$

where the first term on the right-hand side expresses the approximation on which the covariance matrices in (B7) and (42)

are based; if the second and third terms vanish, then this approximation is exact. The second term on the right-hand side

should be small, unless there is a strong correlation between $\beta_{j,k}$ and $x_{j,\ell}$ (which is unlikely unless storm size is correlated

with tracer concentrations, as explained in Sect. 2.1), _and_ $\bar{x}_k$ is large (which is unlikely because $\bar{x}_k = \langle C_{P_{j-k}}\rangle - \langle C_{Q_{j-m-1}}\rangle$

(see Eq. 35), and mass conservation implies that the averages of $C_P$ and $C_Q$ should be similar). The third term on the right-

hand side is a three-way cross-product, technically termed a co-skewness, that bears the same relation to skewness that

covariance does to variance. It has the interesting property that its expected value is zero if the three variables have

symmetrical distributions, even if they are strongly correlated (either positively or negatively, in any combination) with one

another. This behavior arises because the odd number of terms means that, for symmetrical distributions, the product

$\beta'_{j,k} x'_{j,k} x'_{j,\ell}$ is equally likely to be positive or negative for any $j$, and thus the positive and negative values of $\beta'_{j,k} x'_{j,k} x'_{j,\ell}$ will

tend to average out when one averages over all $j$. If the last two terms of Eq. (B9) are small compared to the first one, Eq.

(B9) says that the covariance matrices in (B5)-(B6) will be nearly the same whether $\beta$ is constant or variable, whenever the

constant $\beta_k$ is the average of the variable $\beta_{j,k}$. This in turn implies that the analysis presented in Sect. 4 should result in

estimated coefficients $\hat{\beta}_k$ that closely approximate the average of the time-varying $\beta_{j,k}$, as is confirmed by the benchmark

tests of Sects. 4.6-4.8.




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
