# Peer review of "Quantifying new water fractions and transit time distributions using ensemble hydrograph separation: theory and benchmark tests"

_Hydrology and Earth System Sciences, 2018_

## Referee Comment (RC1) · Anonymous Referee #1 · 4 Oct 2018

Overview:

This article presents a new methodology called ensemble hydrograph separation. The method is distinguished from traditional hydrograph separation (i.e. 2-end-member-mixing model) in multiple ways, including (1) it utilizes differences between conservative-tracer concentrations in end members and a designated "old water" source, rather than the concentrations themselves; (2) the method recasts the end-member-mixing model based on mass balance into a linear-regression equation, which alleviates the necessary assumption of conservation of mass in the mixed flow; and (3) the method generates ensemble-average estimates of "new" water contributions to

streamflow across extend time periods (e.g. months, seasons, and years), rather than estimates at the time scale of single storm events. The author goes on to show how discrete integral equations for conservation of tracer mass between precipitation and streamflow can similarly be recast in the form on a multiple-linear regression model. The slope coefficient in this model is shown to equate to the fractional contribution of total streamflow at a moment in time that is composed of water volumes with specific residence times. When integrated across all plausible residence times, and averaged over all relevant times, the slope coefficient allows an estimation of the ensemble-average transit-time distribution of water within streamflow (possible "forward" or "backward" TTDs, as the author shows). The author also shows how one might approximate the time variance of TTDs under different hydroclimatic conditions by applying the method to different subsets of an entire data series.

The author develops a two-store conceptual model of streamflow and conservative-solute transport for a catchment, then uses the model under various climatic forcing and parameterizations to generate benchmark ensemble TTDs and new-water estimates. The regression approaches noted above are tested to see if they can accurately recreate these independent, benchmark data sets, which they do quite well. The author concludes the article by clearly delineating how the method differs from traditional hydrograph separation and lumped-parameter-transport modeling based on time-invariant TTDs and Storage-Selection Distributions (a.k.a. SAS functions). He discusses possible interesting applications of the method.

Recommendation to the Editor:

I think this article could have a strong impact in the field of catchment hydrology. I believe it can be published with very minor revision.

The writing and visual presentation of material are excellent. I found only one certain typo (see Technical Edit #5), which is remarkable for an article of this length, and greatly appreciated. I think the most exciting possible impact of this method could be retro-

spective application to existing data sets from myriad catchments. We have more than three decades of studies showing that "old" water is generally the majority component of the stream hydrograph, even during individual storm events. But because the rigid assumptions of two-component-mixing models are always violated in these studies, and since most of them did not report uncertainty, actual numeric estimates are dubious (as is any comparison among them). Similarly, there are nearly three decades of studies that employed lumped-parameter-transport modeling based on time-invariant TTDs to estimate mean-transit times (MTTs) of water flow through catchments in many locations. The assumption of the time-invariant TTD was always known to be a bad one (even if the analysis only considered tracer concentrations during baseflow), and is now unnecessary [G. Botter et al., 2010; Gianluca Botter et al., 2011; Harman, 2015; van der Velde et al., 2012]. As such, there is reasonable suspicion about just how accurate are all of those existing estimates of MTT. Further—and likely as a tacit acknowledgement of this poor assumption—most past studies focused on the MTT, but not the actual form of the parameterized TTD. As this author notes, the method proposed here is fairly robust in that it can be employed with tracer data sets collected at variable time intervals, and when significant parts of the time series may be missing. The method can take advantage of all these existing data sets to possibly refine our view of what MTTs are for low-order catchments in different regions, and what the form of the TTDs actually look like (at least the early-time portions of TTDs; the long tails remain difficult to constrain). I would be excited to see the results of this method being applied to all of these existing data sets, and from newer networks of catchment studies like the Critical Zone Observatories.

As one point of criticism, I would ask the author to consider, and possibly elaborate in the discussion, about what might be the advantage of this method over the contemporary methods of lumped-parameter-transport modeling based on time-variable TTDs and SAS distributions/functions? For those researchers organizing new field studies and networks of sites, and that are reasonably competent and diligent with field-data collection (i.e. avoiding lots of data gaps), is there any reason to consider this method?

[Figure]

One may argue that it is not any simpler to implement, and cannot achieve similar temporal resolution when estimating time-variable TTDs and SAS distributions. As discussed by the author, application to short subsets of data (for comparison of TTDs at different times) reduces sample size and enhances uncertainty in the regression approach.

Last, the author makes one conclusion that I would disagree with, and that I believe might negatively influence subsequent applications of this method to field data (see specific comments 4 and 10). I ask the author to consider these comments and possibly revise their remarks. I also thought there were a few instances where the citation of other relevant works could be improved (see specific comments 9, 12, and 13).

Specific Comments:

(1) Page 5; line 24: Another complication would seem to be the fact that CQ(j) will be strongly temporally correlated with CQ(j-1). Most perennial streams will be hydraulically connected to a riparian aquifer, and receiving discharge from that aquifer so long as the stream is gaining. The water volume in that aquifer may be 1 or 2 orders of magnitude greater than the volume of water associated with typical precipitation events. CQ(j-1) (i.e., old water) and CQ(j) should then be strongly temporally correlated. In other cases Cnew(j) might also be strongly temporally correlated with CQ(j), for example, in catchments with widespread impervious surfaces that induce infiltration-excess-overland flow. Whether or not the latter type of temporal correlation would exist would also depend on the temporal resolution of sampling. Is this potential for temporal correlation within the explanatory and dependent variables problematic for the linear regression approach here?

(2) Page 6, line 3-5: What would qualify as small? The condition in line 5 would not necessarily be true would it? In the case of stable-isotopes as tracers, if 1) rainfall is distributed uniformly throughout the year, 2) there is seasonal variation in the mean Cnew(j), and 3) streamflow is generated preferentially by precipitation that falls in the

cold season, then this difference would not be expected to be zero, would it? These conditions are fairly common.

(3) Page 13, lines 8-10: Could you comment on the rationale for randomly assigning $\delta^2H$ values to precipitation in this way, in light of the fact that actual $\delta^2H$ values are often correlated with cumulative rainfall amounts/unit time? I ask only because the existence of this correlation would seemingly have bearing on condition 2 on page 6. If cumulative rainfall amount during a storm is not correlated with $\delta^2H$, then in principle either very negative $\delta^2H$ values, or values near zero (a plausible upper boundary of the variable's range), could be observed for large storms that generate greater Fnew. In that case condition 2 on page 6 might be questionable. Seemingly the random assignment here in some sense violates that condition, yet the results in Figures 1 and 2 are quite good?

(4) Page 26, lines 28-30 and Page 27, lines 1-3: Worst-case scenario based on what? Estimates of actual evaporative enrichment measured in surficial soil water? Or plausible ranges from models [e.g. Allison et al., 1983; Barnes and Allison, 1983]? Certainly in semi-arid environments, or where vegetation is interspersed (e.g. tree plantations or agricultural settings) there could be more than 20 ‰ enrichment of soil water before it percolations below the root zone. The fractionation effect could vary strongly at daily and weekly time scales depending on storm frequency and intensity, and potential evaporation (from plant surfaces and surficial soil). The fractionation effect could in fact be strongly correlated with storm size—not a constant offset or random fluctuation that introduce no bias. And of course this simulation makes no consideration of spatial variability of the fractionation effect, which might occur due to substantial land-cover variability (e.g. catchments with mixed bare-soil and vegetated cover; wetland-meadow-forest transitions; partially snow-covered with spatially- and temporally-variable melt dynamics, etc.). Finally, it's worth noting that the isotope composition of infiltrating water can be different than precipitation (greater or lesser) due to evaporation from, or storage within, plant canopies [Allen et al., 2016]. These

effects can be of variable sign and magnitude during individual storms. I would argue it is these storm-specific effects that matter—not an average offset over the long term—because these storm-specific effects influence the individual values of Cnew – Cold in the regression model.

I think the concluding statement about this analysis (page 27, lines 1-3) should be more suggestive than definitive. People will read this and make the convenient assumption that a single precipitation collector is all that is needed in the field, rather than the more labor-intensive alternative of having multiple precipitation/throughfall/snowfall collectors (to capture some essence of spatial variability, and thus leading to more samples to analyze in the lab). This is an aspect of the method that should at least be examined in a few field studies in catchments that vary climatically and ecologically, rather than assumed unimportant based on this useful, but inconclusive, simulation. The same problem exists among most historical studies of mean transit times in forested catchments: the input data sets in the convolution integral are all biased to some unknown degree due to the spatial heterogeneity of storage, throughfall, and evaporation.

(5) Page 29, line 10: Could you clarify here if lag time m corresponds to the most distant past time when a measurement of CP is available, or can the value of m be chosen even among those past times when CP was measured? You seem to be implying the latter case, but I don't understand why you would potentially lump multiple measured values of CP into the Colder term, rather than using the values individually in the regression analysis. Perhaps this is clarified later on in the text.

(6) Page 40, lines 15-20: This seems related to the question I posed in specific comment 1. If that is correct, perhaps you could foreshadow for the reader at that location in the text that this issue is discussed later.

(7) Page 45, lines 15-25: Could you clarify in this case how the parameter m is determined? Seemingly if you want to isolate some subsets of the time series of Q for comparison of the average-backward TTDs, for example, Q during September versus

[Figure]

April at Smith River, the CQj and distribution of water ages at any time during either interval could be strongly influenced by CP that occurred over many historical time intervals prior to September or April.

(8) Page 47, figures 16a,b: Regarding the orange line, it's interesting and counter-intuitive to me that these average TTDs would show that volume fractions of Q with ages of only a few days would be (slightly) more probable than volume fractions of Q with ages of several days to weeks. It rains so infrequently during the summer, most often there would be little to no water at all within the catchment that had residence time of only a few days. I would think there should be a slight increasing trend in these probabilities from left to right on the transit-time axis, up to a point where the trend then turns back downward. Is this possibly because those few storms that do occur during the summer (and deliver some water to the stream with age of only a few days) represent a disproportionately large fraction of total Q over the summer months?

(9) Page 49, line 4: Certainly there are several good examples of where storage dependence has been examined, even one case in the same climate as the Smith River. These are worth acknowledging [Benettin et al., 2013; Harman, 2015; Heidbuchel et al., 2013; Heidbuchel et al., 2012; Rodriguez et al., 2018; van der Velde et al., 2010; van der Velde et al., 2012].

(10) Page 51, line 19: I would argue that point 5 here should perhaps be omitted. While suggestive, I don't think the simulation that leads to this conclusion is a very realistic representation of isotope fractionation effects over time or space. See specific comment 4 above. Also, while I'm no proponent of traditional hydrograph separation, I don't think the first sentence in point 5 is really true. The method is not necessarily vulnerable to biases in tracer measurements resulting from fractionation. If fractionation has occurred in the end member, the effect should be apparent in the measured delta values. Fractionation doesn't inhibit accurate quantification of the tracer concentration in the end member.

[Figure]

(11) Page 52, lines 7-9: Perhaps put "time-invariant" in italics here for emphasis, since the focus of both of those papers was of course to demonstrate that, while potentially more temporally stable than TTDs, even SAS distributions should be considered time-variant in most cases.

(12) Page 52, lines 11-14: And note some important works preceding those you have cited [e.g. Ali et al., 2014; Fiori and Russo, 2008; Fiori et al., 2009; Rinaldo et al., 2011; Russo and Fiori, 2009].

(13) Page 52, lines 15-19: This paper [Pangle et al., 2017] provides a clear illustration of your point, where multiple hydrologic variables, and a tracer breakthrough curve, can be simulated quite accurately with traditional flow and transport models, while still not accurately reproducing the known age distributions of water in the flow out of the system.

Technical Edits:

(1) Page 2, equation 2: Should you have subscript j along with subscripts "new" and "old" attached to C terms in equations 2 and 3? In practice they are not uniquely measured at every time step, but must at least be assumed at each time step.

(2) Page 61, equation A5: Is it redundant to use overbars and angled brackets on the same term?

(3) Figures 4, 6, and 8: Some kind of discontinuity in the vertical axes of the some of the graphs. Maybe due to pdf rendering? Not important, just bringing it to attention in case it can be easily fixed.

(4) Page 28, line 24: Should the subscripts be j here rather than i? Since times associated with sampling P have been denoted with i whereas those for Q with j?

(5) Page 30, line 17: Looks like a typo on the subscript of second term on the right-hand side of the equal sign.

Cited References

Ali, M., A. Fiori, and D. Russo (2014), A comparison of travel-time based catchment transport models, with application to numerical experiments, J. Hydrol., 511, 605-618, doi:10.1016/j.jhydrol.2014.02.010.

Allen, S. T., R. F. Keim, H. R. Barnard, J. J. McDonnell, and J. Renée Brooks (2016), The role of stable isotopes in understanding rainfall interception processes: a review, Wiley Interdisciplinary Reviews: Water, n/a-n/a, doi:10.1002/wat2.1187.

Allison, G. B., C. J. Barnes, and M. W. Hughes (1983), THE DISTRIBUTION OF DEU-TERIUM AND O-18 IN DRY SOILS .2. EXPERIMENTAL, J. Hydrol., 64(1-4), 377-397, doi:10.1016/0022-1694(83)90078-1.

Barnes, C. J., and G. B. Allison (1983), THE DISTRIBUTION OF DEUTERIUM AND O-18 IN DRY SOILS .1. THEORY, J. Hydrol., 60(1-4), 141-156, doi:10.1016/0022-1694(83)90018-5.

Benettin, P., Y. van der Velde, S. E. A. T. M. van der Zee, A. Rinaldo, and G. Botter (2013), Chloride circulation in a lowland catchment and the formulation of transport by travel time distributions, Water Resources Research, 49(8), 4619-4632, doi:10.1002/wrcr.20309.

Botter, G., E. Bertuzzo, and A. Rinaldo (2010), Transport in the hydrologic response: Travel time distributions, soil moisture dynamics, and the old water paradox, Water Resources Research, 46, doi:W03514 10.1029/2009wr008371.

Botter, G., E. Bertuzzo, and A. Rinaldo (2011), Catchment residence and travel time distributions: The master equation, Geophysical Research Letters, 38, doi:L11403 10.1029/2011gl047666.

Fiori, A., and D. Russo (2008), Travel time distribution in a hillslope: In-sight from numerical simulations, Water Resources Research, 44(12), doi:W12426 10.1029/2008wr007135.

Fiori, A., D. Russo, and M. Di Lazzaro (2009), Stochastic analysis of transport in hillslopes: Travel time distribution and source zone dispersion, Water Resources Research, 45, doi:W08435 10.1029/2008wr007668.

Harman, C. J. (2015), Time-variable transit time distributions and transport: Theory and application to storage-dependent transport of chloride in a watershed, Water Resources Research, 51(1), 1-30, doi:10.1002/2014wr015707.

Heidbuchel, I., P. A. Troch, and S. W. Lyon (2013), Separating physical and meteorological controls of variable transit times in zero-order catchments, Water Resources Research, 49(11), 7644-7657, doi:10.1002/2012wr013149.

Heidbuchel, I., P. A. Troch, S. W. Lyon, and M. Weiler (2012), The master transit time distribution of variable flow systems, Water Resources Research, 48, doi:W06520 10.1029/2011wr011293.

Pangle, L. A., M. Kim, C. Cardoso, M. Lora, A. A. Meira Neto, T. H. M. Volkmann, Y. Wang, P. A. Troch, and C. J. Harman (2017), The mechanistic basis for storage-dependent age distributions of water discharged from an experimental hillslope, Water Resources Research, 53(4), 2733-2754, doi:10.1002/2016WR019901.

Rinaldo, A., K. J. Beven, E. Bertuzzo, L. Nicotina, J. Davies, A. Fiori, D. Russo, and G. Botter (2011), Catchment travel time distributions and water flow in soils, Water Resources Research, 47, doi:W07537 10.1029/2011wr010478.

Rodriguez, N. B., K. J. McGuire, and J. Klaus (2018), Time-Varying Storage–Water Age Relationships in a Catchment With a Mediterranean Climate, Water Resources Research, 54(6), 3988-4008, doi:doi:10.1029/2017WR021964.

Russo, D., and A. Fiori (2009), Stochastic analysis of transport in a combined heterogeneous vadose zone-groundwater flow system, Water Resources Research, 45, doi:W03426 10.1029/2008wr007157.

van der Velde, Y., G. H. de Rooij, J. C. Rozemeijer, F. C. van Geer, and H. P. Broers

(2010), Nitrate response of a lowland catchment: On the relation between stream concentration and travel time distribution dynamics, Water Resources Research, 46, doi:W11534 10.1029/2010wr009105.

van der Velde, Y., P. Torfs, S. van der Zee, and R. Uijlenhoet (2012), Quantifying catchment-scale mixing and its effect on time-varying travel time distributions, Water Resources Research, 48, doi:W06536 10.1029/2011wr011310.

---

## Author Comment (AC1) · 8 Oct 2018

Let me begin by thanking Anonymous Referee #1 (hereafter AR1) for taking on the job of reading, understanding, and evaluating this manuscript. The paper is long, technical, and at times complex, and I appreciate the effort involved in reviewing it.

AR1 presents a nice overview of some salient features of the ensemble hydrograph separation approach, and points out that it could find application in re-analyzing many tracer data sets that have been collected in experimental catchments. I would, however, qualify AR1's statement that "*The method can take advantage of all these existing data sets to possibly refine our view of what MTTs are for low-order catchments in*

[Figure]

*different regions, and what the form of the TTDs actually look like (at least the early-time portions of TTDs; the long tails remain difficult to constrain).*" I share AR1's view with respect to TTD's, but would like to comment further on the implications of the difficulty in constraining the long tails.

The difficulty in constraining long tails (from conservative tracer data) is not unique to ensemble hydrograph separation, but instead is shared by every estimation method. This is not a feature of the estimation methods, but instead is intrinsic in the conservative tracers themselves, because catchment tracer inputs lack sufficient long-timescale variability to leave a measurable long-timescale signal in the output [DeWalle et al., 1997; Seeger and Weiler, 2014; Kirchner, 2016]. Because the long tails of the TTD exert significant leverage over the MTT, the estimation of MTT's from typical conservative tracer data is simply infeasible by any method that does not invoke strong and unverifiable assumptions (such as an assumed shape of a transit time distribution). Thus I would not try to "refine our view of what MTT's are", beyond pointing out that we should seek more reliably quantifiable transit time metrics instead (such as new water fractions introduced here, or the young water fraction Fyw that I introduced in 2016).

AR1 raises an interesting challenge: "*I would ask the author to consider, and possibly elaborate in the discussion, about what might be the advantage of this method over the contemporary methods of lumped-parameter-transport modeling based on time-variable TTDs and SAS distributions/functions? For those researchers organizing new field studies and networks of sites, and that are reasonably competent and diligent with field-data collection (i.e. avoiding lots of data gaps), is there any reason to consider this method? One may argue that it is not any simpler to implement, and cannot achieve similar temporal resolution when estimating time-variable TTDs and SAS distributions. As discussed by the author, application to short subsets of data (for comparison of TTDs at different times) reduces sample size and enhances uncertainty in the regression approach.*"

Advantages of this approach, many of which are already mentioned in the text, include:

a) First and foremost, this approach has been rigorously benchmark tested, and most other approaches have not. Thus we know how well it should work, and what its capabilities, biases, and uncertainties are likely to be, when applied to real-world data. This information simply does not exist for most other approaches.

b) This approach is not based on integration and therefore errors in the inputs do not accumulate over time, as they do in essentially every other method for estimating transit time distributions (except the spectral method of Kirchner et al. (2000, 2001). This is a significant advantage, because it is notoriously difficult to accurately measure catchment-scale inputs (either in quantity or concentration), even with "reasonably competent and diligent field-data collection".

c) What AR1 dismisses as a minor matter of being "reasonably competent and diligent with field-data collection" and "avoiding lots of data gaps" is actually a serious issue that is rarely taken seriously. Most existing methods for analyzing catchment data are based on convolutions (either explicitly, or embedded in a model). Convolutions require *absolutely continuous input data* - not just "avoiding lots of data gaps". There is a presumption in our field that small numbers of data gaps don't matter, as long as one fills them using some kind of ad hoc procedure. Whether this presumption is justified or not is anybody's guess, because it hasn't been tested. But simple mathematics would suggest that this presumption may not, in fact, be justified, because in convolutions, errors accumulate.

d) The existing methods for estimating time-variable TTDs generally do not estimate time-variable TTDs of real-world catchments at all; instead they measure time-variable TTDs of models that are fitted to catchment data, often with rather large prediction errors that the authors somehow manage to just dismiss with a wave of the hand ("large" errors mean those that are much larger than the putative model uncertainties). Whether these fitted TTDs are informative or misleading depends on whether the underlying mechanisms are actually correct, which is unknown and unknowable. (It's not good enough that the mechanisms "seem reasonable", because that says only that

they agree with our preconceptions, not that they accurately describe the real world.)

e) AR1 says that, "One may argue that it is not any simpler to implement." I would argue the opposite. New water fractions can be estimated using bog-standard linear regression; really, how much simpler could that be? And TTD's in this approach are estimated with multiple linear regression, albeit with a few minor twists. I am now writing and testing an R script that handles all the calculations so that users don't have to figure out the details themselves (this script will be published separately, after it goes through several rounds of beta testing in our group). By contrast, the implementations of many other methods are anything but straightforward, in part because they often involve a series of ad hoc choices (among model structures, simulation algorithms, parameter calibration procedures, data assimilation methods, gap-filling protocols, etc. etc. etc.) whose implications are unknown, because they haven't been tested.

f) AR1 also says that, "application to short subsets of data (for comparison of TTDs at different times) reduces sample size and enhances uncertainty in the regression approach," as if this issue were somehow unique to my approach. But of course it isn't unique; instead, it's inherent in any data-driven analysis. The crucial difference is that my analysis recognizes this problem and quantifies its implications, because I've bothered to look. The fact that others may not have bothered to look doesn't mean that their approaches are magically freed from the problem of data scarcity. Of course, one can disguise the data scarcity problem by assuming some rule (such as a SAS function) that specifies how the TTD varies over time, and then fitting this globally to the whole data set. But that puts the answer in from the beginning, because the relationships among the TTD's are determined by the researcher's assumptions about them (e.g. that they obey a SAS function), rather than letting the data determine what the answer is.

Specific Comments:

(1)     *Page 5; line 24: Another complication would seem to be the fact that CQ(j)*

*will be strongly temporally correlated with CQ(j-1). Most perennial streams will be hydraulically connected to a riparian aquifer, and receiving discharge from that aquifer so long as the stream is gaining. The water volume in that aquifer may be 1 or 2 orders of magnitude greater than the volume of water associated with typical precipitation events. CQ(j-1) (i.e., old water) and CQ(j) should then be strongly temporally correlated. In other cases Cnew(j) might also be strongly temporally correlated with CQ(j), for example, in catchments with widespread impervious surfaces that induce infiltration-excess-overland flow. Whether or not the latter type of temporal correlation would exist would also depend on the temporal resolution of sampling. Is this potential for temporal correlation within the explanatory and dependent variables problematic for the linear regression approach here?*

No, for the simple reason that in ensemble hydrograph separation, CQ(j) and Cnew(j) are both normalized by subtracting CQ(j-1). Of course AR1 is correct that streams are mostly fed by aquifers with residence times that are much longer than individual precipitation events, and therefore tracer concentrations in streamflow are serially correlated. This has been known for decades, and many of us have written about it. The whole point of subtracting CQ(j-1) is to removing any effects that are common to both CQ(j) and CQ(j-1), including legacies of past precipitation inputs, measurement biases, isotopic fractionation effects, and so forth. Subtracting CQ(j-1) from Cnew(j) is usually not consequential in practice, because Cnew will typically be much more variable than CQ, but it is done for the sake of consistency, and also to make the derivation conceptually analogous to conventional hydrograph separation.

In the paper's benchmark tests (e.g., Fig. 1), the temporal correlations described by AR1 are present (and often quite strong). Nonetheless, as the benchmark tests show, ensemble hydrograph separation yields quantitatively accurate estimates of the new water fractions and transit time distributions. That is the power of benchmark tests: one does not need to wonder, based on intuition, whether issues like this are, as AR1 puts it, "problematic". The tests show whether they are, or not.

(2)   *Page 6, line 3-5: What would qualify as small?*

As one can see from Eq. A7, "small" in this case means small in comparison to Fnew itself. I will add this to the text.

*The condition in line 5 would not necessarily be true would it? In the case of stable-isotopes as tracers, if 1) rainfall is distributed uniformly throughout the year, 2) there is seasonal variation in the mean Cnew(j), and 3) streamflow is generated preferentially by precipitation that falls in the cold season, then this difference would not be expected to be zero, would it? These conditions are fairly common.*

AR1 is correct that one can imagine scenarios in which the mean of Cnew(j)-CQ(j-1) may not be nearly zero (and I will change "should be nearly zero" to "should usually be small" to account for such cases). But the condition in line 5 is just one of two conditions, which are multiplied together. The second condition is that the slope of the relationship between Fnew(j) and Cnew(j)-CQ(j-1) should be small. This will normally be the case, and it makes the net effect on Fnew small, regardless of what happens to the first term.

Again this points to the importance of rigorous benchmark tests - which, unfortunately, have not been conducted for other methods that have been used to estimate transit times from tracer data. One doesn't need to speculate about whether these issues are significant or not; one can simply do the test. Unlike other benchmark tests, the ones that I have conducted do not start by assuming that the assumptions underlying the method actually hold. Instead, they start with a wide range of reasonable assumptions that are not tailored to the method being tested, and test whether the method still works, even when its underlying assumptions do not hold.

(3)   *Page 13, lines 8-10: Could you comment on the rationale for randomly assigning $iA^2d'2H$ values to precipitation in this way, in light of the fact that actual $iA^2d'2H$ values are often correlated with cumulative rainfall amounts/unit time? I ask only because the existence of this correlation would seemingly have bearing on condition 2 on page*

*6. If cumulative rainfall amount during a storm is not correlated with iA$^2$d'2H, then in principle either very negative iA$^2$d'2H values, or values near zero (a plausible upper boundary of the variable's range), could be observed for large storms that generate greater Fnew. In that case condition 2 on page 6 might be questionable. Seemingly the random assignment here in some sense violates that condition, yet the results in Figures 1 and 2 are quite good?*

One often sees extreme isotopic compositions at the beginning or end of a storm event, when precipitation rates are small. The water that makes up the bulk of the event usually has less extreme isotope values. But in any case, ensemble hydrograph separation will typically be applied to daily and weekly tracer time series (since these are by far the most widely available isotope records), and within-storm isotopic dynamics will be largely masked on those time scales.

*(4)     Page 26, lines 28-30 and Page 27, lines 1-3: Worst-case scenario based on what? Estimates of actual evaporative enrichment measured in surficial soil water? Or plausible ranges from models [e.g. Allison et al., 1983; Barnes and Allison, 1983]? Certainly in semi-arid environments, or where vegetation is interspersed (e.g. tree plantations or agricultural settings) there could be more than 20 ? enrichment of soil water before it percolations below the root zone. The fractionation effect could vary strongly at daily and weekly time scales depending on storm frequency and intensity, and potential evaporation (from plant surfaces and surficial soil). The fractionation effect could in fact be strongly correlated with storm sizeâAÂćTÂŭ not a constant offset or random fluctuation that introduce no bias. And of course this simulation makes no consideration of spatial variability of the fractionation effect, which might occur due to substantial land-cover variability (e.g. catchments with mixed bare-soil and vegetated cover; wetland-meadow-forest transitions; partially snow-covered with spatially- and temporally-variable melt dynamics, etc.). Finally, it's worth noting that the isotope composition of infiltrating water can be different than precipitation (greater or lesser) due to evaporation from, or storage within, plant canopies [Allen et al., 2016]. These*

*effects can be of variable sign and magnitude during individual storms. I would argue it is these storm-specific effects that matter not an average offset over the long term because these storm-specific effects influence the individual values of Cnew -Cold in the regression model.*

*I think the concluding statement about this analysis (page 27, lines 1-3) should be more suggestive than definitive. People will read this and make the convenient assumption that a single precipitation collector is all that is needed in the field, rather than the more labor-intensive alternative of having multiple precipitation/throughfall/snowfall collectors (to capture some essence of spatial variability, and thus leading to more samples to analyze in the lab). This is an aspect of the method that should at least be examined in a few field studies in catchments that vary climatically and ecologically, rather than assumed unimportant based on this useful, but inconclusive, simulation. The same problem exists among most historical studies of mean transit times in forested catchments: the input data sets in the convolution integral are all biased to some unknown degree due to the spatial heterogeneity of storage, throughfall, and evaporation.*

I agree that the concluding statement (page 27, lines 1-3) should be suggestive rather than definitive; that's why the statement said, "Figure 10 thus *suggests* that ensemble hydrograph separation *should* yield realistic estimates of new water fractions...". I will nonetheless remove the term "worst-case scenario", and instead say, "even with substantial confounding by evaporative fractionation."

Nonetheless, the scenario I used was indeed designed to be a worst-case scenario, in the specific sense that it is designed to lead to biased estimates of new water fractions. It intentionally creates an artifactual correlation between the degree of (seasonal) isotopic fractionation and the (seasonal) variation in the true isotopic ratio of the measured precipitation - and such an artifactual correlation could potentially skew the regressions that ensemble hydrograph separation relies on. By contrast, the effects of random variations in isotopic fractionation will tend to average out, and will not lead to biased new water fractions. Likewise, persistent biases from isotopic fractionation will also not bias

the new water fractions, because regressions ignore constant offsets. Isotopic fractionation that is correlated with the true isotopic ratio of the measured precipitation has the greatest potential to mess up the results, which is why I've used that kind of fractionation here.

Of course one can imagine an wide variety of isotopic fractionation scenarios, as well as lots of other complicating factors like spatial heterogeneity. But let's keep this all in perspective. These complications will cause problems for *every* method for inferring transit times from tracer data. The standard of practice in our field seems to be to dream up new techniques and rush immediately to applications with field data, without bothering to test whether they really measure what we think they do, or without even quantifying any of the uncertainties involved. As far as I know (and I would be happy to be corrected on this), the kind of mathematical analyses that are performed in Appendix A - as well as, for that matter, anything nearly as rigorous as the benchmark tests in the rest of the paper - have not been performed for *any* of the other methods for inferring transit times from tracer data.

(5) *Page 29, line 10: Could you clarify here if lag time m corresponds to the most distant past time when a measurement of CP is available, or can the value of m be chosen even among those past times when CP was measured? You seem to be implying the latter case, but I don't understand why you would potentially lump multiple measured values of CP into the Colder term, rather than using the values individually in the regression analysis. Perhaps this is clarified later on in the text.*

The lag time m is simply the longest lag time for which one wants to estimate the TTD. This should not be the most distant time for which one has $C_P$ measurements, because any estimates on that time scale will be massively uncertain (due to a loss of statistical degrees of freedom). In any case, the method does not "lump multiple measured values of $C_P$ into the $C_{older}$ term" at all. Instead, as shown later on that page and the next page, the method uses $C_Q(j-m-1)$ in place of $C_{older}$, in order to factor out long-term patterns in stream concentrations due to legacy effects of prior

inputs (potentially even from before the beginning of the data series). In other words, we use the catchment to integrate over those past inputs - we don't do it ourselves.

(6)    *Page 40, lines 15-20: This seems related to the question I posed in specific comment 1. If that is correct, perhaps you could foreshadow for the reader at that location in the text that this issue is discussed later.*

A better approach is to simply mention the function of C_Q(j-1) as a "filter" for legacy effects of past inputs, directly at the point where the reader needs to know it (between equations 8 and 9). I'll do that.

(7)    *Page 45, lines 15-25: Could you clarify in this case how the parameter m is determined? Seemingly if you want to isolate some subsets of the time series of Q for comparison of the average-backward TTDs, for example, Q during September versus April at Smith River, the CQj and distribution of water ages at any time during either interval could be strongly influenced by CP that occurred over many historical time intervals prior to September or April.*

This is the whole point of using a reference concentration: to correct for the legacy effects of more distant tracer inputs. The parameter m is not determined by analysis, but instead is simply chosen by the user. If you want to look at lags between 0 and 10 time steps, then m is 10. If you want to look at lags between 0 and 20 time steps, m is 20. Whatever the value of m, the streamwater concentration at lag m+1 is used as a reference concentration (for both CQj and CP), to normalize for the inherited effects of precipitation tracers that fell prior to lag m. The smaller the value of m, the more important it is to subtract these legacy effects (and the more effectively the reference concentration does this).

(8)    *Page 47, figures 16a,b: Regarding the orange line, it's interesting and counter-intuitive to me that these average TTDs would show that volume fractions of Q with ages of only a few days would be (slightly) more probable than volume fractions of Q with ages of several days to weeks. It rains so infrequently during the summer, most*

*often there would be little to no water at all within the catchment that had residence time of only a few days. I would think there should be a slight increasing trend in these probabilities from left to right on the transit-time axis, up to a point where the trend then turns back downward. Is this possibly because those few storms that do occur during the summer (and deliver some water to the stream with age of only a few days) represent a disproportionately large fraction of total Q over the summer months?*

Intuition is a tricky thing. Remember that infrequent rainfall means that discharge will contain little zero-day-old water (because it probably didn't rain today), and also little one-day-old water (because it probably didn't rain yesterday), and also little two-day-old water (because it probably didn't rain the day before yesterday), and so forth, all the way out to the beginning of the dry season. The TTD's average these probabilities over the few days that had rain and the many days that didn't. Thus it's not possible, *on average*, to have (for example) more rain two weeks ago than rain one day ago, since each rain event is "two weeks ago" for a day two weeks into the future, and "one day ago" tomorrow.

(9)   *Page 49, line 4: Certainly there are several good examples of where storage dependence has been examined, even one case in the same climate as the Smith River. These are worth acknowledging* [*Benettin et al., 2013; Harman, 2015; Heidbuchel et al., 2013; Heidbuchel et al., 2012; Rodriguez et al., 2018; van der Velde et al., 2010; van der Velde et al., 2012*].

The statement in question is, "Antecedent wetness has been recognized as a controlling factor in catchment storm response (e.g., Detty and McGuire, 2010; Merz et al., 2006; Penna et al., 2011), but its effects on solute transport at the catchment scale have not been widely explored." I meant to refer to solute transport effects estimated from catchment data (of which there have indeed been few or none), as distinct from the behavior of calibrated models (of which there have been several, as pointed out by AR1). All of the references mentioned by AR1 concern inferences drawn from the behavior of calibrated models, where it is generally unclear whether the storage-dependence is

strongly constrained by the data, or whether it would come and go within the plausible ranges of the model parameters, or even whether it results from arbitrary model choices (such as the frequent use of linear reservoirs). None of the mentioned references provide anything remotely resembling Figs. 18a and c, which show the functional dependence of "new water" on antecedent moisture, directly from catchment data (and as far as I can tell after re-reading these papers, some of them do not address the general topic directly at all).

In any case, a more precise statement would be, "Antecedent wetness has been recognized as a controlling factor in catchment storm response (e.g., Detty and McGuire, 2010; Merz et al., 2006; Penna et al., 2011), but its effects on solute transport at the catchment scale have rarely been quantified, outside of the context of calibrated simulation models (e.g., Heidbüchel et al., 2012; van der Velde et al., 2012; Harman, 2015; Rodriguez et al., 2018)." This will be adopted in the revision.

(10)    *Page 51, line 19: I would argue that point 5 here should perhaps be omitted. While suggestive, I don't think the simulation that leads to this conclusion is a very realistic representation of isotope fractionation effects over time or space. See specific comment 4 above. Also, while I'm no proponent of traditional hydrograph separation, I don't think the first sentence in point 5 is really true. The method is not necessarily vulnerable to biases in tracer measurements resulting from fractionation. If fractionation has occurred in the end member, the effect should be apparent in the measured delta values. Fractionation doesn't inhibit accurate quantification of the tracer concentration in the end member.*

The statement in question refers to biases (i.e., persistent offsets) in tracer measurements, and as such, it is mathematically correct as stated.

Concerning conventional hydrograph separation, the problem isn't that fractionation "inhibits accurate quantification" of the tracer concentration, it's that the accurately quantified tracer concentration does not represent what actually goes into the catchment.

One measures the tracer concentration in a rainfall collector, for example, not the (fractionated, or to put it more precisely, differently fractionated) throughfall or soil moisture that eventually becomes streamflow. The difference between what is measured and what is the actual end-member can potentially distort the results of traditional hydrograph separation, whether that difference fluctuates randomly (and thus adds random noise to the results) or is persistent over time (and thus adds persistent bias to the results). Neither random fractionation nor persistent fractionation bias poses a significant problem in ensemble hydrograph separation (the first will tend to average out, and the second will not affect the regression slopes on which the method depends). There are some particular patterns of fractionation effects that could pose a problem for ensemble hydrograph separation (and the paper specifies what they are), but they would also pose at least as big a problem for conventional hydrograph separation.

(11)    *Page 52, lines 7-9: Perhaps put "time-invariant" in italics here for emphasis, since the focus of both of those papers was of course to demonstrate that, while potentially more temporally stable than TTDs, even SAS distributions should be considered time-variant in most cases.*

Emphasizing "time-invariant" in this way could lead readers to believe that time-variant SAS functions somehow avoid the estimation problems mentioned in this paragraph, which is not correct. The better solution is to simply remove the phrase "time-invariant" as it pertains to SAS functions.

(12)    *Page 52, lines 11-14: And note some important works preceding those you have cited* [*e.g. Ali et al., 2014; Fiori and Russo, 2008; Fiori et al., 2009; Rinaldo et al., 2011; Russo and Fiori, 2009*].

These are interesting papers but they are not relevant to the statement in the text, which is, "Yet another approach that is coming into more frequent use is to calibrate a conceptual or physically based model to reproduce, as closely as possible, the observed hydrograph and streamflow tracer time series, and then infer the catchment transit time

distribution from particle tracking within the model." The papers that AR1 mentions all concern simulations of transit time distributions under various physical or conceptual assumptions, but none of them infer real-world transit time distributions by calibrating those models to observational data. Thus none of them are relevant citations for the specific statement made in the text. (If the topic were simulations of transit time distributions more generally, then I would also cite Kirchner et al. 2001, which precedes any of the papers mentioned by AR1 by almost a decade.)

(13)    *Page 52, lines 15-19: This paper [Pangle et al., 2017] provides a clear illustration of your point, where multiple hydrologic variables, and a tracer breakthrough curve, can be simulated quite accurately with traditional flow and transport models, while still not accurately reproducing the known age distributions of water in the flow out of the system.*

If Pangle et al.'s (2017) results show this, it's unfortunate that Pangle et al. don't come right out and say so. In particular, although their section 3.3 says that different parameterizations reproduce the observed Q, S, and H (discharge, storage, and hydraulic head), it does not say that these parameterizations also reproduce the tracer breakthrough curve. Thus one needs to "read between the lines" to draw the conclusion that AR1 draws from Pangle et al. (a conclusion which the authors themselves do not draw, at least in any way that I can find).

Technical Edits:

(1)    *Page 2, equation 2: Should you have subscript j along with subscripts "new" and "old" attached to C terms in equations 2 and 3? In practice they are not uniquely measured at every time step, but must at least be assumed at each time step.*

No. Equation 2 describes conventional hydrograph separation, in which (see line 15) "one assumes that streamflow is a mixture of two end-members *of fixed composition*" (emphasis added). If the composition is fixed then there is no subscript.

(2) *Page 61, equation A5: Is it redundant to use overbars and angled brackets on the same term?*

In general it's not. Angled brackets indicate that whatever is contained between them is averaged. Overbars indicate individual terms that are averaged. Thus, for example, $\langle \bar{x}y \rangle \neq \langle xy \rangle$. In one case in equation A5, the angled brackets are indeed redundant, because $\langle \bar{\beta}\bar{x} \rangle = \bar{\beta}\bar{x}$ . But in this case the brackets are retained for clarity, since all the angle-bracketed terms in (A5) result from the expansion of the angle-bracketed term in (A4).

(3) *Figures 4, 6, and 8: Some kind of discontinuity in the vertical axes of the some of the graphs. Maybe due to pdf rendering? Not important, just bringing it to attention in case it can be easily fixed.*

This is an annoying pdf rendering issue. The original eps files do not have these defects, so they will not appear in the final version.

(4) *Page 28, line 24: Should the subscripts be j here rather than i? Since times associated with sampling P have been denoted with i whereas those for Q with j?*

Correct! You have clearly read the manuscript *very* carefully. This is a silly typo that will be fixed in the final version.

(5) *Page 30, line 17: Looks like a typo on the subscript of second term on the right-hand side of the equal sign.*

Good eyes! This is a rendering error in MS Word's equation editor. It will be fixed.

–

–

DeWalle, D. R., Edwards, P. J., Swistock, B. R., Aravena, R., and Drimmie, R. J.: Seasonal isotope hydrology of three Appalachian forest catchments, Hydrol. Process., 11, 1895-1906, 1997.

Seeger, S. and Weiler, M.: Reevaluation of transit time distributions, mean transit times and their relation to catchment topography, Hydrol. Earth Syst. Sci., 18, 4751-4771, doi:10.5194/hess-18-4751-2014, 2014.

Kirchner, J. W.: Aggregation in environmental systems - Part 1: Seasonal tracer cycles quantify young water fractions, but not mean transit times, in spatially heterogeneous catchments, Hydrol. Earth Syst. Sci., 20, 279-297, doi:10.5194/hess-20-279-2016, 2016.

Kirchner, J. W., Feng, X., and Neal, C.: Catchment-scale advection and dispersion as a mechanism for fractal scaling in stream tracer concentrations, J. Hydrol., 254, 81-100, 2001.

---

## Author Comment (AC2) · 9 Oct 2018

The discussion paper, "Quantifying new water fractions and transit time distributions using ensemble hydrograph separation: theory and benchmark tests," uses weighted regressions in several places (specifically Eqs. 16, 28, and 54). These calculations are based on a simple short-cut that is widely used in the physical sciences, namely that if one multiplies both the $x_j$'s and $y_j$'s by the square root of the weights $w_j$, an ordinary least-squares regression of the square-root-weighted $x_j$'s and $y_j$'s will yield an accurate estimate of the weighted regression slope.

However, this short-cut requires that the volume-weighted averages of the $x_j$'s and

$y_j$'s are both zero (or, as an approximation, almost zero). That condition is met in the calculations in the paper, but might not be met in other situations in which these equations could potentially be applied. To avoid any misunderstanding, the general equations for weighted regression (which do not require this approximation) will be presented in the final version of the paper.

---

## Short Comment (SC1) · 18 Oct 2018

General comments:

This paper has been really pleasant to read. The quality of the writing, of the figures, and of the mathematics is really high. The amount of meaningful explanations is impressive. I think the paper will have a strong impact on the isotope hydrology community. I think we need more approaches such as this one in tracer hydrology. That being said, I would like to mention two things that could be discussed.

The first one is more context about travel time modeling. While the introduction and

the discussion compare well this ensemble approach to the traditional hydrograph separation, a large part of the paper also deals with determining travel time distributions (TTDs). Yet, only little is said about travel time modeling, especially in the introduction. I think it should be mentioned that the ensemble approach deals with a current need in isotope hydrology to have more data-driven approaches and non-parametric TTDs. I think this is exactly what the proposed approach brings compared to already existing approaches, but not more. Unlike reviewer n°1, I believe we should not try to formally compare methods which have different purposes. This approach calculates only the streamflow average TTD suggested by the tracer data, without assuming its shape. This is novel and important. Yet the proposed method can only be used for the period covered by training data (i.e. "backwards"), and for streamflow only. Using StorAge Selection (SAS) functions with assumed shapes allows one to obtain the time-varying TTDs at every moment and in every flux (backwards), and the Residence Time Distributions (RTDs). But more importantly, SAS function allow one to simulate other time-varying solute fluxes (e.g. Benettin et al., 2015) with the calibrated model in a forward way (even outside the period covered by training data). Note that a model based on SAS functions can consist of just a handful of parameters (e.g. Benettin et al. 2017) which makes it really competitive. Yet, I also agree that there are clear limitations in approaches based on SAS functions.

My second comment relates to the potential limitations of the proposed approach. I think that all the choices made to derive the mathematical solutions were presented as if they are the best choices for any tracer data set, or the only choices possible. This may not be true in all cases. The discussion would benefit from an objective assessment of the problems that could occur when trying to apply the approach to real tracer data. In my opinion, this ensemble approach will be accurate only for the left tail of the TTDs, while it truncates (cf. equation 30) older ages. This is a critical problem in travel time modeling in general (Stewart et al., 2012; Stewart & Morgenstern, 2016). It is already mentioned in the reply to reviewer n°1, but I think it should be clearly written in the discussion as well.

Specific comments:

(1) To give more context in the introduction you could mention and describe briefly the common methods to estimate TTDs, namely the Lumped Parameter Models (e.g McGuire & McDonnell, 2006, and references therein), flux tracking in conceptual models (e.g. Hrachowitz et al., 2013), SAS functions applied to a single control volume (e.g. Benettin et al., 2017), and particle tracking in distributed models (e.g. Davies et al., 2013; Danesh-Yazdi et al., 2018). Doesn't the ensemble approach answer the need to have alternatives to these methods, which all need to assume an underlying model for water transport?

(2) P7, L11-13: Least squares regression means that any real data set with "outliers" (which may just be tracer values one did not expect) is likely to adversely affect the results from the ensemble approach, as it is suggested here. Same for the least squares solution in equation 38. This is in my opinion one of the limitations of the proposed method. This should perhaps be mentioned in the discussion. Can iteratively reweighted least squares or another robust regression technique be used instead? I agree that this approach assumes no model for the transport of tracers, yet it does assume a model for the errors between the regression and the measurements (i.e. the residuals). This is similar to the choice of an objective function in traditional model calibration, and deserves attention. For example, commonly used assumptions about streamflow residuals were shown to be often violated, because of autocorrelation, non-normality, and heteroscedasticity (Schoups & Vrugt, 2010). Are the tracer residuals in this work likely to show non-normality, autocorrelation, and heteroscedasticity as well? Although $CQ(j)$ and $Cnew(j)$ are both "normalized" by subtracting $CQ(j-1)$, there could be autocorrelation of higher order than just 1. How does the variance of errors change with larger flashy events?

(3) P12, L19-20 All the benchmarking is done for a catchment without evapotranspiration. This points to a more general concern with the ensemble approach. No assumption is made explicitly about what happens to the tracer masses between precipitation

and streamflow. This means that the method may try to find direct "connections" (in a loose statistical sense here) between tracer inputs from the past and current tracer fluctuations in the stream. Intermediary (unconsidered) processes may still be important to explain the transformation from one to the other. I especially think of processes affecting the lumped catchment tracer mass balance, which is an expression that was not considered in the approach. In that regard, how are the results expected to change if ET is actually used in the benchmark tests? Is the approach robust for real catchments where ET can be a major part of the water balance? Here I am not considering the effects of fractionation which were already dealt with, but the selective removal of certain tracer masses (associated with particular ages, i.e. different soil/groundwater mixtures) by ET, which will hence not be available for streamflow.

(4) P20, L4-6: These estimates seem to differ as much as 50% from the known values for the damped catchment and weekly data on figure 4. Many tracer data sets are at weekly resolution and come from "damped" catchments (e.g. Tetzlaff et al., 2009; Pfister et al., 2017). Data-driven approaches are by nature highly sensitive to the quality of data (e.g. variability, resolution, and measurement uncertainty). The proposed approach could thus show some limitations due to its strict data needs in some cases. This could be mentioned in the discussion.

(5) P20, L19-20: A weekly sampling routine is likely to contain more "baseflow" samples which reflect older water contributions. This results in an underestimation of QFnew as shown here. Yet how can the fraction of new water with respect to discharge be underestimated while the fraction of new water with respect to precipitation is overestimated? These quantities refer to the same mass of water in streamflow.

(6) P28, L2: Here it is assumed that values of Cp and CQ at all times corresponding to indexes j or j-k are known (except a few, which require the solutions proposed in 4.2 and 4.3). In practice it is very likely that the sampling interval is irregular, such that there is not a perfect correspondence between measurement times, and required times indexed by j or j-k. Any recommendation on how to best adjust the measurement

time series so that these terms are defined properly would be welcome. Similarly, the method requires the same number of measurements in precipitation and in streamflow. How could we deal with this in various research catchments as this is often not the case?

(7) P29, L12: Here it is assumed that the most recent precipitation events have more weight in the current tracer fluctuations in the stream than older inputs. This is implicitly reflected by the truncation of the sum in equation 30. This is also reflected by the estimated travel times that mostly stay below a few months. Although this assumption about tracer contributions is likely to be valid, catchment travel times are known to be generally in the order of magnitude of a few years and even decades (McGuire & McDonnell, 2006). This is all the more true when age estimates are based on tritium measurements (Stewart et al., 2010). Would the ensemble approach be robust in catchments where streamflow is volumetrically dominated by water older than a few months?

(8) P29, L12: Regarding the linear algebra, how large can the truncation index m be in practice, given that computationally intensive large matrix operations are carried out? This is especially of interest since the matrices grow with the number of measurements in both dimensions, while m needs to be as large as possible for the ensemble method to work well. In my opinion, the discussion should encourage the reader to consider if this approach shows limitations for his/her considered travel times, which may be up to a decade. Can this approach go beyond the left-hand tail of the TTDs or is it limited to the left tail?

(9) P33, L8: Over several years of data, doesn't neglecting 1 mm of precipitation per day sum up to a large value? It could be useful to include some discussion on the effect of that threshold on the results. Are the results highly sensitive to that choice or not?

(10) P39, Figure 11: Deviations between the benchmark TTDs and the estimated ones

are visible here. How could these deviations be described more quantitatively to be more objective?

(11) P40, Figure 12: It looks like the uncertainties are larger for the TTDs which shape is not a classical "L" anymore. The explanation given here is that the effective sample size neff (equation 13) is small because of tracer autocorrelation. Can we not say that an autocorrelation in the tracer time series is universal, as well as a shape of the TTD far from a simple "L"? This seems related to the issue described in comment (2).

(12) There are not many data-driven methods that can yield non-parametric TTDs, which explains why this new approach is really beneficial to estimate TTDs. Yet, I believe that the problem solved here is somewhat similar to what Turner et al. (1987) solved as well, using Kalman filtering approaches (see the parallel between equations 30 and 31 here, and their equations 1 and 2) (see also Turner and McPherson, 1990). They unfortunately did not detail the math behind their approach. Nevertheless, their work present time-varying average transit times, including uncertainties, also derived without assumptions on the shape of the TTDs. This is worth mentioning and comparing to the presented approach in the discussion. Furthermore, Klaus et al., (2015) also presented a data-driven approach that could be worth mentioning and briefly comparing to the presented one. Finally Kim et al. (2016) could estimate not only TTDs but also SAS functions from artificial tracer data (of course under well controlled lab conditions). Their work is worth mentioning because they are able to distinguish the "external" variability of travel times from the "internal" one, unlike the ensemble approach presented here.

(13) P54, L30-32: Doesn't this mean that m should be set bigger?

Small technical comments:

(14) Figure 1 & 4: the colors are not consistent between the legend and the lines in the lower subplots.

References: Benettin, P., S. W. Bailey, J. L. Campbell, M. B. Green, A. Rinaldo, G. E. Likens, K. J. McGuire, and G. Botter (2015), Linking water age and solute dynamics in streamflow at the Hubbard Brook Experimental Forest, NH, USA, Water Resour. Res., 51, 9256–9272, doi:10.1002/2015WR017552.

Benettin, P., C. Soulsby, C. Birkel, D. Tetzlaff, G. Botter, and A. Rinaldo (2017), Using SAS functions and high-resolution isotope data to unravel travel time distributions in headwater catchments, Water Resour. Res., 53, 1864–1878, doi:10.1002/2016WR020117.

Danesh-Yazdi, M., Klaus, J., Condon, L. E., & Maxwell, R. M. (2018). Bridging the gap between numerical solutions of travel time distributions and analytical storage selection functions. Hydrological Processes, 32, 1063–1076. https://doi.org/10.1002/hyp.11481

Davies, J., K. Beven, A. Rodhe, L. Nyberg, and K. Bishop (2013), Integrated modeling of flow and residence times at the catchment scale with multiple interacting pathways, Water Resour. Res., 49, 4738–4750, doi:10.1002/wrcr.20377.

Hrachowitz, M., Savenije, H., Bogaard, T. A., Tetzlaff, D., & Soulsby, C. (2013). What can flux tracking teach us about water age distribution patterns and their temporal dynamics? Hydrology and Earth System Sciences, 17(2), 533–564. https://doi.org/10.5194/hess-17-533-2013

Kim, M., Pangle, L. A., Cardoso, C., Lora, M., Volkmann, T. H. M., Wang, Y., et al. (2016). Transit time distributions and StorAge Selection functions in a sloping soil lysimeter with time-varying flow paths: Direct observation of internal and external transport variability. Water Resources Research, 52, 7105–7129. https://doi.org/10.1002/2016WR018620

Klaus, J., Chun, K. P., McGuire, K. J., & McDonnell, J. J. (2015). Temporal dynamics of catchment transit times from stable isotope data. Water Resources Research, 51, 4208–4223. https://doi.org/10.1002/2014WR016247

McGuire, K. J., & McDonnell, J. J. (2006). A review and evaluation of catchment transit time modeling. Journal of Hydrology, 330(3–4), 543–563. https://doi.org/10.1016/j.jhydrol.2006.04.020

Pfister L, Martínez‐Carreras N, Hissler C. et al. Bedrock geology controls on catchment storage, mixing, and release: A comparative analysis of 16 nested catchments. Hydrological Processes. 2017. https://doi.org/10.1002/hyp.11134

Schoups, G., and J. A. Vrugt (2010). A formal likelihood function for parameter and predictive inference of hydrologic models with correlated, heteroscedastic, and non‐Gaussian errors, Water Resour. Res., 46, W10531, doi:10.1029/2009WR008933.

Stewart, M. K., & Morgenstern, U. (2016). Importance of tritium-based transit times in hydrological systems, WIREs Water 2016, 3:145–154. doi:10.1002/wat2.1134

Stewart, M. K., Morgenstern, U., & McDonnell, J. J. (2010). Truncation of stream residence time: how the use of stable isotopes has skewed our concept of streamwater age and origin, Hydrological Processes 24, 1646-1659, doi:10.1002/hyp.7576

Stewart, M. K., Morgenstern, U., McDonnell, J. J., & Pfister, L. (2012). The 'hidden streamflow' challenge in catchment hydrology: A call to action for stream water transit time analysis. Hydrological Processes, 26(13), 2061–2066. https://doi.org/10.1002/hyp.9262

Tetzlaff, D., Seibert, J., McGuire, K. J., Laudon, H., Burns, D. A., Dunn, S. M., et al. (2009). How does landscape structure influence catchment transit time across different geomorphic provinces? Hydrological Processes, 23(6), 945–953. https://doi.org/10.1002/hyp.7240

Turner, J. V., & Macpherson, D. K. (1990). Mechanisms Affecting Streamflow and Stream Water Quality: An Approach via Stable Isotope, Hydrogeochemical, and Time Series Analysis. Water Resources Research, 26(12), 3005-3019.

Turner, J.V., Macpherson, D.K., Stokes, R.A., 1987. The mechanisms of catchment

flow processes using natural variations in deuterium and oxygen-18. Journal of Hydrology 94, 143–162.

---

## Author Comment (AC3) · 21 Oct 2018

**I thank Nicolas Rodriguez (hereafter NR) for his comments. I have reproduced those comments below (in normal type), with my responses (in bold).**

This paper has been really pleasant to read. The quality of the writing, of the figures, and of the mathematics is really high. The amount of meaningful explanations is impressive. I think the paper will have a strong impact on the isotope hydrology community. I think we need more approaches such as this one in tracer hydrology.

**Thanks.**

[Figure]

That being said, I would like to mention two things that could be discussed.

The first one is more context about travel time modeling. While the introduction and the discussion compare well this ensemble approach to the traditional hydrograph separation, a large part of the paper also deals with determining travel time distributions (TTDs). Yet, only little is said about travel time modeling, especially in the introduction.

**This is because the paper does not deal with simulation modeling (nor is it intended to).**

I think it should be mentioned that the ensemble approach deals with a current need in isotope hydrology to have more data-driven approaches and non-parametric TTDs. I think this is exactly what the proposed approach brings compared to already existing approaches, but not more. Unlike reviewer n?1, I believe we should not try to formally compare methods which have different purposes. This approach calculates only the streamflow average TTD suggested by the tracer data, without assuming its shape. This is novel and important. Yet the proposed method can only be used for the period covered by training data (i.e. "backwards"), and for streamflow only.

**As with any technique for analyzing the behavior of a natural system, yes, this method does use data, and thus "can only be used for the period covered by the training data".**

**But the statement that the method is useful "for streamflow only" is puzzling, given that Section 5.4 explicitly mentions that "these methods could potentially be applied to infer transit times in other catchment fluxes, such as groundwater seepage or evapotranspiration." Of course, one would first need tracer data from those fluxes (as would also be required for any other method with the same objectives).**

Using StorAge Selection (SAS) functions with assumed shapes allows one to obtain the time-varying TTDs at every moment and in every flux (backwards), and the Residence

Time Distributions (RTDs). But more importantly, SAS function allow one to simulate other time-varying solute fluxes (e.g. Benettin et al., 2015) with the calibrated model in a forward way (even outside the period covered by training data). Note that a model based on SAS functions can consist of just a handful of parameters (e.g. Benettin et al. 2017) which makes it really competitive. Yet, I also agree that there are clear limitations in approaches based on SAS functions.

**This is a misleading comparison. Of course with a simulation model one can _simulate_ all kinds of things (without any real-world constraint), but _analysis_ of real-world data is a fundamentally different task. With a simulation model, _of course_ one can _simulate_ age distributions "at every moment and in every flux", including times and fluxes for which one has no data (and thus for which one has no idea whether the simulations are realistic or not). But what does one learn from doing so? One primarily learns about the consequences of one's modeling assumptions, but not about whether they accurately describe the real-world system.**

My second comment relates to the potential limitations of the proposed approach. I think that all the choices made to derive the mathematical solutions were presented as if they are the best choices for any tracer data set, or the only choices possible.

**I presented the choices that were actually made, along with the rationales for them. I neither stated nor implied that they are the best choices for any tracer data set, or the only choices possible, as NR claims.**

This may not be true in all cases. The discussion would benefit from an objective assessment of the problems that could occur when trying to apply the approach to real tracer data.

**I don't know what NR means by an "objective" assessment. The whole point of the paper is to quantitatively test the approach using a benchmark model (so that we know what the right answer would be), using data that are a reasonable**

**approximation to real tracer data. The results are presented in 17 figures and extensive discussion. What more "objective assessment" would one want?**

In my opinion, this ensemble approach will be accurate only for the left tail of the TTDs, while it truncates (cf. equation 30) older ages.

**It is not correct to say that this approach "truncates" the TTD; instead it (correctly) *makes no assumption* about the TTD beyond the specified range of lag times.**

This is a critical problem in travel time modeling in general (Stewart et al., 2012; Stewart & Morgenstern, 2016). It is already mentioned in the reply to reviewer n?1, but I think it should be clearly written in the discussion as well.

**As I have already replied to reviewer #1, the problem with estimating long tails is intrinsic to the use of conservative tracers. It is a problem of the (low) information content in the tracer data on those time scales, and that problem cannot be solved by clever analytical tricks. This can be mentioned in the discussion. Whether it is "a critical problem" depends on whether one is interested in the long tail and the mean transit time (which are difficult or impossible to constrain with conservative tracers), or the shorter-term behavior (which is the focus of my approach, and which we can actually learn a lot about from tracer data).**

1. To give more context in the introduction you could mention and describe briefly the common methods to estimate TTDs, namely the Lumped Parameter Models (e.g McGuire & McDonnell, 2006, and references therein), flux tracking in conceptual models (e.g. Hrachowitz et al., 2013), SAS functions applied to a single control volume (e.g. Benettin et al., 2017), and particle tracking in distributed models (e.g. Davies et al., 2013; Danesh-Yazdi et al., 2018). Doesn't the ensemble approach answer the need to have alternatives to these methods, which all need to assume an underlying model for water transport?

**Yes, exactly. These methods not only require a** *physically correct* **underlying model for water transport (which can be highly problematic in practice), they also require** *continuous input data* **(because in any time-integrating model, errors accumulate). These methods can be briefly mentioned in the introduction or discussion, but the paper is not (and should not become) a review and comment on the broad topic of transit time modeling.**

2. P7, L11-13: Least squares regression means that any real data set with "outliers" (which may just be tracer values one did not expect) is likely to adversely affect the results from the ensemble approach, as it is suggested here. Same for the least squares solution in equation 38. This is in my opinion one of the limitations of the proposed method.

**A limitation compared to what?** *Every other method* **for separating hydrographs, estimating TTD's, or calibrating SAS functions (etc.) is** *also* **potentially vulnerable to outliers. And of course parameter calibrations in simulation models are** *also* **affected by outliers, in ways that are often poorly understood.**

This should perhaps be mentioned in the discussion. Can iteratively reweighted least squares or another robust regression technique be used instead?

**This is something I am investigating. Briefly, although iteratively reweighted least squares (IRLS) can be applied straightforwardly to conventional multiple regressions (where you have complete data), the same is not true when you have missing data (because then you have missing residuals too, and thus no obvious way to identify outliers). In principle IRLS can be used term-by-term to estimate each of the covariances in a multiple regression (in place of Eqs. 40 and 41), but the effects of doing so are not easy to determine a priori. As always, the choice between robust methods (like IRLS) and non-robust methods (like least squares) represents a tradeoff: with robust methods, you get reliable results even if your**

**data are messy, but you lose precision and sensitivity if your data are not messy.**

I agree that this approach assumes no model for the transport of tracers, yet it does assume a model for the errors between the regression and the measurements (i.e. the residuals).

**So does _any_ data analysis method (although technically we are not minimizing a sum of squares, because we will always have missing data during rainless periods).**

This is similar to the choice of an objective function in traditional model calibration, and deserves attention. For example, commonly used assumptions about streamflow residuals were shown to be often violated, because of autocorrelation, non-normality, and heteroscedasticity (Schoups & Vrugt, 2010). Are the tracer residuals in this work likely to show non-normality, autocorrelation, and heteroscedasticity as well?

**This rhetorical question chooses to ignore the fact that I have _explicitly provided a framework_ for quantifying the effects of autocorrelation.**

**In any case, having taught environmental data analysis for many years, I can reassure NR that I am quite familiar with the statistical assumptions underlying regression. I am also familiar with the rather large literature on how badly one needs to violate those assumptions in order to substantially affect the results (which is really the question in practice, not whether some theoretically ideal assumptions exactly hold or not - they almost never do). The general message from that literature is that standard regression is surprisingly robust unless its assumptions are very badly violated, or unless you are trying to make inferences about extremely improbable events (very small p-values), or unless you don't es-timate standard errors (and keep them in mind in your interpretation), or unless you are overfitting (in which case you have lots of problems to worry about).**

**Regarding non-normality and heteroskedasticity, of course these could be**

looked at (at the cost of greatly expanding the length and complexity of the paper), but the results would be highly assumption-dependent. What should you assume about your sampling and laboratory errors? What should you assume about the nature of the mis-match between the assumptions that underlie the method, and the behavior of the real-world system? One could rapidly get lost in a high-dimensional assumption space.

And, again, I will point out that if these issues are a problem for this approach, they are likely to be an equally bad (or even worse) problem for many other methods of estimating transit times.

Look, let's keep this in perspective. What _other_ hydrograph separation or transit time methods have been tested as comprehensively as those that are presented in this paper? One can always ask to investigate an endless list of statistical conjectures. But why should other approaches get a free pass when they've hardly been tested at all?

Although CQ(j) and Cnew(j) are both "normalized" by subtracting CQ(j-1), there could be autocorrelation of higher order than just 1.

Of course, but the autocorrelation in the _variables_ isn't relevant; it's the autocorrelation in the _residuals_ that matters.

How does the variance of errors change with larger flashy events.

It's hard to give a general answer. But remember, these are concentrations, not water fluxes, so a lot will depend on whether big events have higher or lower tracer variance in precipitation.

3. P12, L19-20 All the benchmarking is done for a catchment without evapotranspiration.

That is false. See Section 3.6.

This points to a more general concern with the ensemble approach. No assumption is made explicitly about what happens to the tracer masses between precipitation and streamflow. This means that the method may try to find direct "connections" (in a loose statistical sense here) between tracer inputs from the past and current tracer fluctuations in the stream. Intermediary (unconsidered) processes may still be important to explain the transformation from one to the other. I especially think of processes affecting the lumped catchment tracer mass balance, which is an expression that was not considered in the approach. In that regard, how are the results expected to change if ET is actually used in the benchmark tests? Is the approach robust for real catchments where ET can be a major part of the water balance? Here I am not considering the effects of fractionation which were already dealt with, but the selective removal of certain tracer masses (associated with particular ages, i.e. different soil/groundwater mixtures) by ET, which will hence not be available for streamflow.

**As the manuscript points out, this approach will determine the lagged fractions of whatever input is sampled (presumably precipitation, but could be generalized to multiple inputs) that appear in whatever output is sampled (presumably streamflow, but could be something else). There is no requirement that these are the only inputs and outputs. Specifically, Eq. 29 says that the discharge Q_j is the sum of contributions to discharge q_jk across each time lag k, but it does not say that this is the only way that water can leave the system (that is, the q_jk do not necessarily add up to P).**

**The method is not based on an input-output mass balance for the catchment. This implies that there should be no problem if (as often happens in nature) ET is a significant fraction of the water balance, and if the age distribution of the ET flux is different from the age distribution of the discharge flux. In such a case, of course, the age distribution of discharge will be different from what it would be with ET=0, but that will be reflected in tracer concentrations that correlate differently with precipitation. Thus the proposed method will estimate the age**

distribution of the sampled output (in this case Q); it will not estimate the age distribution of the un-sampled output (ET).

4. P20, L4-6: These estimates seem to differ as much as 50% from the known values for the damped catchment and weekly data on figure 4.

**Yes, by cherry-picking one comparison (from among 8 in Fig. 4), one can find a roughly 50% discrepancy (which is actually a small discrepancy between two small numbers). This concerns the average "forward" new water fraction (the blue lines in Fig. 4g). The true value from age tracking in the model is 0.11, and the estimate from ensemble hydrograph separation is 0.07. One way to look at this is that ensemble hydrograph separation underestimates the "forward" new water fraction by about 40% (of the true value), but the other way to look at it is that the discrepancy is 0.04 in absolute terms.**

**Putting the matter differently, what would be the likely error of an *a priori* "guesstimate" of the *forward* new water fraction in this system, without any formal analysis? How does a discrepancy of 0.04 look compared to *that*?**

Many tracer data sets are at weekly resolution and come from "damped" catchments (e.g. Tetzlaff et al., 2009; Pfister et al., 2017). Data-driven approaches are by nature highly sensitive to the quality of data (e.g. variability, resolution, and measurement uncertainty). The proposed approach could thus show some limitations due to its strict data needs in some cases. This could be mentioned in the discussion.

**I don't know what NR means by "strict data needs". Strict compared to what? I've demonstrated that the method yields quantitatively realistic estimates across a wide range of catchment behaviors, with both weekly and daily data (that contain both errors and gaps). Thus the data needs here are considerably *less* strict than those of many other approaches. I will also note that among the few other benchmark tests that have been published, some make the remarkably unreal-**

**istic assumption that the input data (and sometimes also the output data) are completely error-free.**

**Thus it's not clear how to respond to NR's comment, beyond noting that of course any empirical approach requires data, and thus will depend on the quality of the data. (Conversely, any approach that does not depend on data, and thus is free of data quality concerns, probably doesn't teach us much about the real world!).**

5. P20, L19-20: A weekly sampling routine is likely to contain more "baseflow" samples which reflect older water contributions.

**That is not correct. In regular weekly sampling, baseflow will be sampled proportionally to its frequency of occurrence, because regular sampling does not preferentially include or exclude baseflow samples.**

This results in an underestimation of QFnew as shown here.

**The premise of that statement is incorrect. The underestimation in the figure has nothing to do with preferential oversampling of base flow, because that oversampling did not occur.**

Yet how can the fraction of new water with respect to discharge be under- estimated while the fraction of new water with respect to precipitation is overestimated? These quantities refer to the same mass of water in streamflow.

**Yes, they are the same mass of water, but expressed as a ratio to two different things (discharge and precipitation). One can be overestimated while the other is underestimated because uncertainties propagate differently in Eq. 9 vs. Eq. 28.**

6. P28, L2: Here it is assumed that values of Cp and CQ at all times corresponding

to indexes j or j-k are known (except a few, which require the solutions proposed in 4.2 and 4.3).

**That is not correct. There is no requirement that Cp or CQ is sampled at all times, or that the number of missing values is just "a few". The benchmark tests here assume 5% missing values from rainfall and streamflow, in addition to the much larger number of rainfall samples that are missing because there was not enough rain to make an isotope measurement. One can change the percentage of missing values from 5% to 10%, 20%, or more, of course. The uncertainties will grow accordingly (depending in part on how large m is).**

In practice it is very likely that the sampling interval is irregular, such that there is not a perfect correspondence between measurement times, and required times indexed by j or j-k. Any recommendation on how to best adjust the measurement time series so that these terms are defined properly would be welcome. Similarly, the method requires the same number of measurements in precipitation and in streamflow.

**That is not correct. Nothing in the text or the math states this or implies it. The calculations presented here assume for convenience that the time bases of the measurements are the same (that is, if precipitation is sampled weekly then streamflow is also sampled weekly). But there is no requirement that even if (for example) streamflow is sampled every week, precipitation must be sampled every week.**

How could we deal with this in various research catchments as this is often not the case?

**The only general answer is to say that one can use benchmark tests to look at the effects of various scenarios of missing data. It is difficult to generalize about the wide range of possible scenarios.**

7. P29, L12: Here it is assumed that the most recent precipitation events have more

weight in the current tracer fluctuations in the stream than older inputs.

**That is not correct. The math does not assume this, the text does not say this, and counter-examples are presented (See, for example, Figure 12b-c).**

This is implicitly reflected by the truncation of the sum in equation 30.

**That is not correct. The truncation of the sum does not assume that the remaining terms are necessarily small.**

This is also reflected by the estimated travel times that mostly stay below a few months. Although this assumption about tracer contributions is likely to be valid, catchment travel times are known to be generally in the order of magnitude of a few years and even decades (McGuire & McDonnell, 2006). This is all the more true when age estimates are based on tritium measurements (Stewart et al., 2010). Would the ensemble approach be robust in catchments where streamflow is volumetrically dominated by water older than a few months?

**The mean transit times reported by McGuire and McDonnell (2006) are mostly _baseflow_ mean transit times, whereas Fig. 3 reports mean transit times _only for rainy days_ (so that these can be directly compared to the new water fraction). The distinction is important, because baseflow mean transit times will always be longer (sometimes much longer) than mean transit times that are averaged over all flows, which in turn will always be longer than mean transit times averaged over rainy days.**

**For example, the mean transit time as shown in Fig. 3 for the benchmark model with the "flashy" parameter set is 189 days (or about six months), but the _baseflow_ mean transit time (defined for these purposes as the mean transit time when precipitation has been less than 1 mm/day for the previous three days) is 563 days, almost _three times longer_. Among the Monte Carlo parameter sets underlying Figures 2, 3, 5, and 10, baseflow mean transit times range as high as**

[Figure]

**three years or more.**

8. P29, L12: Regarding the linear algebra, how large can the truncation index m be in practice, given that computationally intensive large matrix operations are carried out? This is especially of interest since the matrices grow with the number of measurements in both dimensions, while m needs to be as large as possible for the ensemble method to work well.

**The premise is false. Nowhere does the paper say that m "needs to be as large as possible for the ensemble method to work well", for the simple reason that it isn't true. If m is too large, the TTD estimates will become too uncertain (and the standard error estimates will show this). Conversely, the ensemble method works very well for m=1 (that is, calculations of Fnew), thus rather clearly demonstrating that there is no need for m to be large.**

**Solving matrix problems is computationally intensive, but not at a scale that matters for this problem (geophysicists routinely solve matrices that are orders of magnitude larger than those that will be relevant here). Even an excel spreadsheet can do these calculations for matrices with dimensions of m>100 (I know because I've done it), and scripting languages like R or python invoke fast low-level solvers that can efficiently handle much, much larger problems. The practical limitation on m will be data, not computer power.**

In my opinion, the discussion should encourage the reader to consider if this approach shows limitations for his/her considered travel times, which may be up to a decade. Can this approach go beyond the left-hand tail of the TTDs or is it limited to the left tail?

**By definition the approach handles the left tail of the distribution, because it estimates TTDs to some maximum lag m. The question is how much of the interesting behavior of the TTD is within the range of m for which the TTD can be reliably**

**estimated. That will depend on many factors, including (a) what behaviors one considers to be interesting, (b) how many tracer samples are available, and at what frequency, and (c) perhaps most importantly, the timescales of variability in the input tracer time series (for any fluctuation-based analysis, the input must be variable on the relevant time scales). These limitations are inherent in the use of stable isotope tracers. There is no intrinsic time limit in the method itself.**

9. P33, L8: Over several years of data, doesn't neglecting 1 mm of precipitation per day sum up to a large value? It could be useful to include some discussion on the effect of that threshold on the results. Are the results highly sensitive to that choice or not?

**Remember, the method does not rely on mass balances (this is essential, because any method based on mass balance - and there are many - will be inherently vulnerable to biases from ET and from the un-representativeness of precipitation measurements). Neglecting small precipitation inputs has almost no effect on the results, except for "forward" new water fractions calculated by Eqs. 21 and 22 under certain circumstances (where a precipitation threshold is needed to avoid giving huge weight to tiny precipitation inputs). Remember, too, that the real-world catchment will also "neglect" small precipitation inputs because they will typically evaporate from the canopy or ground surface, and thus contribute next-to-nothing to streamflow and stream tracer concentrations.**

10. P39, Figure 11: Deviations between the benchmark TTDs and the estimated ones are visible here. How could these deviations be described more quantitatively to be more objective?

**Of course one could quantify the deviations in terms of a root-mean-square error or median absolute deviation, if one wanted to. This seems obvious enough**

that it doesn't need to be explicitly stated. The deviations are generally within the reported standard errors, so there is not much to be learned from them (because any measurements of the deviations would themselves have uncertainties of roughly 100%).

11. P40, Figure 12: It looks like the uncertainties are larger for the TTDs which shape is not a classical "L" anymore. The explanation given here is that the effective sample size neff (equation 13) is small because of tracer autocorrelation. Can we not say that an autocorrelation in the tracer time series is universal, as well as a shape of the TTD far from a simple "L"? This seems related to the issue described in comment (2).

**We certainly _cannot_ say that TTD's are universally far from L-shaped. In cases where TTD shapes have been evaluated with methods that are sensitive to the shape of the TTD (rather than with methods that just assume a given shape a priori), L-shaped distributions appear to be quite common in real-world watersheds.**

**The actual uncertainties (as reflected in the scatter clouds surrounding each data point) are not much larger for the humped distributions; they just look larger because the axis scales are different. However, the standard errors (the error bars) are indeed overestimated for these humped distributions. As the manuscript says, I think this has to do with the estimation of n_eff.**

12. There are not many data-driven methods that can yield non-parametric TTDs, which explains why this new approach is really beneficial to estimate TTDs. Yet, I believe that the problem solved here is somewhat similar to what Turner et al. (1987) solved as well, using Kalman filtering approaches (see the parallel between equations 30 and 31 here, and their equations 1 and 2) (see also Turner

and McPherson, 1990). They unfortunately did not detail the math behind their approach. Nevertheless, their work present time-varying average transit times, including uncertainties, also derived without assumptions on the shape of the TTDs. This is worth mentioning and comparing to the presented approach in the discussion.

**NR is correct that Turner et al.'s equations 1 and 2 and my equations 30 and 31 both express convolutions, but convolutions also underlie essentially every method for using tracer data to infer transit times. As NR notes, it is not very clear what Turner et al. actually did. Nonetheless, it's clear that their approach is not really nonparametric; instead they have apparently fitted a parametric model whose parameters are allowed to vary smoothly over time, as estimated with the Kalman filter. Nonetheless, the fact that their fits to the stream isotope data had $R^2 > 0.999$ suggest that the underlying model was massively overfitted.**

Furthermore, Klaus et al., (2015) also presented a data-driven approach that could be worth mentioning and briefly comparing to the presented one.

**The Klaus et al. analysis seems to me to be a calibrated model rather than a data-driven approach. As far as I can tell it appears to assume mass conservation between input and output (no ET), which, as NR notes, would be problematic in real-world applications. The "uniform mixing distribution" assumption also seems nonphysical, since it "gives _higher_ weight to _less_ frequent tracer concentrations" (emphasis added).**

**But most notably, Klaus et al.'s "proof of concept" simply runs their model in forward mode to generate a test time series, and then runs _exactly the same model_ in inverse mode, on exactly the same time series (without even introducing any measurement errors). This only demonstrates the mathematical consistency of the forward and inverse models. It does essentially nothing to demonstrate that the inverse model will give realistic results when applied to real-world**

data (which will *not* come from the same model, and which *will* have errors). Nonetheless, Klaus et al. conclude that, "The virtual and modeled $^{18}$O time series matched exceptionally well...", apparently not recognizing that this result is mathematically inevitable because they have tested their model against itself.

Finally Kim et al. (2016) could estimate not only TTDs but also SAS functions from artificial tracer data (of course under well controlled lab conditions). Their work is worth mentioning because they are able to distinguish the "external" variability of travel times from the "internal" one, unlike the ensemble approach presented here.

**The method used by Kim et al. cannot be applied to real-world field conditions, and thus is not relevant to the topic of the present paper.**

13. P54, L30-32: Doesn't this mean that m should be set bigger?

**Setting m bigger does not make this issue go away. One really should just take the last few lags with a grain of salt, as the manuscript says. Such edge effects often arise in various inversion problems.**

Small technical comments:

(14)    Figure 1 & 4: the colors are not consistent between the legend and the lines in the lower subplots.

**This is intentional. The problem is that if one uses a yellow color in the legend that is as light as the yellow in the figure, the line in the legend almost disappears, whereas if one uses a dark enough yellow in the figure that it can also be seen in the legend, then the figure becomes muddy.**

---

## Referee Comment (RC2) · R. Rigon (Referee) · 25 Oct 2018

This paper presents some interesting ideas. The main topic, in my opinion, is the derivation by regression of the average of the backward probability distributions which can be used to infer the (mean) catchment behavior either related to transport or to the hydrologic response. Estimating the role of antecedent conditions by studying the mean shape of the distribution functions, which is the last topic treated in the paper, is a great intuition that could become a classic.

Because I like the most the topic of the backward probabilities with respect to the one of new water/old water, I would change the title to put major emphasis on transit time

distributions than on water fractions. I believe that finding a way to characterize the backward distributions by regression is a more general and better achievement than the hydrograph separation.

In the paper there is also section on the response time distributions, also known in literature as the forward distributions. Estimations made could be biased by the example used, i.e. of having just one outgoing flux. Since there is a simple relation between backward and forward probabilities. i.e. the Niemi relation (e.g. equation 34, and the whole section 7 in Rigon et al., 2016, or equation 1 in Botter, 2012), n outgoing fluxes require to make explicit n-1 partition coefficients, a fact that it is not so evident in this work because of the simple example used as benchmark. As Rigon et al. section 7 shows, there is also an empirical version of the Niemi's finding that works at any time time t, and this is the case treated. As not clearly stated in the paper, the empirical case does not match with a pdf but to a pdf divided by the partition coefficient. In this context, It should be noted by the "astute reader" that while the backward distributions assume knowledge of all the past, forward distributions assume knowledge of all the future, which cannot be the case of the analysis under my review (see also section 6 of Rigon et al., 2016). In any case the Author needs to be a little more explicative on these facts. A brief section is dedicated to talk about evapotranspiration and fractionation effects. Evapotranspiration is not present in the model used as benchmark and the way it is introduced is not clear. Reading the paper I will not be able to reproduce the Author's results and I suggest that this part, being unessential for the present study, could be omitted. The paper is exceedingly long. Some technical parts on regressions, whilst important for research reproducibility, distract from the core topics and can be moved in my opinion to an Appendix or to some complimentary material. A further remark that embarrassed me a little. I believe that the coefficients beta of regressions, could be better understood in terms of the backward travel time distributions, and in my view, Rigon et al. 2016 can be a useful citation.

Therefore, also looking to the minor notes I make below, I think the correct judgment is
to go for major revisions (which I think will not require a lot of time though).

Riccardo Rigon

Detailed comments

Page 2 - Line 15 - Equation 1 - I think that an operational definition of new and old is required. In the subsequent text," new" is the discharge produced by the last rainfall interval (a day or a week) and "old" is the rest of the discharge. Specifying it here could be useful.

Page 6- Line 20, Equation (10). Maybe saying that this is the mean backward distribution of travel times evaluated at lag 1 could be helpful. Probably I am biased to think that way and does not correspond to the generic reader of this kind of papers. Anyway, it is my opinion.

Page 7 - Franky, I do not think sections 2.3 to 2.7 are so relevant. They probably reflect the genesis of the paper but they are full of technicalities and certainly scooped out by the more general section 4 of which these are just a particular case.

Page 26 - Section 3.6 - Effects of evaporative fractionation - This section could be interesting but where is evapotranspiration in the model used as a benchmark ? So, how could have been it evaluated Figure 10 ? The indication given in the section are not exhaustive, and I suspect that going deeper in the subject wold require major work. I suggest to take away it.

Page 28 - Definitions - All of it could be much more clearly explained in term of age-ranked functions (e.g. van der Velde, 2012, Rigon et al., 2016). I understand that the Author is a pioneer of the topic and derives everything from the scratch without being taught by anyone, but this is not useful for the general reader who will have great help from referring to those papers too.

Page 29 - Equation 31 - $q_{j,k}$ -> $q_{ij}$. Both notations are good but they should be used consistently throughout the paper
[Figure]

Page 30 - Solution method - This is essential for results reproducibility, but, at the same time, not central for understanding the concepts. I think moving it as well as section 4.3 and 4.4 to an Appendix or to the complimentary material would make the paper more readable.

Page 31 - Line 25. I would simply cut sentence 26 up to "missing values" at page 32

Page 42 - Forward time distribution. I make my points in the general comments. I believe using the work by Niemi is easier and founded on literature.

Page 43 - line 20 - Rewriting equation 57 into 58 seems to me a little pedantic.

Page 45 - Section 4.8 - I think it is sort of a Columbus'egg, a brilliant idea. The only possible objection is that results (not the method) can be biased by the use of the benchmark model.

Page 50 - Discussion - Should be shortened.

References

Botter, G. (2012). Catchment mixing processes and travel time distributions. Water Resources Research, 48(5), n/a–n/a. http://doi.org/10.1029/2011WR011160

Rigon, R., Bancheri, M., & Green, T. R. (2016). Age-ranked hydrological budgets and a travel time description of catchment hydrology. Hydrology and Earth System Sciences, 20(12), 4929.

van der Velde, Y., Torfs, P., Zee, S., and Uijlenhoet, R.: Quan- tifying catchment-scale mixing and its effect on time-varying travel time distributions, Water Resour. Res., 48, W06536, doi:10.1029/2011WR011310, 2012.

---

## Author Comment (AC4) · 27 Oct 2018

**I thank Riccardo Rigon (hereafter RR) for his comments, even if in some areas we do not agree. Below I reproduce his comments (in normal text), with my responses (in bold).**

This paper presents some interesting ideas. The main topic, in my opinion, is the derivation by regression of the average of the backward probability distributions which can be used to infer the (mean) catchment behavior either related to transport or to the hydrologic response. Estimating the role of antecedent conditions by studying the mean shape of the distribution functions, which is the last topic treated in the paper, is

a great intuition that could become a classic.

Because I like the most the topic of the backward probabilities with respect to the one of new water/old water, I would change the title to put major emphasis on transit time distributions than on water fractions. I believe that finding a way to characterize the backward distributions by regression is a more general and better achievement than the hydrograph separation.

**I suppose it is a matter of taste whether one thinks of new water fractions as special cases of transit time distributions, or whether one thinks of transit time distributions as generalizations of new water fractions, and transit time estimation as a generalization of hydrograph separation (as I do).**

In the paper there is also section on the response time distributions, also known in literature as the forward distributions. Estimations made could be biased by the example used, i.e. of having just one outgoing flux. Since there is a simple relation between backward and forward probabilities. i.e. the Niemi relation (e.g. equation 34, and the whole section 7 in Rigon et al., 2016, or equation 1 in Botter, 2012), n outgoing fluxes require to make explicit n-1 partition coefficients, a fact that it is not so evident in this work because of the simple example used as benchmark. As Rigon et al. section 7 shows, there is also an empirical version of the Niemi's finding that works at any time time t, and this is the case treated. As not clearly stated in the paper, the empirical case does not match with a pdf but to a pdf divided by the partition coefficient.

**The theoretical approaches of Rigon and Botter, including partition coefficients, are relevant to the theoretical modeling of forward transit time distributions, if those distributions measure the probability that a water drop that enters as precipitation today will leave the catchment today, tomorrow, or the day after (and so forth)** *by all possible exit mechanisms combined* **(streamflow, evapotranspiration, etc.). To take an artificially simplified example: if all streamwater leaves the catchment after seven days and all evapotranspiration (ET) leaves the catch-**

ment after one day, then obviously the forward transit time could range from one day to seven days, depending on what fraction of the input ultimately leaves via streamflow vs. ET.

**But these issues are really not relevant to the problem of _estimating forward transit time distributions from data using tracers_, because any such estimate will always be specific to _one particular exit mechanism_ (whichever one is sampled). That is, the "forward" TTD's that are estimated in my approach are _necessarily_ "forward" TTD's for the molecules that _enter as precipitation and eventually leave by streamflow_, since these are the fluxes that are sampled. We have _no information_ about the life expectancies of the molecules that leave by ET, since we have no tracer data from the ET flux. This is not a special limitation to my approach; it is a general problem that will arise in every data-driven approach, unless and until we succeed in measuring tracers in catchment-scale ET fluxes. I will revise the text to make this clearer.**

In this context, It should be noted by the "astute reader" that while the backward distributions assume knowledge of all the past, forward distributions assume knowledge of all the future, which cannot be the case of the analysis under my review (see also section 6 of Rigon et al., 2016).

**Sure, if one wants to estimate the _infinite_ tails of the transit time distributions, which my approach intentionally does not try to do, and which is _impossible in practice_ anyhow (due to accumulation of errors, long before one reaches the end of the time series).**

**But as the equations in section 4 demonstrate (and as the benchmark test confirm), one can obtain ensemble estimates of both forward and backward distributions at time steps 0..m, by looking back only m+1 time steps from each discharge time step (and one does not need to look forward at all).**

In any case the Author needs to be a little more explicative on these facts. A brief

section is dedicated to talk about evapotranspiration and fractionation effects. Evapotranspiration is not present in the model used as benchmark and the way it is introduced is not clear. Reading the paper I will not be able to reproduce the Author's results and I suggest that this part, being unessential for the present study, could be omitted.

**The point of this section is not to quantify evapotranspiration per se, but instead to test whether the estimates of new water fractions will be substantially affected by evaporative fractionation of the liquid that is left behind. While this section might appear "unessential" from RR's perspective, my perspective is different. It is too easy for skeptics to dismiss an approach like mine by saying "it doesn't account for evaporative fractionation so we don't trust any of these results." Therefore I think it is indeed essential to show that the results are not substantially biased by evaporative fractionation.**

The paper is exceedingly long. Some technical parts on regressions, whilst important for research reproducibility, distract from the core topics and can be moved in my opinion to an Appendix or to some complimentary material.

**Sure, the paper is long. But it also accomplishes a lot.**

**The "technical parts" can be skipped by readers who don't care about them (as the last sentence of the introduction clearly explains), but I think it is important that they are there. The reason is not just research reproducibility. It is also that one does not really understand what is going on without getting into how rain-free days are handled, and without seeing that the end result is not purely a multiple regression. Yet another reason is that these sections explain techniques, such as available-case analysis (Glasser's method) and Tikhonov-Phillips regularization, that are broadly useful but not widely known in our field.**

**In any case, moving these sections to appendices would actually make the paper _longer_ in total, because of the need to mention all the key points both in the main text and again in the appendices. It would also make the presentation substan-**

**tially less clear, due to the need for many "pointers" linking the main text and the appendices (and vice versa).**

A further remark that embarrassed me a little. I believe that the coefficients beta of regressions, could be better understood in terms of the backward travel time distributions, and in my view, Rigon et al. 2016 can be a useful citation.

**Equations 53 and 56, and the accompanying benchmark tests, demonstrate that the beta coefficients are _not_ , by themselves, the backward travel time distribution; instead they must be corrected for the fraction of time (or discharge, in the case of discharge-weighted TTDs) associated with time lags at which no rain fell.**

Therefore, also looking to the minor notes I make below, I think the correct judgment is to go for major revisions (which I think will not require a lot of time though).

Riccardo Rigon

Detailed comments

Page 2 - Line 15 - Equation 1 - I think that an operational definition of new and old is required. In the subsequent text," new" is the discharge produced by the last rainfall interval (a day or a week) and " old" is the rest of the discharge. Specifying it here could be useful.

**Of course in the specific context of _ensemble_ hydrograph separation, this is what "new" and "old" mean.**

**But Eqs. 1-3 describe _conventional_ hydrograph separation (not ensemble hydrograph separation). In this context it would be _factually incorrect_ to specify "new" and "old" in the way RR suggests, because this is not what they mean in conventional hydrograph separation.**

Page 6- Line 20, Equation (10). Maybe saying that this is the mean backward distribution of travel times evaluated at lag 1 could be helpful. Probably I am biased to think

that way and does not correspond to the generic reader of this kind of papers. Anyway, it is my opinion.

**I don't think this is helpful, except for readers who are already deep into the language and concepts of travel time distributions (who will already see this without having it explained to them).**

Page 7 - Franky, I do not think sections 2.3 to 2.7 are so relevant. They probably reflect the genesis of the paper but they are full of technicalities and certainly scooped out by the more general section 4 of which these are just a particular case.

**Although new water fractions may just be the m=0 case of a transit time distribution from a _theoretician's_ perspective, from a _practical_ perspective they are worth considering in their own right.**

**First of all, many catchments will have enough tracer data to reliably estimate new water fractions, but not nearly enough to estimate transit time distributions.**

**Second, the analyses presented in Figures 2-10 can be conducted easily on new water fractions, but not on transit time distributions (at least without huge masses of data).**

**Third, new water fractions will be much easier for readers to understand, because they do not require the full mathematical and statistical machinery that is involved in estimating transit time distributions. Readers will find it much easier to have an intuitive grasp of what is going on in these sections, than in the corresponding explanations of transit time distributions. Thus these sections are essential in building the reader's understanding.**

Page 26 - Section 3.6 - Effects of evaporative fractionation - This section could be interesting but where is evapotranspiration in the model used as a benchmark ? So, how could have been it evaluated Figure 10 ? The indication given in the section are not exhaustive, and I suspect that going deeper in the subject would require major

work.

I suggest to take away it.

**The description of the benchmark model makes clear that evapotranspiration is not explicitly simulated, but that instead the precipitation inputs can be considered as effective precipitation, net of evapotranspiration losses. This is a standard approach in conceptual hydrological modeling, and it is appropriate here because it eliminates the need to specify all the parameters that would be required for an explicit simulation of ET.**

**As the manuscript clearly explains, the purpose of this section is not to look at the effects of evaporation on the mass balance, but to look at how evaporative fractionation might affect the isotopic composition of discharge, and thus estimates of new water fractions. This is necessary and important, for the reasons that I outlined in my response when RR also raised this issue above in his general comments.**

Page 28 - Definitions - All of it could be much more clearly explained in term of ageranked functions (e.g. van der Velde, 2012, Rigon et al., 2016). I understand that the Author is a pioneer of the topic and derives everything from the scratch without being taught by anyone, but this is not useful for the general reader who will have great help from referring to those papers too.

**Let's be clear: we are talking about just one paragraph of very simple definitions, where to cover the same ground using the van der Velde et al. (2012) or Rigon et al. (2016) formalisms would take significantly longer. Also, in this context age-ranked functions represent unnecessary complications, because they suggest that the system somehow is aware of water's age and selects or rejects it accordingly. More generally: it is important not to introduce terms and concepts that require specifying the age or age distribution of water in storage, because this is _unknowable in practice_ .**

Page 29 - Equation 31 - q_{j,k} -> q_{ij}. Both notations are good but they should be used consistently throughout the paper.

**Thanks for catching that typo (though I may switch to the comma-delimited notation throughout, so that cases like q_{i+k,k} and q_{j,k} are represented consistently).**

Page 30 - Solution method - This is essential for results reproducibility, but, at the same time, not central for understanding the concepts. I think moving it as well as section 4.3 and 4.4 to an Appendix or to the complimentary material would make the paper more readable.

**I think it makes more sense to highlight for the reader that if they don't care to know how this is really done, they can skip to section 4.5.**

**The point of including this material in the main text (it's only about 10% of the paper) is not just for purposes of documentation; it is also so that readers know that these techniques exist, because they are broadly applicable in other areas of environmental analysis as well. I have already heard from several individuals who have appreciated seeing this material.**

Page 31 - Line 25. I would simply cut sentence 26 up to "missing values" at page 32

**Sorry, this would make the presentation incoherent and misleading. It is important for readers to understand why complete case analysis will yield biased results. It is also important for them to understand why imputation methods are also a bad idea in this context.**

Page 42 - Forward time distribution. I make my points in the general comments. I believe using the work by Niemi is easier and founded on literature.

**The Niemi approach is _incorrect in this context_, because it requires assuming that the configuration, routing, and storage volume of the system _are the same_ at high and low flows. These are assumptions that I _do not_ make, because they**

**are generally not correct.**

Page 43 - line 20 - Rewriting equation 57 into 58 seems to me a little pedantic.

**I am _not_ being pedantic. Even though Eqs. 57 and 58 are _algebraically_ equivalent, they have _fundamentally different statistical behavior_, and will lead to _different_ estimates of the beta coefficients. That is because the variance introduced by Q_j will affect the results differently when it is factored into to the right-hand side (the explanatory variables) vs. the left-hand side (the response variable).**

**The point of showing this is precisely to caution users against thinking that they can rewrite the regression equations in this paper into algebraically equivalent forms, and get equivalent results. That's not what will happen. I will add something to the text to clarify this point.**

Page 45 - Section 4.8 - I think it is sort of a Columbus' egg, a brilliant idea. The only possible objection is that results (not the method) can be biased by the use of the benchmark model.

**I do not understand this objection. There is no good way to demonstrate the potential of the method without a benchmark model.**

**Obviously the specific results that are obtained will depend on the specific benchmark model that is used, as I point out in the paper. That's why I rather clearly state in the last paragraph of section 5.2 that the results presented here demonstrate that these analyses can yield accurate results (which cannot be demonstrated with real-world data because then we cannot know independently what the right answer is), but that they should not be taken as examples of what those real-world results would be.**

Page 50 - Discussion - Should be shortened.

**Even at its current length, the discussion is less than 10% of the whole paper**

**(including appendices). In my view it is an essential aid for readers to understand several key points of context and interpretation.**

References

Botter, G. (2012). Catchment mixing processes and travel time distributions. Water Resources Research, 48(5), n/a-n/a. http://doi.org/10.1029/2011WR011160

Rigon, R., Bancheri, M., & Green, T. R. (2016). Age-ranked hydrological budgets and a travel time description of catchment hydrology. Hydrology and Earth System Sciences, 20(12), 4929.

van der Velde, Y., Torfs, P., Zee, S., and Uijlenhoet, R.: Quantifying catchment-scale mixing and its effect on time-varying travel time distributions, Water Resour. Res., 48, W06536, doi:10.1029/2011WR011310, 2012.

---

## Referee Comment (RC3) · R. Rigon (Referee) · 29 Oct 2018

Dear Editor, Dear Author,

in brief my thinking:

a - Finding a way to estimate hydrograph separation or travel time distribution averages through regression is an interesting achievement

b - Doing linear regressions, either with plenty of data or data scarcity, cannot be considered an advanced topic in 2018. Reference to appropriate literature should be enough and could substitute many pages of this paper.

[Figure]

c - Niemi's relation validity is granted always, if properly modified to account for the missing knowledge of the partitions coefficient required. In Rigon et al. 2016 there is a section dedicated to it.

d - The explanation given to account for evapotranspiration is not clear, at least to me. For what I understand, the Author did not introduced a new modelling procedure but tried to simulates the effects of fractionation on the final outcomes by introducing a sinusoidal alteration of the output signal obtained. If I did not understood properly, the Author should make an effort to express things better. If I understood properly, that was not so easy, anyway. I personally have doubts on the procedure he used, but I understand the point of the Author.

e - I think that the technique developed by the Author is worth to be published. However, it accesses a limited number "m", as called in the paper, of instants (less than the number of recorded inputs, much less, for having good statistics). This limitation have effects both on the backward and the forward probabilities estimations. The techniques does not get everything. With respect to the backward probabilities, it is NOT able to get really old water distributions, i.e with expected values of decades years old, unless the time series of appropriate length is available. Regarding to the forward expectations, the techniques does NOT allow to estimate the right partition coefficients if multiple fluxes are present, but only an approximate value for them. In both the cases, long time series in input could be required to get right answers. These facts should be clarified better to the reader and to the potential users of the methods developed.

All the best,

riccardo rigon

---

## Referee Comment (RC4) · R. Rigon (Referee) · 29 Oct 2018

Dear Editor, Dear Author,

in brief my thinking:

a - Finding a way to estimate hydrograph separation or travel time distribution averages through regression is an interesting achievement

b - Doing linear regressions, either with plenty of data or data scarcity, cannot be considered an advanced topic in 2018. Reference to appropriate literature should be enough and could substitute many pages of this paper.

[Figure]

c - Niemi's relation validity is granted always, if properly modified to account for the missing knowledge of the partition coefficients required. In Rigon et al. 2016 there is a section dedicated to it.

d - The explanation given to account for evapotranspiration is not clear, at least to me. For what I understand, the Author did not introduce a new modelling procedure but tried to simulate the effects of fractionation on the final outcomes by introducing a sinusoidal alteration of the output signal obtained. If I did not understand properly, the Author should make an effort to express things better. If I understood properly, that was not so easy, anyway. I personally have doubts on the procedure he used, but I understand the point of the Author.

e - I think that the technique developed by the Author is worth to be published. However, it accesses a limited number "m", as called in the paper, of instants (less than the number of recorded inputs, much less, for having good statistics). This limitation has effects both on the backward and the forward probabilities estimations. The technique does not get everything. With respect to the backward probabilities, it is NOT able to get really old water distributions, i.e with expected values of decades years old, unless the time series of appropriate length is available. Regarding to the forward expectations, the techniques does NOT allow to estimate the right partition coefficients if multiple fluxes are present, but only an approximate value for them. In both the cases, long time series in input could be required to get right answers. These facts should be clarified better to the reader and to the potential users of the methods developed.

All the best,

riccardo rigon
* * *

---

## Author Comment (AC5) · 2 Nov 2018

**Dear Riccardo, dear Editor,**

**I thank Riccardo Rigon for his further comments. As before, I will quote his comments verbatim (in plain text) and intersperse my responses (in boldface).**

Dear Editor, Dear Author,

in brief my thinking:

a - Finding a way to estimate hydrograph separation or travel time distribution averages through regression is an interesting achievement

[Figure]

**Thank you.**

b - Doing linear regressions, either with plenty of data or data scarcity, cannot be considered an advanced topic in 2018. Reference to appropriate literature should be enough and could substitute many pages of this paper.

**As section 4 makes clear, this is not just "doing linear regressions", even if the starting point is something that looks like a regression equation. And it's not a question of "data scarcity", but of data gaps _that would make these solutions impossible_ using standard techniques.**

**"Reference to appropriate literature" is not enough because the approach documented here _does not exist_ in the literature yet. For one thing, the covariance matrix must be altered, in two different ways, to account for the two different reasons that precipitation tracers can be missing (no rainfall, and lost samples). This _has never been done before_, and it is not trivial to figure out how to do it.**

**The approach also relies on Glasser's method of so-called "available case analysis", which is unknown within the hydrology community. It would also very hard to figure it out from the original literature; the only reason I could grasp it is that I had previously re-invented this particular wheel, so I knew how it had to work.**

**One could argue, perhaps, that Tikhonov-Phillips regularization is a standard technique in geophysical inverse methods, but it is largely unknown in hydrology except in unit hydrograph papers from past decades. And those papers typically use Tikhonov's original approach, which yields biased results, whereas here I use Phillips' approach, which is much less widely used but yields unbiased results.**

**It took me nearly two years to figure this all out, and I had the advantage of having taught statistics for many years beforehand. It would be a disservice to readers to just point them to the literature and expect them to figure out how to**
**perform the analysis presented here.**

c - Niemi's relation validity is granted always, if properly modified to account for the missing knowledge of the partition coefficients required. In Rigon et al. 2016 there is a section dedicated to it.

**Sorry, that was my mistake. I thought Prof. Rigon's earlier comment was focusing on Niemi's flow-normalized time approach (which was the focus of the cited 1977 paper), and I now see that he was referring to something different. The factors of Q/P in Eqs. 27, 59, and 62 are ensemble estimates of the partition coefficients that he refers to. I will see what I can do to clarify the revised manuscript on this point.**

**In the revision, I will also address an important point that was missed in the first version, namely the need divide by the length of the time step when estimating transit time distributions. This is necessary for dimensional consistency, and also to make estimates comparable across different sampling intervals.**

d - The explanation given to account for evapotranspiration is not clear, at least to me. For what I understand, the Author did not introduce a new modelling procedure but tried to simulate the effects of fractionation on the final outcomes by introducing a sinusoidal alteration of the output signal obtained.

**This is correct.**

If I did not understand properly, the Author should make an effort to express things better. If I understood properly, that was not so easy, anyway. I personally have doubts on the procedure he used, but I understand the point of the Author.

**I will see what I can do to clarify the presentation on this point.**

e - I think that the technique developed by the Author is worth to be published. However, it accesses a limited number "m", as called in the paper, of instants (less than the number of recorded inputs, much less, for having good statistics). This limitation has

effects both on the backward and the forward probabilities estimations. The technique does not get everything.

**I never said, or implied, that it "gets everything". I can of course state explicitly in the manuscript that the approach cannot estimate TTD's beyond m lags, although this should already be clear from the math and the figures.**

With respect to the backward probabilities, it is NOT able to get really old water distributions, i.e with expected values of decades years old, unless the time series of appropriate length is available.

**This is correct. But _this is also true of every other approach_ that uses conservative tracers. It is not a question of having a long enough time series. The key problem is that even if you had a long enough time series, the tracer inputs (typically deuterium or oxygen-18) would also need to be variable enough on decadal time scales that one could separate the (highly damped) signal from the noise.**

Regarding to the forward expectations, the techniques does NOT allow to estimate the right partition coefficients if multiple fluxes are present, but only an approximate value for them. In both the cases, long time series in input could be required to get right answers. These facts should be clarified better to the reader and to the potential users of the methods developed.

**The factors of Q/P in Eqs. 27, 59, and 62 are ensemble estimates of the partition coefficients that Prof. Rigon refers to. In any case, one would not need partition coefficients to estimate the (ensemble, or "marginal") forward transit time distribution of the water that ultimately leaves by any one specified outflow (ET, for example, or discharge), as long as one has tracer time series in that outflow (and the input, of course).**

**To estimate the forward transit time of the \*entire\* precipitation input, one would need tracer data for all of the possible exit fluxes (principally stream discharge**

and ET). Thus the primary limitation is the availability of data.

I will make it clear that both the backward and forward TTD's that are estimated from the tracers reflect only the linkage between whatever two fluxes that the tracers are measured in (typically precipitation and discharge).

---

## Author Response (AR1)

Summary of changes made to "Quantifying new water fractions and transit time distributions using ensemble hydrograph separation: theory and benchmark tests"

James W. Kirchner

- I changed the text as specified in my responses to each of the comments received during the open review process. These point-by-point responses have already been published and are not reproduced in these notes.

- I clarified several additional points in the discussion (see Section 5.1).

- As promised in an author comment during the open review process, I provided a more general form of the equations needed to estimate the precipitation-weighted and discharge-weighted new water fractions and TTD's. This required four more equations to be added to the text.

- I have not included the analysis code as supplementary material, because the code that I used to generate the results in the paper is not sufficiently user-friendly. I am developing a more user-friendly version of the analysis code in R, and will release it after it has been sufficiently beta-tested within my group.

[revised manuscript text omitted]
_{\cancel{j}}} - C_{\text{older}_{\cancel{j}}}\right) = \sum_{k=0}^{m} \frac{q_{\cancel{j}k}}{Q_{\cancel{j}}} \left(C_{P_{\cancel{j}-k}} - C_{\text{older}_{\cancel{j}}}\right) \quad , \tag{32}$$

$$\left(C_{Q_j} - C_{\text{older}_j}\right) = \sum_{k=0}^{m} \frac{q_{jk}}{Q_j} \left(C_{P_{j-k}} - C_{\text{older}_j}\right) \quad , \tag{34}$$

which readers will recognize as the multi-lag counterpart of Eq. (7).

Analogous to the approach in Sect. 2, here I account for the concentration of older inputs $C_{\text{older}_j}$ using the streamflow concentration at lag $m + 1$, just beyond the longest lag $m$, with the goal of filtering out long-term patterns that could otherwise distort the correlations between $C_{P_{j-k}}$ and $C_{Q_j}$. Thus $C_{Q_{j-m-1}}$ serves as a reference level for measuring fluctuations in precipitation and streamflow tracer concentrations, analogous to $C_{Q_{j-1}}$ in Eq. (8). Adding a bias term $\alpha$ and

10 an error term $\varepsilon_j$ yields

$$\left(C_{Q_{\cancel{j}}} - C_{Q_{\cancel{j}-m-1}}\right) = \sum_{k=0}^{m} \frac{q_{\cancel{j}k}}{Q_{\cancel{j}}} \left(C_{P_{\cancel{j}-k}} - C_{Q_{\cancel{j}-m-1}}\right) + \alpha + \varepsilon_{\cancel{j}} \quad , \tag{33}$$

$$\left(C_{Q_j} - C_{Q_{j-m-1}}\right) = \sum_{k=0}^{m} \frac{q_{jk}}{Q_j} \left(C_{P_{j-k}} - C_{Q_{j-m-1}}\right) + \alpha + \varepsilon_j \quad , \tag{35}$$

which almost looks like a conventional multiple linear regression equation,

$$y_{\cancel{j}} = \sum_{k=0}^{m} \beta_k \, x_{\cancel{j}k} + \alpha + \varepsilon_{\cancel{j}} \quad , \tag{34}$$

$$y_j = \sum_{k=0}^{m} \beta_k \, x_{jk} + \alpha + \varepsilon_j \quad , \tag{36}$$

where

$$y_{\cancel{j}} = \left(C_{Q_{\cancel{j}}} - C_{Q_{\cancel{j}-m-1}}\right) \quad \text{and} \quad x_{\cancel{j}k} = \left(C_{P_{\cancel{j}-k}} - C_{Q_{\cancel{j}-m-1}}\right) \quad , \tag{35}$$

[revised manuscript text omitted]